# Variation of sediment supply by periglacial debris flows at Zelunglung in the eastern syntaxis of Himalayas since the 1950 Assam Earthquake

Kaiheng Hu[1, 2], Hao Li[1, 2], Shuang Liu[1, 2], Li Wei[1, 2], Xiaopeng Zhang[1, 2], Limin Zhang[3], Bo Zhang[1, 2], Manish Raj Gouli[1, 2]

[1]Institute of Mountain Hazards and Environment, Chinese Academy of Sciences, Chengdu, 610213, China
[2]University of Chinese Academy of Sciences, Beijing 100049, China
[3]Department of Civil and Environmental Engineering, The Hong Kong University of Science and Technology, Clear Water Bay, Hong Kong, China

*Correspondence:* Kaiheng Hu (khhu@imde.ac.cn)

**ABSTRACT.** Periglacial debris flows in alpine mountains are influenced by strong earthquakes or climatic warming and play a crucial role in delivering sediment from hillslopes and downslope channels into rivers. Rapid and massive sediment supply to rivers by the debris flows has profoundly influenced the evolution of the alpine landscape. Nonetheless, there is a dearth of knowledge concerning the roles tectonic and climatic factors played in the intensified sediment erosion and transport. In order to increase our awareness of the mass wasting processes and glacier changes, five debris flows that occurred at the Zelunglung catchment of the eastern Himalayan syntaxis since the 1950 Assam earthquake were investigated in detail by field surveys and long-term remote sensing interpretation. Long-term seismic and meteorological data indicate that the four events of 1950-1984 were the legacies of the earthquake, and recent warming events drove the 2020 event. The transported sediment volume indexed with a non-vegetated area on the alluvial fan has reduced by 91% to a stable low level nearly 40 years after 1950. It is reasonable to hypothesize that tectonic and climatic factors alternately drive the sediment supplies caused by the debris flows. High concentrations of coarse grains, intense erosion, and extreme impact force of the 2020 debris flow raised concerns about the impacts of such excess sediment inputs on the downstream river evolution and infrastructure safety. In regard to the hydrometeorological conditions of the main river, the time to evacuate the transported coarse sediments is approximately two orders of magnitude longer than the recurrence period of periglacial debris flows.

## 1 Introduction

Glacier-related hazards are widely developed in alpine regions around the world, such as the Alps, Himalayas, Caucasus, Tianshan, and Andes (Huggel et al., 2004; Iribarren Anacona et al., 2015; Petrakov et al., 2007; Richardson and Reynolds, 2000; Shen et al., 2013). These hazards, including ice/rock avalanches, periglacial debris flows, glacial lake outburst floods (GLOFs), and dammed lakes, have caused substantial economic and human losses in the high mountains and their surrounding area (Bajracharya and Mool, 2009; Hu et al., 2019; Tian et al., 2017; Yu et al., 2021). Under climate change, characterized by rising temperatures and more frequent extreme precipitation events(Castino et al., 2016; Frich et al., 2007; Giorgi et al., 2016;

Luan and Zhai, 2023; Myhre et al., 2019), high-altitude regions are increasingly affected by more destructive and frequent ice/rock avalanches, as well as low-angle glacier detachments (Wang et al., 2024; Zhang et al., 2023).

Earthquakes, climate warming, geothermal heating, rainfall, and meltwater all directly trigger glacier-related hazards (Haeberli and Whiteman, 2021; Huggel et al., 2004). The Himalayan mountains, which are tectonically active and sensitive to climate change, have experienced many glacier-related disasters triggered by large-magnitude earthquakes or climate warming in recent years. For example, the 2015 Gorkha earthquake triggered a catastrophic ice-rock collapse in Nepal's Langtang Valley, causing over 350 casualties (Kargel et al., 2016). Between 2017 and 2018, multiple ice-rock avalanches occurred in the Sedongpu catchment, Milin County, Tibet Autonomous Region (TAR), China, triggering large-scale glacial debris flows that twice dammed the Yarlung Tsangpo River (Hu et al., 2019; Jia et al., 2019; Li et al., 2022). On 7 February 2021, about $27 \times 10^6$ m$^3$ of rock and ice collapsed and quickly transformed into a debris flow in Chamoli, Uttarakhand region of India, which killed more than 200 people and severely damaged two hydropower projects (Shugar et al., 2021). The rising frequency and magnitude of such disasters have profound hydrogeomorphic and socio-economic impacts on the high-altitude and surrounding regions, including sediment yield and transportation, alpine landscape evolution, river management, food and water security, hydropower utilization, and infrastructure construction (Evans and Clague, 1994; Kääb et al., 2021), leading to the challenges of transboundary hazards and international collaboration.

Periglacial debris flows, influenced by earthquakes or climatic events, are a major agent of sediment evacuation from steep slopes to rivers in high-altitude mountains. These flows result in massive ice loss and sediment transport, causing long-term impacts on the high mountain environment. The Institute of Mountain Hazards and Environment, Chinese Academy of Sciences (IMHE, CAS) reported that periglacial debris flows in the Guxiang catchment of southeastern Tibet transported a total of 200 Mm$^3$ of sediment into an upstream tributary of the Brahmaputra River between 1953 and 1999 (Wang et al., 2022). Similarly, the ice-rock avalanches of the Sedongpu in October 2018 delivered approximately 33.2 Mm$^3$ of sediment into the Yarlung Tsangpo River (Hu et al., 2019). The total mass loss caused by glacier-rock avalanches in Sedongpu between 2014 and 2018 reached > 70 Mm$^3$ of glacier and rock and > 150 Mm$^3$ of moraine deposits (Li et al., 2022). Furthermore, after the glacier detachment of the Sedongpu in 2018, a huge volume of ~335 Mm$^3$ material was eroded from its glacier bed and transported into the Yarlung Tsangpo (Kääb and Girod, 2023). Such sudden, massive sediment inputs greatly influence sediment transport capacity, knickpoint formation, river water quality, downstream floods, and delta progradation. For instance, the 2021 Chamoli event resulted in extremely suspended sediment as 80 times high as the permissible level in the Ganga River, ~900 km from the source (Shugar et al., 2021). Sediment fluxes have increased two- to eight-fold in many glacierized and peri-glacierized basins between the 1950s and 2010s (Zhang et al., 2022a). Until now, most previous studies have focused on the residence time and transport of earthquake-triggered landslide sediment at an orogenic scale in non-glacierized environments (Dadson et al., 2004; Dai et al., 2021; Parker et al., 2011; Wang et al., 2015). Little attention has been given to the sediment evacuation progress by post-seismic debris flows at a catchment in glacierized environments owing to relatively low likelihood of debris flows and absence of long-term site-specific data.

In order to investigate the long-term effects of earthquakes on sediment evacuation in a glaciated catchment, the Zelunglung (ZLL) catchment, a tributary of the Yarlung Tsangpo river in southeastern Tibet that has large areas of temperate glaciers and disturbed intensely by the Ms 8.5 earthquake in 1950, was chosen as our study case. The catchment had long-term remote sensing imagery for interpreting glacier changes and associated debris flows and relatively well-documented records of at least four historical periglacial debris flows in 1950, 1968, 1972, and 1984 since the 1950 Assam earthquake (Zhang and Shen, 2011; Zhang, 1992). The most recent debris-flow event occurred on 10 September 2020, triggered by a small-scale ice-rock avalanche. It is believed that historical earthquakes and ongoing climate warming drove such events (Bessette-Kirton and Coe, 2020; Deline et al., 2015; Stoffel et al., 2024; Zhang et al., 2022b). Field surveys were carried out before and after the 2020 event, including three aerial photography sessions on 9 September, 11 September 2020, and 21 December 2022, using a DJI Unmanned Aerial Vehicle (UAV). Dynamic process and sediment characteristics of the 2020 event were examined with the details of aerial photos and field measurements. The ZLL glacier and alluviation fan changes were interpreted with high-resolution optical remote sensing images from 1969 to 2022. The non-vegetated area of the alluvial fan was used as an index to reflect the variation of sediment supply caused by the periglacial debris flows. By integrating with historical data on neighboring earthquakes, temperature, and precipitation, we analyzed the trend of periglacial debris flows over different periods. This case study is helpful for a better understanding of the controlling factors and sediment transportation of periglacial debris flows in High Mountain Asia (HMA).

## 2 Study area

The ZLL catchment (94°56′13.4″E, 29°36′25.6″N) at Zhibai Village in China's TAR is a tributary on the right bank of the lower Yarlung Tsangpo River, originating from the west side of Namche Barwa massif (7782 m) in the easternmost part of the Himalayas. The main stream flows westward into the Yarlung Tsangpo at an elevation of 2810 m, with a local relief of 4972 m (**Figs. 1b and 1c**). The catchment extends over 41.21 km², with 17.06 km² (41.4%) covered by glaciers (RGI 7.0) (**Fig. 1c**). High lateral moraines on both sides of the main glacier divide the drainage network into the main channel, south branch, and north branch (**Fig. 1c**). The south branch, with a total length of 9.8 km and an average gradient of 275‰, originates from the southern cliff at an elevation of ~5900 m. Hanging glaciers along the ridge and freeze-thaw cycles in the cold region make the study area prone to ice and rock avalanches (**Fig. 1d**).

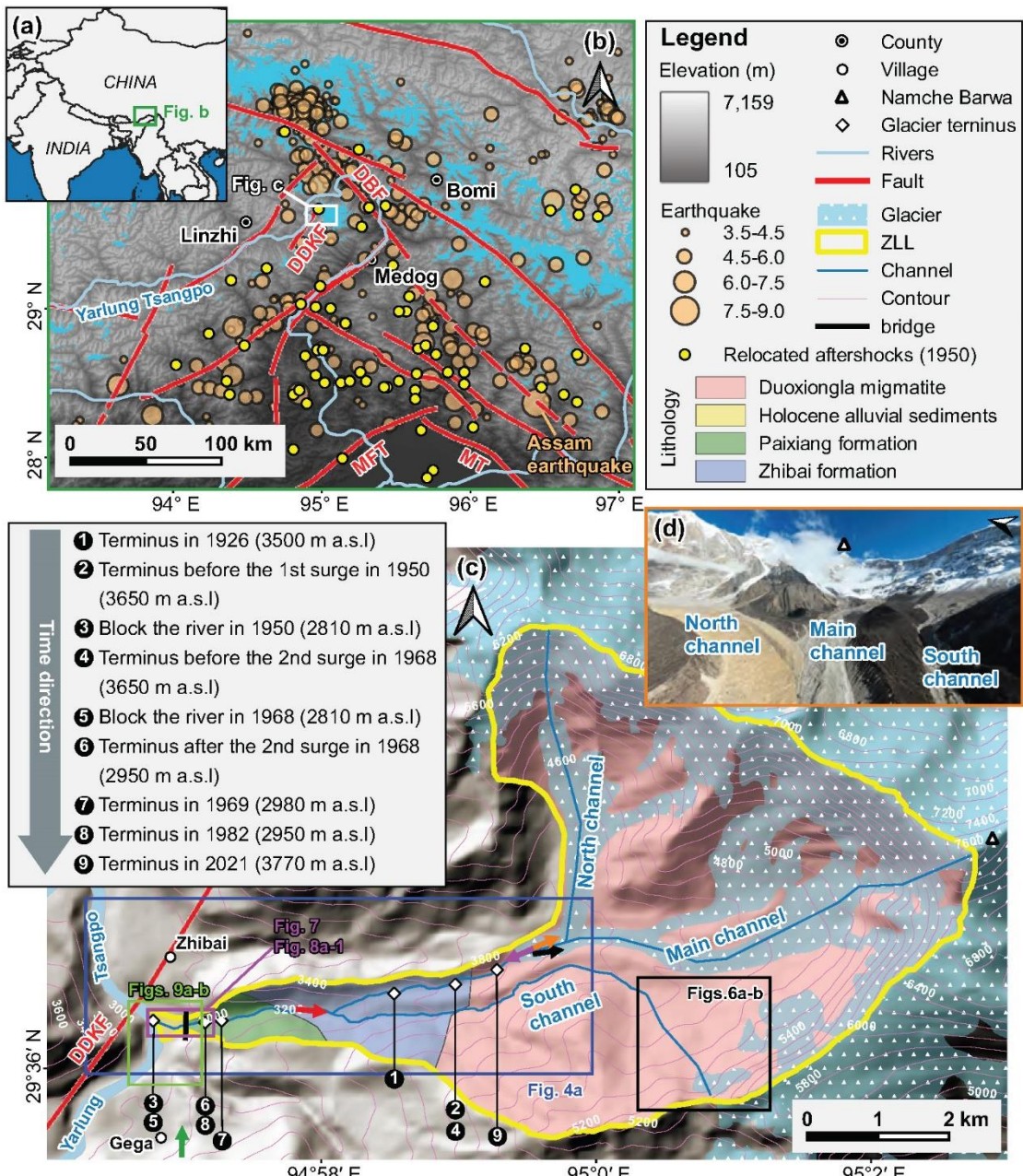

Figure 1: (a) Regional overview map of southeastern Tibet. (b) Regional settings of the eastern syntaxis of Himalayas. Fault data were obtained from Wu et al. (2024). Historical earthquakes from 1940 to 2020 were sourced from the United States Geological Survey (USGS) National Earthquake Information Center (NEIC) (https://earthquake.usgs.gov/earthquakes/search/). Relocations of aftershocks within the first four months following the 1950 Assam mainshock were obtained from Coudurier-Curveur et al. (2020). (MFT: Main Himalayan Frontal Thrust, MT: Mishmi Thrust, DBF: Damu-Bianba Fault, DDKF: Daduka Foult) (c) Topographic, geological and glacier terminus change maps of the Zelunglung catchment. The  lithological information is based on Zhang and Shen (2011), and the glacier map is derived from the RGI 7.0 dataset (RGI, 2023). The orange, rose-red, green, black and red coloured

 **arrows represent the view direction of figures 1d, 5a, 5b, 6c and 6d. (d) Aerial photo of the Zelunglung glacier and channels on December 21, 2022.**

The regional tectonic units are the Lhasa terrane, the Indus-Yarlung Tsangpo suture, and the eastern syntaxis of the Himalayas from north to south (Hu et al., 2021). The catchment lies in the eastern syntaxis, which is undergoing uplift at a rate of 5-10 mm/a (Ding et al., 2001). The exposed stratum in the ZLL is known as the Namche Barwa Group complex, which is composed of Duoxiongla migmatite, Zhibai group, and Paixiang group gneiss. The Quaternary deposits consist of Holocene alluvium at its outlet, thick layers of glacial till, and glacio-fluvial accumulation, especially hundreds of meters of huge thick moraine layers with large boulders accumulated on both sides of the main channel (**Fig. 1c**) (Han and Feng, 2018; Zhang and Shen, 2011). Many active faults are distributed around the study area, such as the Main Himalayan Frontal Thrust (MFT) and Mishmi Thrust (MT), which are considered the seismogenic faults of the 1950 Assam earthquake (Coudurier-Curveur et al., 2020), Damu-Bianba Fault (DBF) that is the seismogenic fault of the 2017 Milin earthquake (Hu et al., 2019), and Daduka Fault (DDKF) next to the ZLL downstream (**Fig. 1b**). Neotectonic activity renders this area highly susceptible to strong and frequent earthquakes.

This catchment lies in the rain shadow area of Mt. Namche Barwa, and its precipitation is controlled by the Indian Ocean's humid monsoon through the Yarlung Tsangpo Gorge (Li et al., 2024b). The climate has a strong vertical difference: semi-humid climate zone beneath 3200 m, cold temperate climate zone between 3200-4000 m, and cold climate zone above 4000 m. According to the data recorded at the Linzhi meteorological station 46.2 km west of the ZLL, the annual air temperature, with a mean value of 9.8 °C, increased at an average rate of 0.36 °C/10a from 2000 to 2021, which is much higher than the global average (Chen et al., 2015). The annual precipitation ranges from 514 mm to 972 mm, exhibiting notable inter-annual variation, with no distinct trend over the past 20 years (**Fig. 2**).

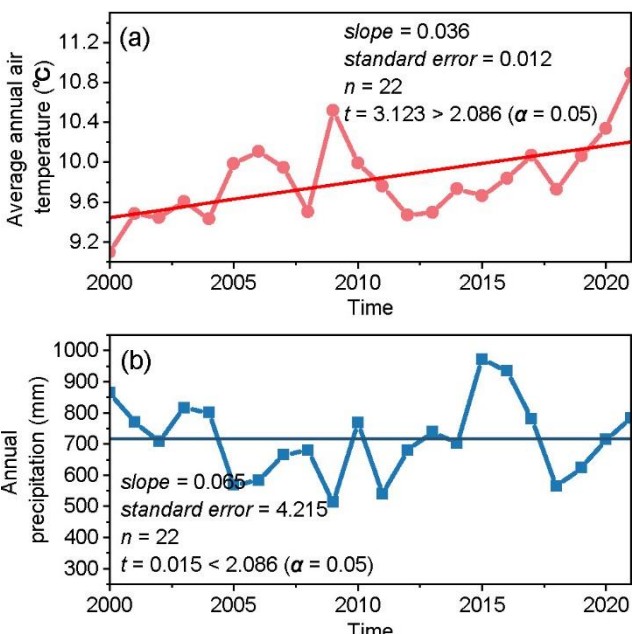

**Figure 2: Annual temperature and precipitation data from 2000 to 2021 at Linzhi Meteorological Station. (Data source:**
**https://www.ncei.noaa.gov/maps/annual/ ).**

The ZLL catchment, characterized by unique geographical and climatic conditions, experienced extensive glaciation and

frequent glacial activity over geological time. The ZLL experienced at least three glaciations in the Last Glacial Maximum

(LGM), Neoglaciation, and Late Holocene (Hu et al., 2020). The LGM moraine extended into the Yarlung Tsangpo and

dammed the river (Huang et al., 2014; Liu et al., 2006; Montgomery et al., 2004; Zhu et al., 2012). The glacier surges/debris

flows - dammed lake - outburst flood disaster events since the last glacial period also had an important impact on the landform

and paleogeographical environment of the Yarlung Tsangpo Valley (Wang et al., 2021a). The modern glaciers in this area are

strongly influenced by the Indian monsoon and are highly sensitive to climate change. Hence, the ZLL glacier has advanced

and retreated many times since the last century. The high instability and rapid changes of the glacier result in several glacier

surges or calving events. As shown in **Fig. 1c**, the glacier snout was 3500 m a.s.l in 1926 (Ward, 1926). Since the 1950s, the

ZLL glacier experienced three surges or rapid advances (Zhang, 1985, 1992). The first surge occurred on August 15, 1950.

Following the 1950 Assam earthquake, the terminus of ZLL glacier advanced from 3650 m a.s.l to the Yarlung Tsangpo at

2810 m a.s.l with a horizontal displacement of up to 4.5 km. This event destroyed the Zhibai Village completely at the mouth

of the Zhibai gully, killed 98 people, and formed an ice dam as high as tens of meters in the main river. The second surge

occurred one afternoon in August or September of 1968 (corresponding to July 1968 in Tibetan calendar) when it was sunny

(Zhang, 1985, 1992). The advance also resulted in a temporary ice dam in the Yarlung Tsangpo and deposited a glacial boulder

of 4.0×5.0×5.5 m upstream of the dam (Zhang, 1985). It is worth noting that the position of the ice tongue before the second

glacier surge has returned to the position before the first surge (3650 m a.s.l), and the peak velocity of these surges reached

1.5 km/d. After the second surge, the main glacier split into 6 segments due to differential ablation, and the terminus of the
lowest segment of the glacier was at 2950 m a.s.l. The terminus of the lowest segment was about 2980 m a.s.l in 1969 as shown
by the Corona reconnaissance satellite images (Kääb et al., 2021). The terminus of the lowest part of the glacier had probably
been at 2950 m a.s.l before 13 April 1984 when an ice mass of 80000 $m^3$ detached at 3700 m a.s.l and traveled horizontally
150 m, which was the third rapid advance of the ZLL glacier (Zhang, 1992). After that, no glacier surges or detachments were
recorded, but small-scale mountain torrents or debris flows occurred almost yearly (Zhang and Shen, 2011). At present, the
glacier terminus is about 3770 m a.s.l.

**3 Data and methodology**

**3.1 Data sources**

**3.1.1 Satellite images**

We collected 30 remote sensing images from various sources dating back to 1969, with resolutions ranging from 1m to 15m
(**Table 1**). Keyhole images before 1982 were sourced from the Keyhole reconnaissance satellites
(https://earthexplorer.usgs.gov/), which originally served as the primary data source for the United States Department of
Defence and intelligence agencies for Earth imaging. These high-resolution images provide valuable visible data from an era
before commercial satellite imagery. Images from 1988 to 2007 originated from the Centre National d'Études Spatiales (CNES)
SPOT series data (https://regards.cnes.fr/user/swh/modules/60). Images from 2009 are sourced from the RapidEye series and
Planet satellites (https://account.planet.com/), known for their short revisit periods and high resolution. To comprehensively
document the historical debris flow activity in ZLL, we diligently selected images captured after each rainy season (October
to December) whenever feasible. Due to high cloud cover and limited image availability in the study area, images from the
following year (before May) were used to fill significant data gaps (e.g., before 2000) (Li et al., 2017). Although the Landsat
satellite series may offer more continuous observational records, their relatively coarse resolution renders them unsuitable for
our study area.

**Table 1: Data sources of the satellite images used in this study.**

| No. | Date | Data sources | Resolution (m) |
|-----|------|--------------|----------------|
| 1 | 1969/12/08 | Keyhole | 1 |
| 2 | 1972/2/28 | Keyhole | 1 |
| 3 | 1973/3/26 | Keyhole | 1 |
| 4 | 1975/12/21 | Keyhole | 4 |
| 5 | 1979/4/10 | Keyhole | 1 |
| 6 | 1982/10/15 | Keyhole | 1 |

| | | | |
|---|---|---|---|
| 7 | 1988/2/20 | Spot1 | 15 |
| 8 | 1989/12/1 | Spot1 | 15 |
| 9 | 1990/12/21 | Spot2 | 12 |
| 10 | 1991/11/25 | Spot3 | 12 |
| 11 | 2000/11/17 | Spot4 | 10 |
| 12 | 2002/12/5 | Spot5 | 6 |
| 13 | 2004/12/28 | Spot5 | 6 |
| 14 | 2005/10/10 | Spot5 | 6 |
| 15 | 2006/12/21 | Spot5 | 6 |
| 16 | 2007/11/29 | Spot5 | 6 |
| 17 | 2009/12/22 | RapidEye | 5 |
| 18 | 2010/12/15 | RapidEye | 5 |
| 19 | 2011/11/23 | RapidEye | 5 |
| 20 | 2012/12/15 | RapidEye | 5 |
| 21 | 2013/12/7 | RapidEye | 5 |
| 22 | 2014/12/13 | RapidEye | 5 |
| 23 | 2015/12/6 | RapidEye | 5 |
| 24 | 2016/12/13 | Planet | 3 |
| 25 | 2017/12/11 | Planet | 5 |
| 26 | 2018/12/13 | Planet | 3 |
| 27 | 2019/12/7 | Planet | 3 |
| 28 | 2020/12/10 | Planet | 3 |
| 29 | 2021/12/12 | Planet | 3 |
| 30 | 2022/12/10 | Planet | 3 |

**3.1.2 Earthquake and climate**

Earthquake and climate datasets were used to investigate the potential linkages between these factors and debris-flow
occurrence. Earthquake records within approximately 400 km of the ZLL catchment during 1940–2020 were obtained from
the United States Geological Survey (USGS) National Earthquake Information Center (NEIC)
(https://earthquake.usgs.gov/earthquakes/search/). In addition, gridded mean values of annual mean air temperature, summer
air temperature, annual precipitation, and summer precipitation for the ZLL catchment during 1940–2021 were derived from
the 1-km monthly precipitation and mean temperature dataset for China (1901–2021) (Peng, 2019, 2020). The reliability of
these datasets has been verified against 496 independent meteorological observation stations across China (Peng et al., 2019) .

## 3.2 Methodology

This study combines field surveys, aerial drone photography, and satellite imagery analysis to investigate debris flow events in the ZLL catchment. High-resolution orthoimages and digital surface models (DSMs) were generated to assess terrain changes caused by the most recent debris flow. As volume data are unavailable for the four historical debris-flow events, the non-vegetated area (NVA) was used as a proxy for sediment volume for time series analyses. The integration of these methods provides detailed insights into the debris flow history and its influencing factors.

### 3.2.1 Field surveys

We conducted three field surveys at ZLL between 2020 and 2022. The first survey (September 9, 2020) employed a *DJI MAVIC-2* UAV to perform geomorphological photogrammetry of the downstream channel and alluvial fan. During the second survey (September 11, 2020), we combined low-altitude UAV photogrammetry with measurements from an *IMETER LF1500A* laser rangefinder to characterize the downstream channel morphology, particularly near Zhibai Bridge, and to analyse debris-flow erosion and deposition patterns. UAV photographs also provided close-up views of inaccessible upstream sections. Tape measurements were used to determine bridge displacement and boulder sizes on the fan, while low-altitude UAV orthophotos were used to support post-event interpretation of boulder distribution. Fine-grained deposits (< 100 mm) were sampled from the fan apex for laboratory analyses. The third survey (December 21, 2022) used UAV imaging to generate a complete 3D view of ZLL (**Fig. 1d**).

### 3.2.2 Drone image interpretation

We employed Pix4DMapper and Arcmap10.8 to generate the UAV digital orthophoto maps (DOMs) and DSMs, as well as to perform DSM differencing. As ground control points (GCPs) were not deployed during drone photography, we generated the DSM and DOM for September 9 in Pix4DMapper. Subsequently, 20 relatively stable points, unaffected by debris flow events, were selected as GCPs in Arcmap using the September 9 DOM as a reference. These control points were then applied in Pix4DMapper to generate the DSM and DOM for September 11. The DSMs of difference (DoD) analysis was subsequently conducted in Arcmap. To determine the uncertainty in our DoD differencing result we followed methods outlined by Shugar et al. (2021). Fifteen stable areas on old debris flow terraces adjacent to the valley floor, mainly roads and unseeded farmlands, were identified. The standard deviation of DoD values within these areas was calculated and used to estimate a two-sigma DoD uncertainty. We assumed that the errors of all DoD grid cells have the same direction, providing a conservative estimate of the total sediment volume evacuated from the catchment (Anderson, 2019).

Utilizing post-event DOM with a resolution of 0.175 m captured on September 11, we visually interpreted the spatial distribution of particles along a 1,125 m reach from the edge of the alluvial fan to the upstream channel on Arcmap10.8. High resolution and accurate color representation of the drone aerial images enabled reliable identification of coarse particles (long axis larger than 50 cm). The interpretation results were compared with measurements obtained with a caliper during the 2022

field survey. The long axis of the equivalent ellipse of these particles represents the particle size. Coarse particles were
categorized into four size ranges: 50-100, 100-300, 300-600, and >600 cm. Spatial statistics of these particles were calculated
at 25-m intervals along the central flow line.

### 3.2.3 NVA interpretation

The inundation of debris flow on the alluvial fan often destroys vegetation cover and causes the affected area desertification.
Generally, the NVA depends on the flow magnitude. So, the NVA of the alluvial fan shortly after a glacial debris flow can
serve as a proxy of the volume of transported sediment. We employed a visual interpretation approach to delineate NVAs
within the ZLL alluvial fan (**Fig. 3**). The Keyhole black and white photos and the SPOT single-band black and white images
showed distinct tonal differences between vegetated and unvegetated areas. In the true-color images obtained from RapidEye
and Planet, the NVA boundaries were highly conspicuous. Based on differences in color, tone, texture, and shading between
vegetated and non-vegetated areas in satellite imagery for a given year, we delineated the debris flow inundation zone (i.e.,
NVA) for that year. When subsequent debris flows occurred and extended beyond the gully outlet, three scenarios were
observed: (1) the subsequent debris flow was of a larger magnitude, exceeding the previous inundation zone; (2) the subsequent
debris flow was smaller in magnitude but caused damage to newly established vegetation; and (3) the subsequent debris flow
was very small in scale, confined to the channel or a very limited area along its banks, and did not affect vegetation. The third
scenario is more appropriately classified as a minor seasonal flood with negligible sediment transport compared with debris
flows. Therefore, in our interpretation and statistical analyses, the NVA was restricted to the first two scenarios (**Fig. 3**). The
ZLL interpretation zone was limited to the region between the two adjacent confluences of its upstream and downstream
catchments with the main river.

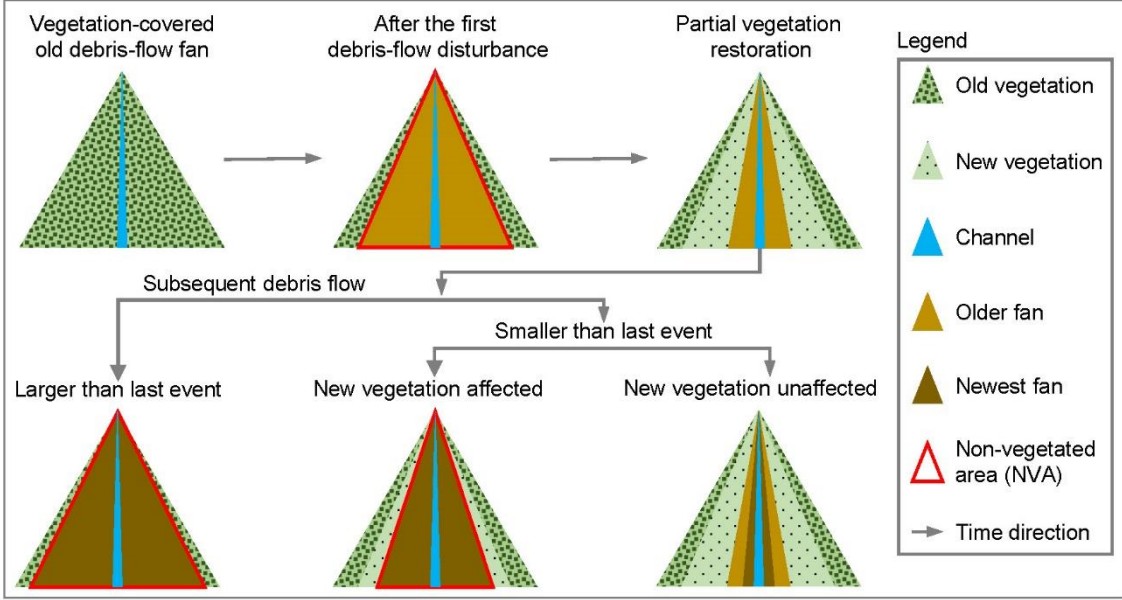

**Figure 3**: Conceptual illustration of non-vegetated area (NVA) interpretation

Due to potential misalignment between remote sensing images from different sources, image matching was performed before manual delineation of the NVAs (Cui et al., 2022). To eliminate the errors of geospatial locations of the images from different sources, we used the 2020 Planet image as the reference image and selected ground control points with clear markers on this image, such as road junctions, rivers, and typical topographic points. Third-order polynomial transformation was applied to match the images from other sources accurately with the 2020 image, ensuring a positional error of less than 20 m relative to the reference image. The original Keyhole images without geographical coordinates and projection system information were georeferenced with the 2020 Planet image with the ground control points. We assume that the visual interpretation error of NVAs is approximately one grid cell on either side of the boundary. Moreover, we verified the interpretation results of the remote sensing images with the UAV orthoimages.

## 4 Results

### 4.1 Multi-periodic glacial debris flows

Glacier surges or ice-rock avalanches can be transformed into debris flows that deliver massive amounts of sediment into the river or deposit on the alluvial fan. Four large-magnitude debris flows accompanied by glacier instability occurred in 1950, 1968, 1973, and 1984 (Zhang, 1992; Peng et al., 2022). The 1968 event caused significant deposition in the alluvial fan, characterized by a rough surface and indistinct channels (**Fig. 4-a2**). The magnitude of the 1950 event was perhaps more significant than that of the 1968 event. According to Zhang (1992), the detached glacier in 1950 climbed over the ~80 meters lateral moraine on the north at an elevation 4000 m and traveled downstream along the Zhibai gully (**Figs. 4-a1 and Fig. 5a**). Based on the erosional scar photo on the lateral moraine (Zhang, 1992) and the 2022 UAV photo, the residual depositional area of the 1950 event in the upstream gully is ~ 65,000 m$^2$ (**Fig. 5a**). Although the glacier detachment happened in ZLL in 1950, most of the sediment deposited in the Zhibai channel and its alluvial fan. Fine sediment from the catchment could be quickly transported downstream by river flows, but most coarse sediment is still left on the bank or the alluvial fans. There are two terraces on the banks of the main river along the confluences of the ZLL catchment and Zhibai gully (**Fig. 5b**). T1 and T2 terraces are ~10 m and ~ 150 m above the river level, respectively (**Fig. 5c**). The 1950 and 1968 events completely dammed the Yarlung Tsangpo (Zhang, 1992). Compared with the 1969 Keyhole image (**Fig. 4a**), it was likely that the T1 terrace is the residual dam of the 1968 event. The debris flows in the 1950 glacier surge event eroded the T2 terrace (**Fig. 4-a2**), which implies that the T2 terrace formed before 1950. The residual inundation area of the 1950 event is ~0.78 km$^2$ (**Fig. 5b**). From the 1972 and 1973 images, it was observed that fresh debris deposits inundated the north part of the fan and did not go beyond the 1968 accumulation zone (**Figs. 4-a2, 4-b2 and 4-c2**). The same lobes and deposition boundary and the marked collapse of the terminal glacier (**Figs. 4-b3 and 4-c3**) indicate that the so-called 1973 event mentioned by Peng et al. (2022) likely happened in 1972. The fan in December 1975 exhibits significant brightness variations (**Fig. 4-d2**), with pronounced

channelization above the glacier (**Fig. 4-d1**), suggesting possible debris flow activity prior to this time. Compared with 1975, the fan in 1979 displays a flatter terrain and more distinct channelization (**Fig. 4-e2**), indicating the modification of the rough fan surface by debris flow activity. This also implied that, due to limited information at the time, additional events during this period may have gone unrecorded. By 1982, noticeable vegetation had recovered in the middle part of the fan (**Fig. 4-f2**). Concurrently, accelerated glacier ablation exposed lateral moraines (**Figs. 4-e3 and 4-f3**), while the glacier terminus developed an extensive crevasse network (**Figs. 4-f4 and 4-f5**). These fractured ice bodies and moraine materials, under the impact of ice avalanche at 3700 m described by Zhang (1992), contributed to the formation of the 1984 large-scale debris flow.

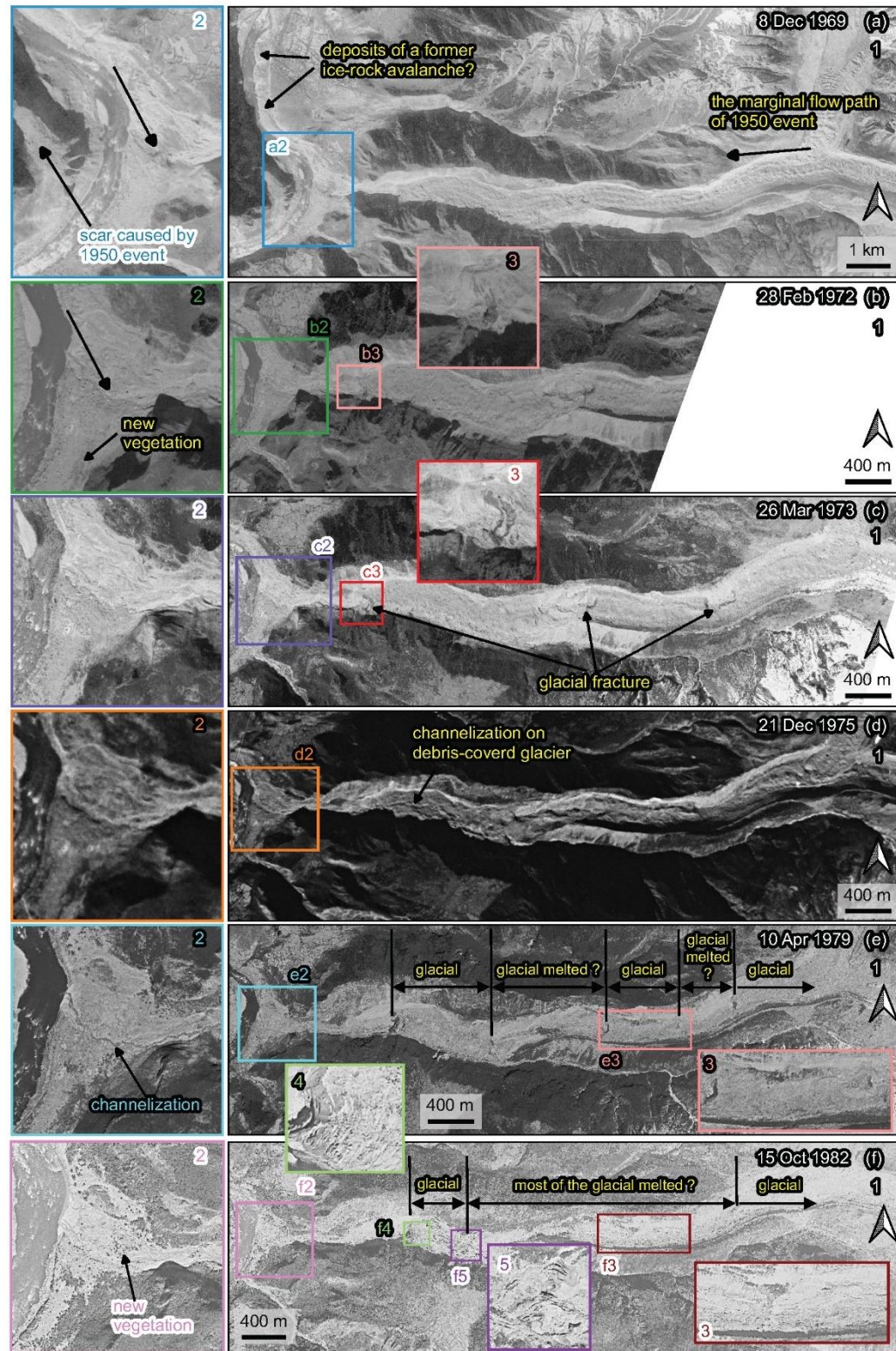

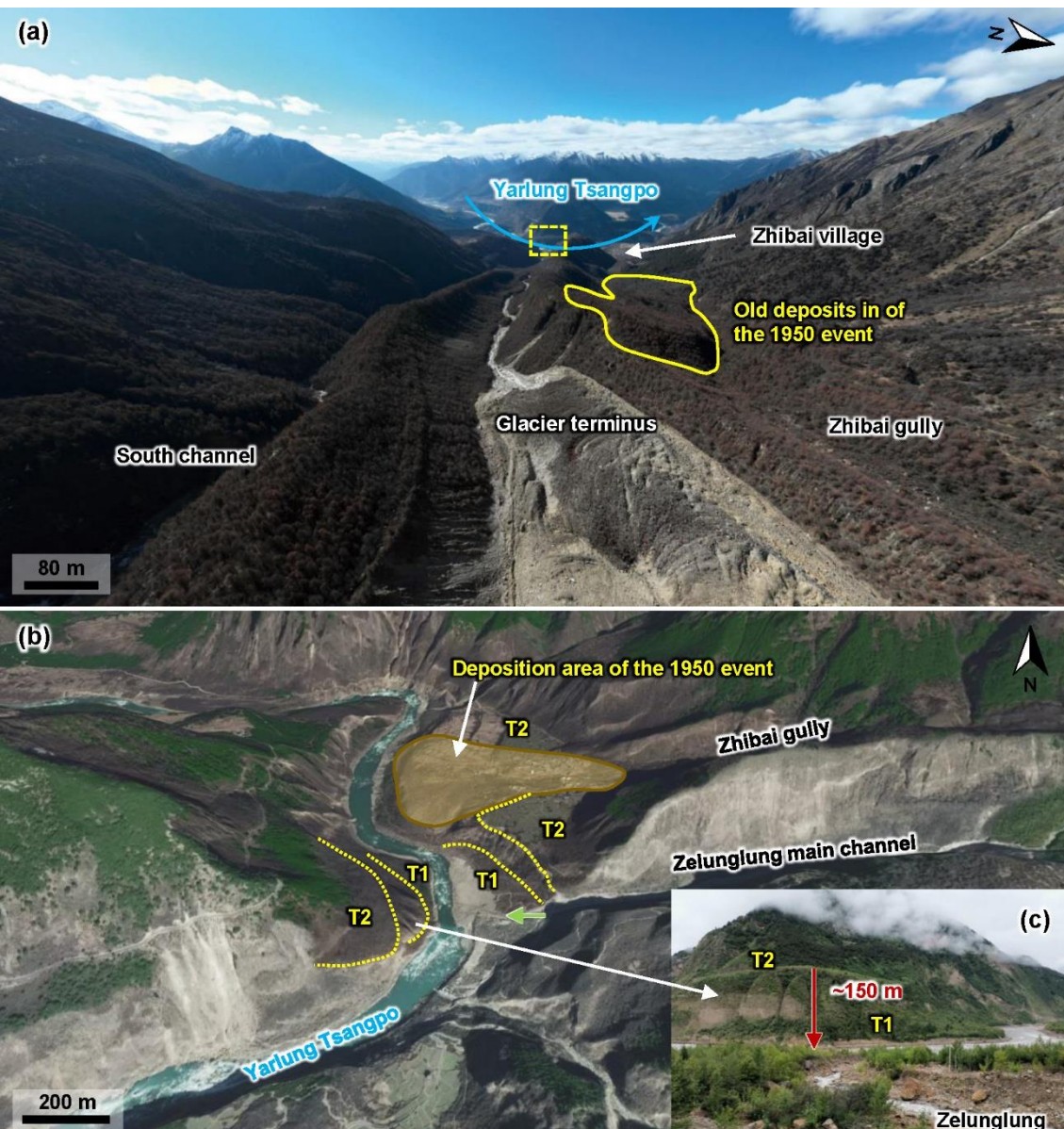

Figure 4: Variations of the Zelunglung alluvial fan and channel during 1969 – 1982. The images are taken from Keyhole reconnaissance satellites (https://earthexplorer.usgs.gov/).

Figure 5: Remnant geomorphic evidence of historical glacial debris flows. (a) Aerial photo of the Zelunglung main channel on 21 December 2022, and the old deposits in Zhibai gully left by the 1950 event (the view angle direction is denoted by rose-red arrow in figure 1c, and the dashed rectangle indicates the location of figure c). (b) Century Space satellite image (9 February 2021) illustrating two terraces on the main river banks (the view angle direction is denoted by green arrow in figure 1c). (c) Picture of the terraces on the opposite bank of the Zelunglung taken on 8 September 2020. (T1 and T2 represent the terraces formed in two different periods. The view angle direction is denoted by green arrow in figure b).

An ice-rock avalanche triggered the recently documented glacial debris flow on Sep. 10, 2020. The 2020 ice-rock avalanche initiated on the top ridge of the south branch at an elevation of 5500 m. The scar area of initiated ice and rock was $1.35 \times 10^4$ $m^2$ on the upper cliff (**Figs. 6a-b**). The initiated volume was estimated to be $7.0 \times 10^4$ $m^3$ by using the bedrock landslide area-volume empirical relationship ($V = \alpha A^\gamma$; $\alpha = 0.186, \gamma = 1.35$) (Larsen et al., 2010). In the Google image on December 4, 2017 (**Fig. 6-c2**), it could be seen that there was a protruding rock mass on the cliff below the unstable ice-rock block. The rock mass developed many lateral cracks, and the top was covered with fresh, weathered materials, indicating freezing severe weathering. The fallen ice-rock block partially disintegrated and impacted colluvial deposits on steep hillslope below the cliff at elevations 4570–4800 m, forming a muddy fresh area of 0.134 $km^2$ (**Fig. 6b**). It is noted that there was an ice-rock residual of $\sim 7.14 \times 10^3$ $m^2$ left under the cliff (**Fig. 6-b3**). When the debris flows traveled downstream, parts of old channel sediment and lateral moraines were eroded while some of the flow mass was deposited on the banks. The flows also triggered many small landslides on both banks of the middle stream (**Fig. 6d**). The UAV photo shows the influx of debris flows, transformed from the entrained sediment and melting water, exceeded the average water level of the south channel. The flow cross-section was ~ 80 m wide at the top and ~ 10 m high in the thalweg based on the UAV photo and OpenCycle topographic map (**Fig. 6-c3**). According to the description of local villagers, the first debris flow surge arrived at ZLL outlet at about 5:00 pm on September 10, and the second larger one arrived about one hour later.

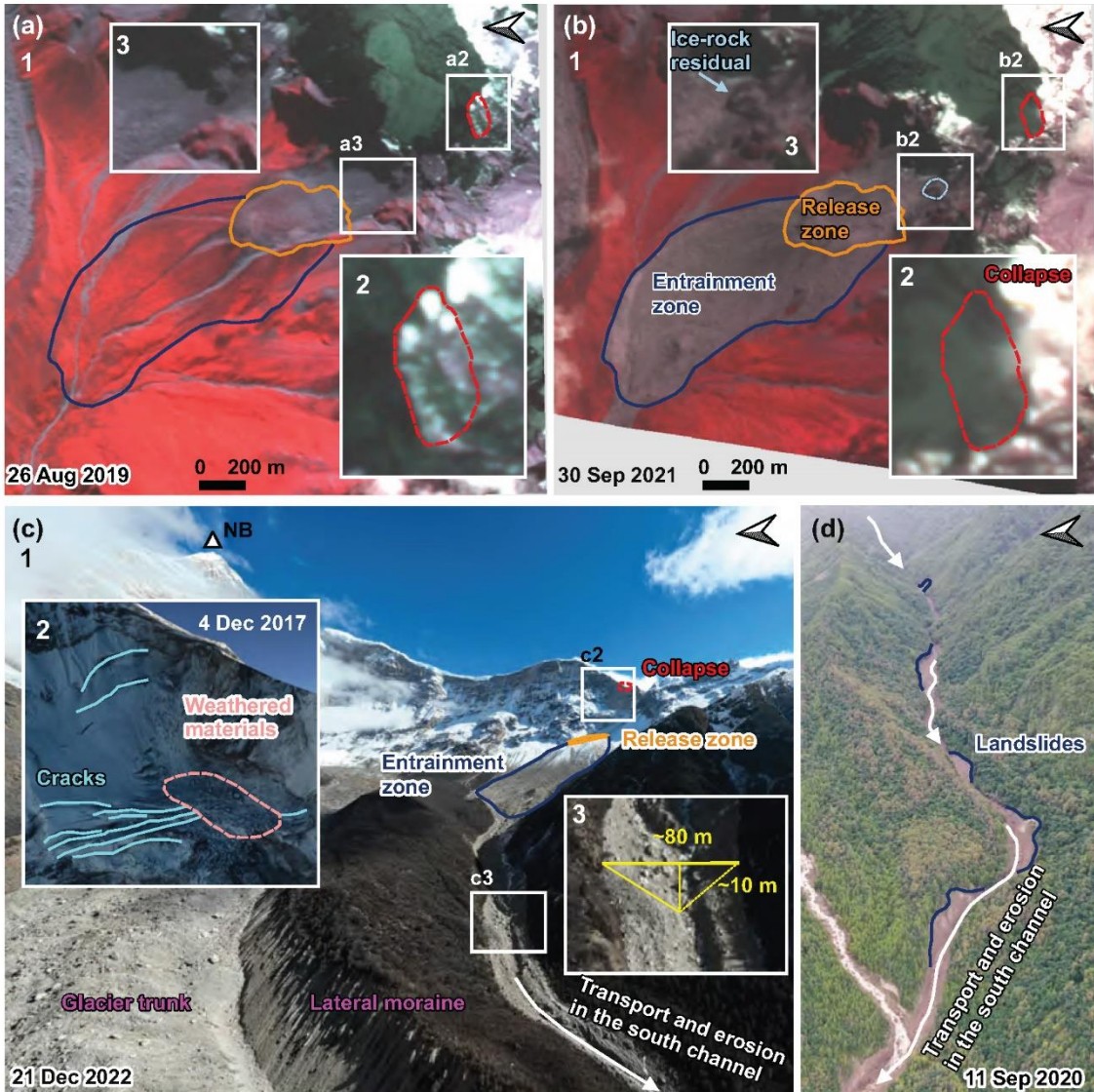

**Figure 6:** The initiation and propagation of the2020 Zelunglung periglacial glacier debris flow. (a) The planet image of the initiation area before the event. (a2) enlarged region over the pre-collapse site. (a3) Enlarge the region over the hillslope before the collapse. (b) The planet image of the initiation area after the event. (b2) enlarged region over the post-collapse site. (b3) enlarged region over the hillslope after the collapse. (base data of a-b: © 2024 Planet Labs PBC) (c) An aerial photo of the source area and the south channel on 21 December 2022 was taken by the UAV. (c2) Google Earth imagery of the initiation area on 2 December 2017 (base data: ©Google Earth). (c3) The region was enlarged over the south channel on 21 December 2022. (d) An aerial photo of the downstream channel on 11 September 2020 was taken by the UAV.

**4.2 Sediment characteristics of the 2020 event**

There is a big difference between the sediment composition in the source and depositional areas. The initiated ice-rock debris and colluvial deposits on steep hillslopes consisted of angular rocks of various sizes. However, we observed that the deposits

in the downstream areas are sub-rounded stones, and the downstream banks and channel bed were composed of sands and
boulders up to several meters in diameter (**Fig. 7**). That means most of the angular rocks resided in the upslope or upstream
channel and did not move downward. Numerous boulders were on the channel and banks before the 2020 event, as seen from
the aerial photo on 9 September 2020 (**Fig. 7a**). The in situ boulders were mobilized by the upstream flows and reorganized
spatially. The boulders were prone to move together on the flat banks such as a flat storage yard near the bridge and the fan
middle (**Fig. 7b**).

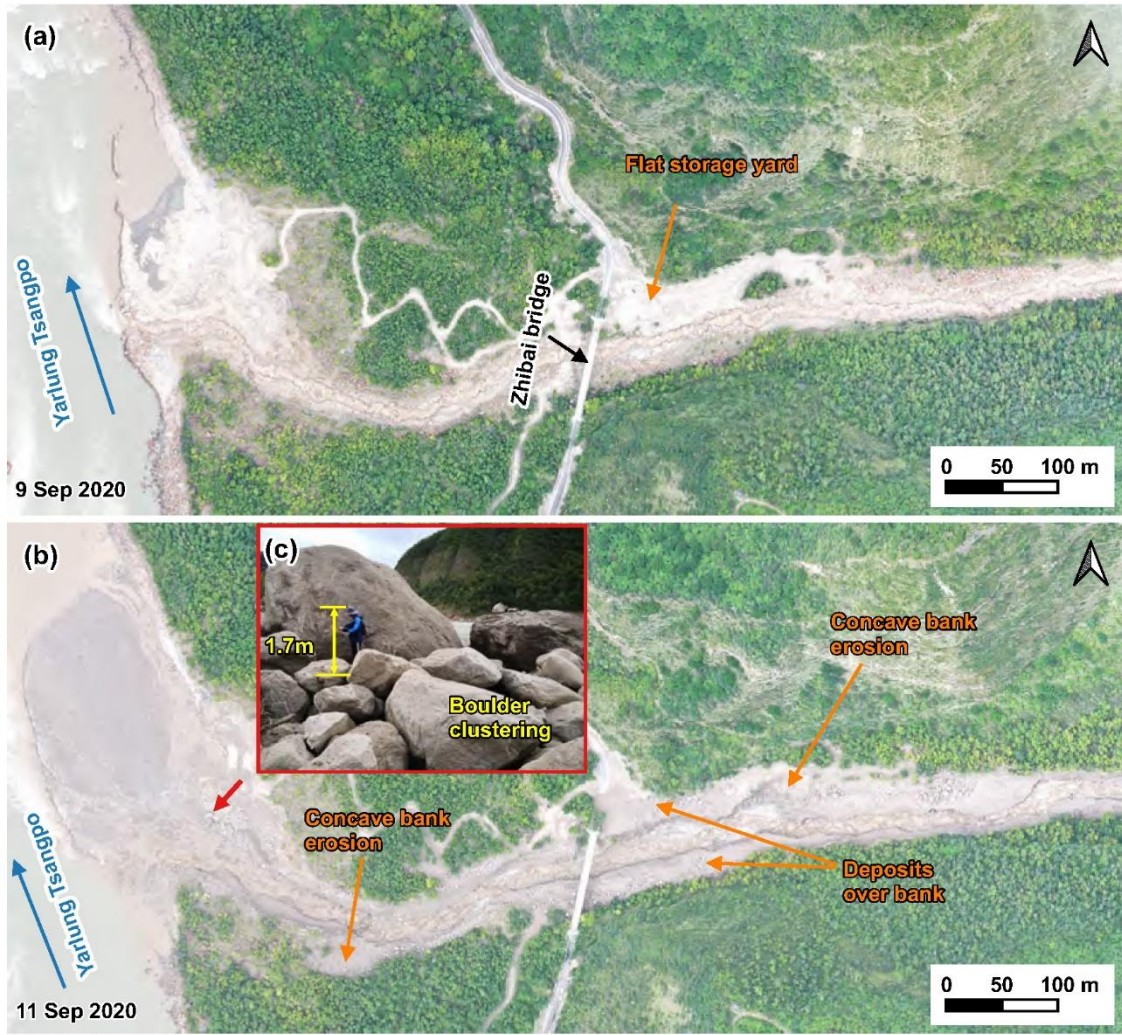

**Figure 7: Comparison of pre-and post-event aerial photos on the downstream channel and alluvial fan. (a) the UAV photo on 9**
**September 2020; (b) the UAV photo on 11 September 2020; (c) On-site picture of the boulder clustering on 11 September 2020 (the**
**camera angle direction is denoted by red arrow in figure b).**

A total of 3943 coarse particles were identified in the downstream channel and fan area, and 63% of the particles were
concentrated in three zones, A1, A2, A3 (**Figs. 8a-b**). These zones correspond to gentle banks or floodplains. The composition

of the particles in A1-A3 exhibited similar grain size distribution (**Fig. 8b**). The majority of particles were 100 and 300 cm in size, while those larger than 600 cm were the fewest. The number of particles with 100-300 cm size accounted for 77.4% of the total. Likewise, the particles with sizes of 50-100 cm, 300-600 cm, and >600 cm, accounted for 14.3%, 7.7%, and 0.6% of the total, respectively. If the particle volume is estimated with the equivalent ellipsoid volume, i.e. $V = (4\pi abc)/3$ (where $a$ is major radius, $b$ is short radius, $c$ is polar radius and equal to b), the two groups of particles with the sizes of 100-300 cm and 300-600 cm have the largest volume. The spatial distribution of these particles in the 45 segments is shown in **Figure 8c**. The same four size ranges are used (50-100 cm, 100-300 cm, 300-600 cm, and > 600 cm). The particles with the first three sizes had three peaks in A1, A2, and A3 (**Fig. 8c**). The first peak is located on the right bank highland of A1. The second peak is located on both channel sides above Zhibai Bridge. The third peak is at the top of the alluvial fan. In the A1 highland, the particle size decreased toward the outer edge of the channel (**Fig. 8-a2**), while the coarse particles in A2 were poorly sorted (**Fig. 8-a3**). In A3, the coarse particles on the surface showed the parallel superposition of two depositional units, and the particle size of each depositional unit generally decreases toward the outer edge of the channel (**Fig. 8-a4**).

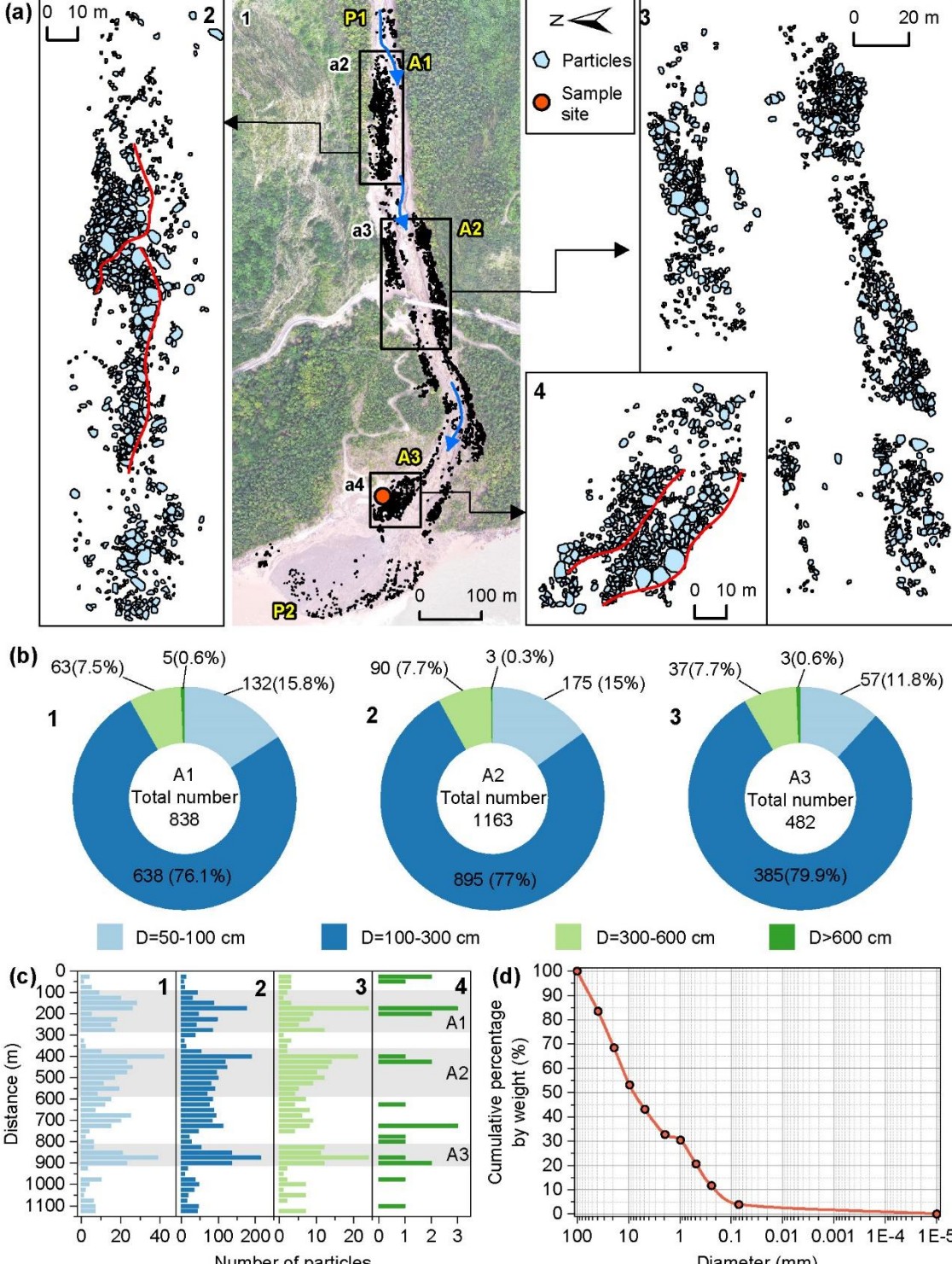

**Figure 8: Distribution of the grain size. (a) The distribution of coarse particles along the channel and alluvial fan. P1 and P2 represent the places where the count starts and ends, respectively. A1-A3 are the three main deposition sites. The blue arrow is the direction of the debris flows. The bottom image is an orthographic image taken by a drone on September 10, 2020. The locations of the enlarged regions (a2)-(a4) are shown as black boxes. (a2)-(a4) enlarged region over the three main deposition sites A1-A3. Panels (b1)-(b3) show the counts of four groups of the particles in the three main deposition sites A1-A3. Panels (c1)-(c4) show the counts of four groups of the particles in the 45 segments along the channel from P1 to P2. Particles with diameters of 50-100 cm, 100-300 cm, 300-600 cm, and particles larger than 600 cm in panels b-c are shown in light blue, blue, light green, and green. (d) Cumulative grain size distribution of the on-site sample with size < 100 mm.**

A vibrating sieve measured one sample taken from the debris-flow deposits with the size < 100 mm. The concentration of sediment finer than 0.075 mm was low, only 3.8% of the whole sample's mass (**Fig. 8d**). D50 and D90 of the sample were 8.3 mm and 62.9 mm, respectively, as linearly interpolated from the sieve-measured data. The field evidence showed that the debris flows strongly eroded the downstream channel. Lateral erosion took place nearly along the whole downstream channel. The channel width increased from 17 m to 33 m at 70 m upstream of the bridge. Concave bank erosion widened the channel by 14 m downstream. Comparing the DSMs before and after the 2020 event, the maximum erosional depth reached 20.47 m, with a mean depth of 4.17 m in the downstream channel. The maximum deposition depth was 13.51 m, and the mean depth was 3.4 m in the depositional fan (**Fig. 9b**). The mean DoD value across the 15 stable areas was −0.032 m, with an RMSE of 0.24 m, a maximum of 0.295 m, a minimum of −0.383 m, and a standard deviation of 0.247 m. Accordingly, the two-sigma DoD uncertainty was 0.493 m (**Figs. 9a-b**). Based on the DoD, we estimated that a total of $12.8 \times 10^4$ m$^3$ (two-sigma confidence intervals $10.95 \times 10^4$-$14.66 \times 10^4$ m$^3$) of debris was transported out of the catchment (**Fig. 9b**).

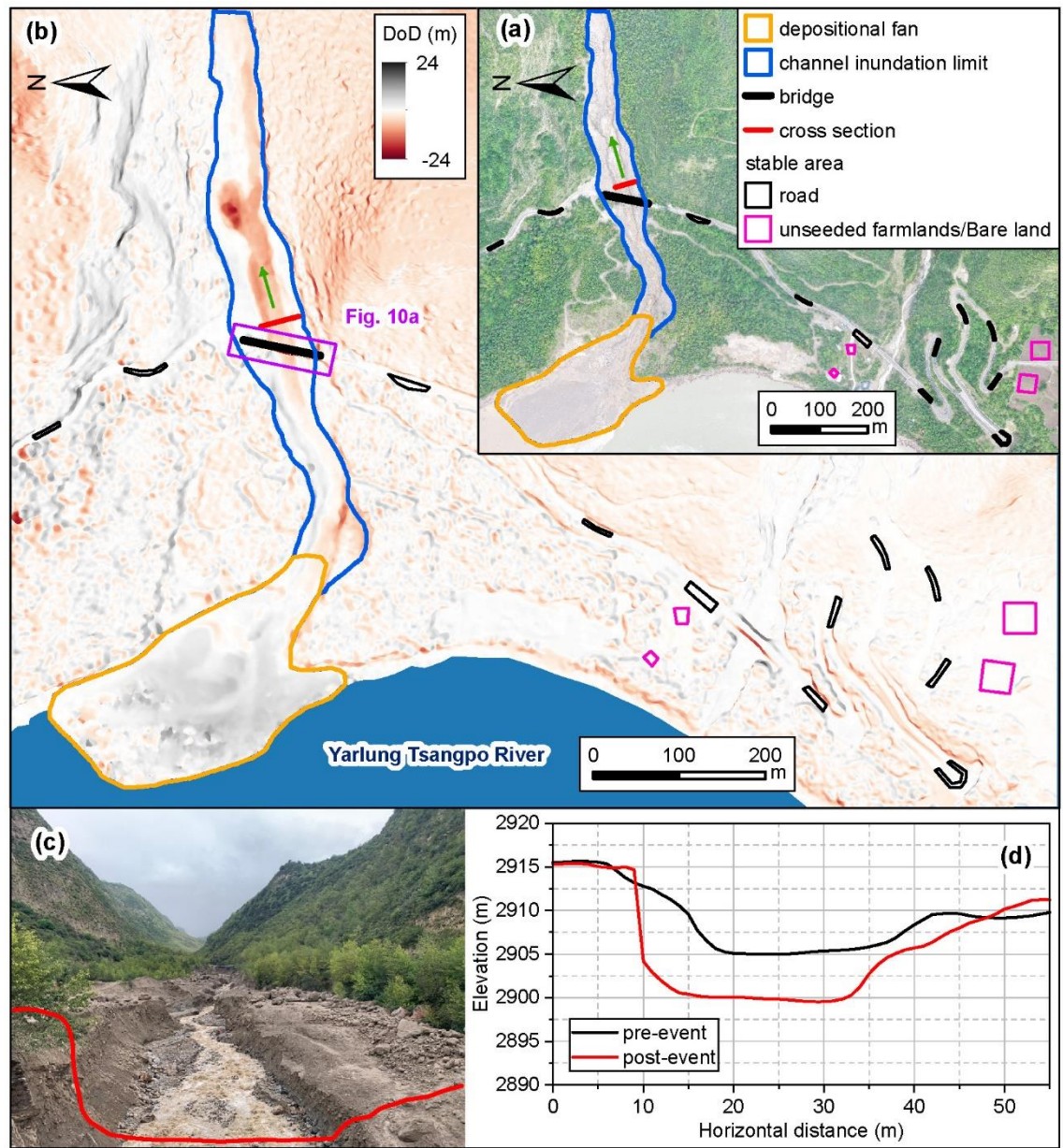

Figure 9: Geomorphic changes of the downstream channel and alluvial fan after the debris flows of 2020. (a) Post-event UAV digital
orthophoto map of the ZLL downstream area and its surrounding area. (b) Erosion and deposit depth caused by the debris flows.
(c) Photo of the channel after the debris flows (photo taken on 11 Sep 2020). The red line represents the cross-section next to the
Zhibai Bridge, and the camera angle direction is denoted by green arrow in figure a-b. (d) Cross-sections before (black) and after
(red) the debris flows.

The ZLL debris flows had a very high content of coarse particles and wide distribution. The impact of the coarse particles was

witnessed by the damages of the Zhibai bridge, a 100m long cable bridge with a steel frame (**Fig. 10a**). The foundation of the

bridge was exposed by the strong erosion capacity of the debris flows (**Fig. 10b**). The middle steel frame was intensely

impacted by run-up boulders and highly deformed (**Fig. 10c**). The concrete bridge body displaced 16 cm in vertical direction and 36 cm in horizontal direction (**Figs. 10d and e**). The velocity of the largest boulder with a size of 9.9 m was 12.6 m/s, and the impact force of the largest boulder was estimated to be $3.64\times10^6$ kN. The velocity of the debris flow at the selected cross section near the Zhibai bridge was 9.65 m/s, the peak value of debris-flow runoff was 1743.4 m³/s(**Fig. 9**)(Li et al., 2024a).

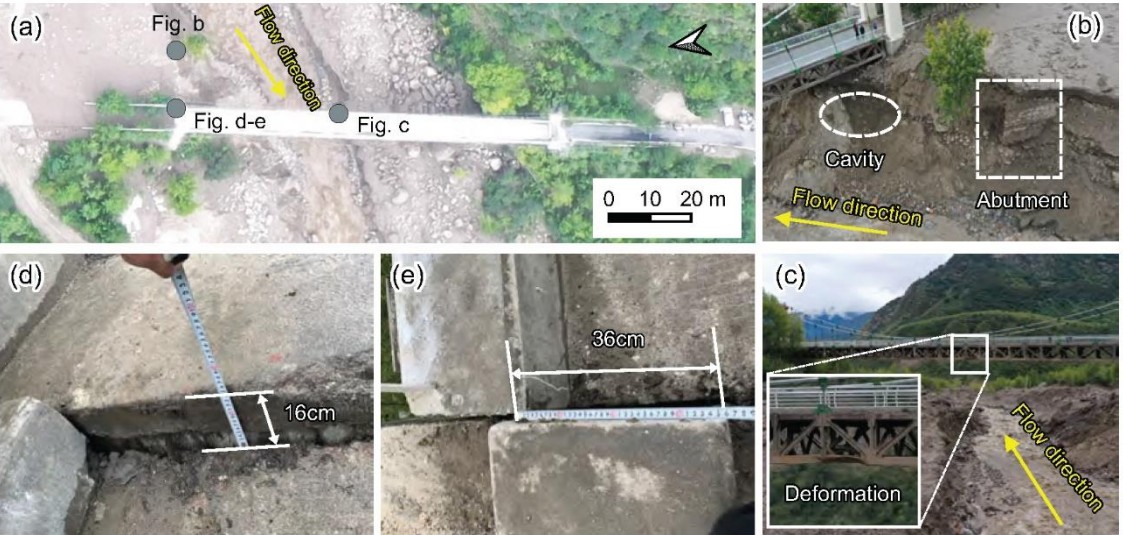

**Figure 10: Damages to the Zhibai Bridge caused by debris flows (photos taken on 11 Sep 2020). (a) UAV orthophoto of the Zhibai Bridge, with its location outlined by a blue rectangle in Fig. 9b. Gray points indicate the locations of photos (b)–(e) taken with a handheld camera. (b) Photo of the damaged bridge foundation. (c) Photo of the damaged steel frame. (d) Photo of on-site measurements of the vertical displacement of the bridge. (e) Photo of on-site measurements of the horizontal displacement of the bridge.**

**4.3 Multi-periodic sedimentation in the confluence**

From the Keyhole satellite image in 1969, the deposited debris from the 1968 event remained on the confluence and covered a 2.5 km downstream reach of the Yarlung Tsangpo River from the junction (**Fig. 4a**) (Kääb et al., 2021). During 1969 – 1979, the area of the accumulated fan remained at about 0.28 km² (**Fig. 11**). The 1972 image shows vegetation gradually developed from the edge of the accumulation fan (**Fig. 4-b2**). A new channel developed along the 1972 deposition boundary across the middle of the fan (**Fig. 4b-2**). Since then, the area without vegetation cover had reduced to 0.048 km² in 2005 and kept a slight fluctuation from 1985 to 2005 (**Fig. 11**). It indicates that only rainfall-induced small-scale flash floods or debris flows occurred during 1985-2005, which is confirmed by Zhang and Shen (2011). The NVA increased slowly, with a slight variation from 2005-2019. In 2020, the NVA abruptly increased to 0.112 km² due to the ice-rock avalanche that happened on September 10 (**Fig. 11**). The expansion of NVA in 2020 demonstrates that it was the largest debris flow event in the ZLL since 1972. At the same time, the river channel narrowed down by more than 60 meters compared to before. The multi-periodic sedimentation in the ZLL and Zhibai fans led to rapids in this reach, forming a knickpoint before the river enters the Yarlung Tsangpo Grand Canyon.

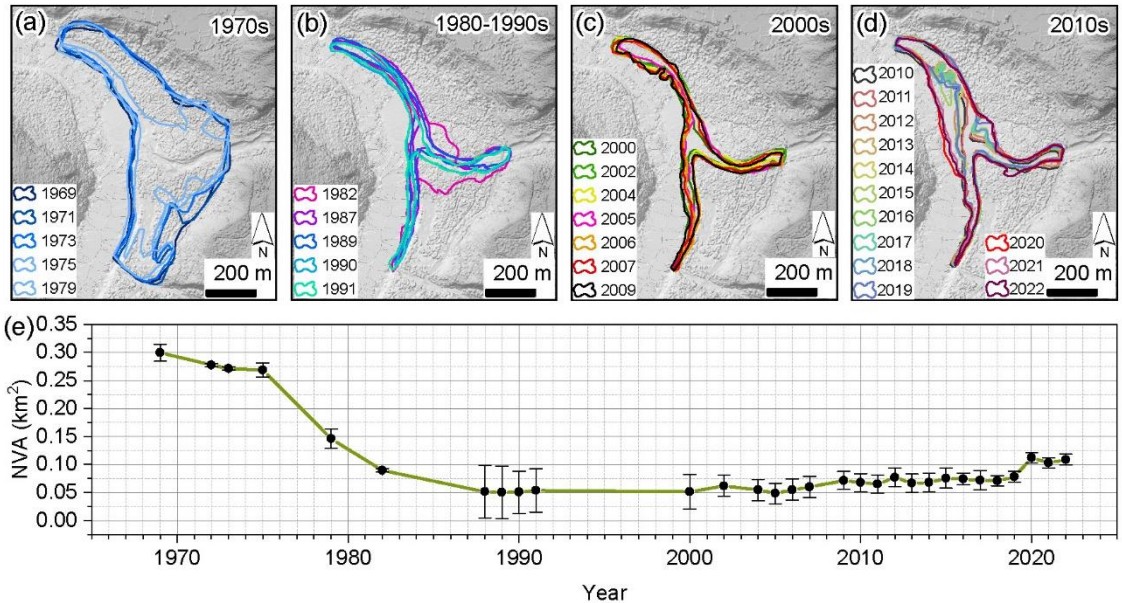

370

**Figure 11: Evolution of the non-vegetated area (NVA) in the Zelunglung alluvial fan from 1969 to 2022.** (a)-(d) The non-vegetated area maps in different decades. (e) Changes in the non-vegetated area of the Zelunglung alluvial fan from 1969 to the present.

## 5 Discussions

### 5.1 Erosion and sedimentation of periglacial debris flows

The 2020 event was characterized by strong entrainment capacity and a pronounced volume growth effect. The initiation area of the 2020 event was often covered by snow and ice, and the ice-snow melting water easily infiltrates into the debris-ice mixtures. Once the slope material was entrained into the mass flow, such a nearly saturated mixture could quickly turn into a debris flow. Peng et al. (2022) estimated a debris loss of 1.14 $Mm^3$ in the scarp area except for the initiated ice and rock. But they mistook the hillslope below the cliff as the source area of the event. That means the volume of the debris mass flowed downward into the south channel should include half of the initiated ice-rock mass and the debris loss of 1.14 $Mm^3$. The entrained volume is at least 16 times the initiated volume. In addition, eyewitnesses observed two surges in the outlet, one of which may have resulted from two ice-rock avalanches with different volumes probably happened on the ridge. But it is more likely that there was only one ice-rock avalanche during the event, but a synchronization of the ice-rock impacts in the scarp area. The blockage by large boulders and the induced landslides on the narrow channel resulted in two successive debris-flow surges, which ultimately amplified the magnitude of the debris flows (**Fig. 6d**) (Cui et al., 2013; Li et al., 2025; Liu et al., 2020).

The field evidence shows some features of periglacial debris-flow transportation that differ from fluvial transport. Periglacial debris flows can transport rocks or boulders not only in midstream steep channels but also in gentle downstream channels or alluvial fans. In the downstream channel, with an average gradient of 13.8%, a relatively high velocity (~10 m/s) enabled the flows to mobilize boulders of 5.0 meters in diameter (Costa, 1983). The transportation mode of coarse grains is a kind of "Relay-race style", one event by one event. The angularity of the fragmented rocks in the upstream reduced their mobility, and the attenuated overland flow had less transport capacity, causing most of the angular rocks to reside in the upslope or upstream channel and did not move downward. The large sub-rounded or sub-angular boulders in the lower reaches came from the middle of the downstream reaches. We guess that grain segregation happened initially, and only fine parts of the ice-rock mass and melting water traveled downward the midstream. The resident angular rocks would be rounded gradually by the periglacial stream and transported downward by the subsequent floods or debris flows.

In the downstream reach, slope and flow depth are critical factors controlling the boulder transport. Interstitial slurry among the boulders could separate from the boulders when the debris flows moved on a gentle slope or spread over an open fan (**Fig. 7c**). The interstitial slurry provided buoyancy for the boulders and reduced resistance between them and the bed. Once there was no interstitial slurry, the boulders stopped quickly. The large stones easily slowed down when the flow depth and the velocity decreased on the edges of the debris flows. When the debris flows moved to A1, the flow depth was far higher than the channel depth. Many coarse particles were left on the highland. When the debris flow entered the bend at a high speed, a large velocity difference was generated on the concave-convex bank, i.e., the super-elevation effect (Chen et al., 2009). The debris flows produced the super-elevation effect when they moved to A2, a partially curved channel. Then, some coarse particles overflowed the channel and deposited on the A2 banks. When the debris flows moved out the catchment outlet and had no boundary constraint, the other coarse particles gradually deposited from the fan top to the fan edge due to loss of kinetic energy (**Fig. 8a**). The distinct depositional units in A1 and A3 reflected the gradual accumulation of multiple debris-flow surges (Sohn, 2000; Major, 1998), which may correspond to the two successive debris-flow surges in ZLL at 5:00 pm and 6:00 pm.

Debris flows usually have steep coarse-grained surge fronts (snouts) and inter-surge watery flows (McCoy et al., 2013; Yan et al., 2023). The periglacial debris flows in ZLL had similar spatial compositions. The granular flows (coarse-grained snouts) at the fronts exerted a powerful impact on obstacles, and the inter-surge watery flows or water-rich tails with relatively low sediment concentration played critical roles in erosion. Peng et al. (2022) numerically simulated the final erosion and deposition along the flow path. The maximum erosion depth was 7.41 m at the beginning of the downstream channel. We think the simulation underestimates the erosion depth because the final erosion accumulates several erosive watery flows. It is noteworthy that, as the DSM data were acquired during the high-flow season of the Yarlung Tsangpo River, part of the deposited material may have been eroded or submerged by the river, leading to an underestimation of the actual volume of sediment transported out of the catchment.

## 5.2 Controlling factors of debris flows and sediment yield

Strong ground vibrations caused by earthquakes can intensify cracking within the ice/rock mass, ultimately leading to the formation of substantial failure surfaces (Kilburn and Voight, 1998). Additional loading by earthquakes and coseismic-ice/rock avalanches could damage the englacial conduit and subglacial drainage system. These changes can cause dynamic alterations to the glacier's thermal sensitivity, exacerbating its instability (Zhang et al., 2022b). As critical solid material sources, these highly active ice/rock masses caused by seismic disturbance are prone to avalanches, calving, detachment and remobilization to form glacial debris flows (Deng et al., 2017; Zhang et al., 2022b). It was observed that the four events in the ZLL in 1950, 1968, 1984, and 2020 were preceded by seismic activity (**Fig. 12a**). However, not all earthquakes influenced the instability of ZLL glaciers and hillslopes. Keefer (1984) presented an upper bound curve of maximum distance from epicenter to disrupted slide or fall (**Fig. 13**). In the absence of detailed fault information, we conducted a rapid and preliminary assessment of the impacts of historical earthquakes using this curve. Since 1940, only 12 earthquakes fall below the bound curve, including the 1947 earthquake, the 1950 Assam earthquake and its aftershocks, the 1985 earthquake, and the 2017 Milin earthquake. Although 13 earthquakes of $Mw > 5.1$ occurred in 1968 and 6 earthquakes of $Mw \geq 4.5$ occurred in 1984, none of these seismic events fell within the range of influence as defined by the Keefer curve (**Fig. 13**). This suggests that these earthquakes did not have a significant influence on the debris flow events of 1968 and 1984. Relocated aftershocks of the 1950 earthquake (Coudurier-Curveur et al., 2020) indicate that the seismogenic faults—the MFT and MT—extend their influence well beyond the ZLL (**Fig. 1b**). This seismic event also contributed to a prolonged period of debris flow activity, persisting for decades, in Guxianggou, approximately 50 kilometers northeast of the ZLL (Du and Zhang, 1981). The 1950 debris flow event was associated with the 1950 Assam earthquake (Zhang, 1992), and the root causes of the 1968, 1972 and 1984 events were the structural damage to the glacier and its exposure to lower altitudes with higher temperatures, both resulting from the 1950 earthquake. If including the inundated area of ~0.78 km$^2$ in 1950, the alluvial area disturbed by debris flows or floods decreased by 91% until 1990 (**Fig. 12d**). This means the earthquake effect became negligible 40 years later, as the understability of the glacier/materials caused by the earthquake may have improved. While the highest frequency of earthquakes occurred near the time of the 2020 event, they could be ignored due to their small magnitude ($Mw \leq 5.2$) and long distance (>30km) (**Fig 13**). This is because even the 2017 Mw 6.4 Milin earthquake, with an epicenter 24 km from the ZLL, had a very limited impact area (310 km², ~10 km impact radius) (Hu et al., 2019), and there were no reports or signs of such glacier-related hazards in the ZLL. However, there are direct proofs that the Milin earthquake caused the 2018 glacier surges and extra large-scale debris flows in the Sedongpu (Hu et al., 2019; Zhang et al., 2022b), 25 km downstream of the ZLL.

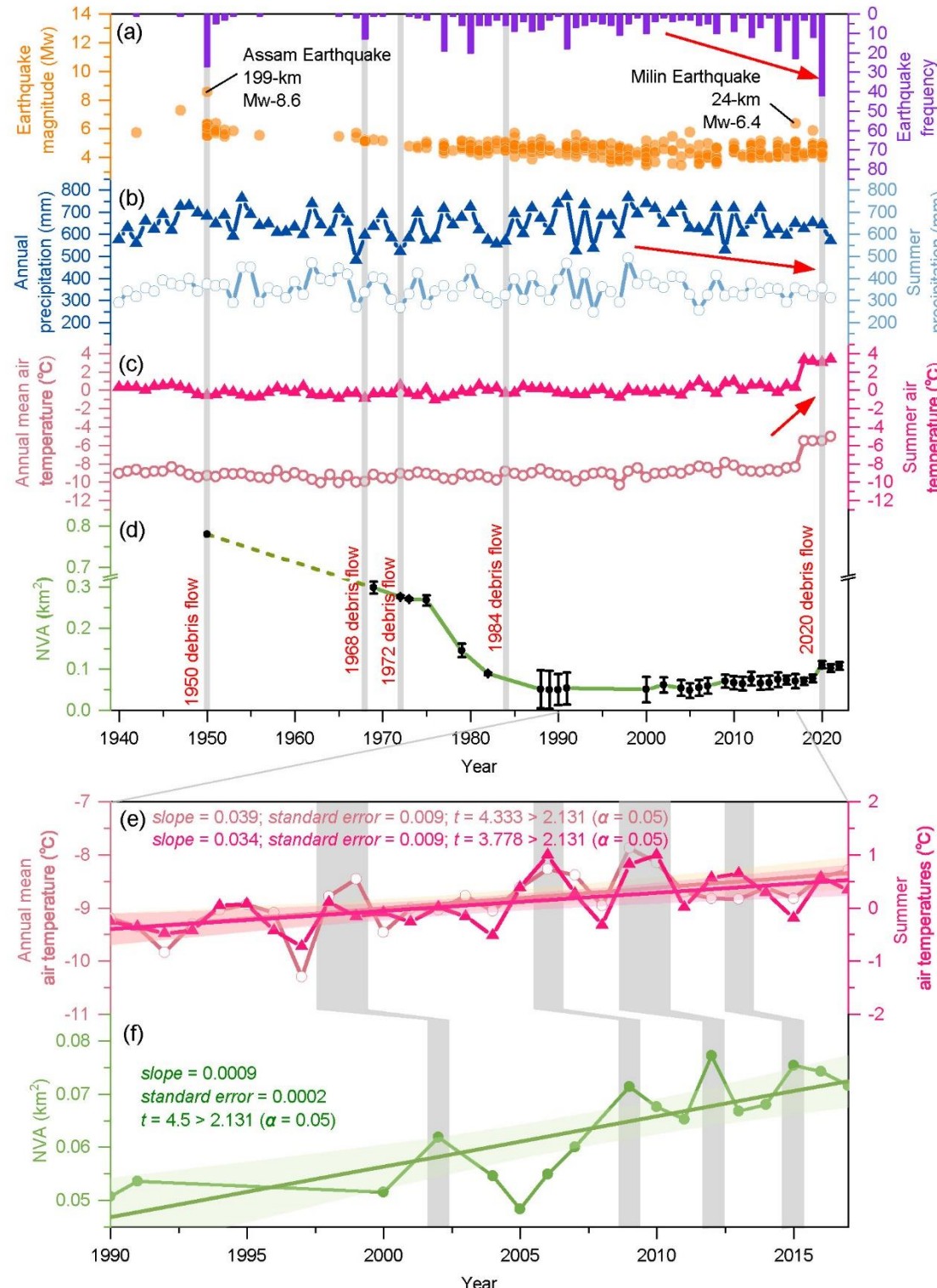

Figure 12: (a) Seismic events within a 200 km distance to the Zelunglung from 1940 to the present. (b) Changes in the annual mean and summer air temperatures in the Zelunglung from 1940 to the present. (c) Changes in the annual and summer precipitation in the Zelunglung from 1940 to the present. (d) Changes in the non-vegetated area of the Zelunglung alluvial fan from 1950 to the present (although the deposition of the 1950 event did not happen at the Zelunglung's outlet like the later events, we plot the NVA of the 1950 event as the starting point). (e) Changes in the annual and summer precipitation in the Zelunglung from 1990 to 2017. (f) Changes in the non-vegetated area of the Zelunglung alluvial fan from 1990 to 2017.

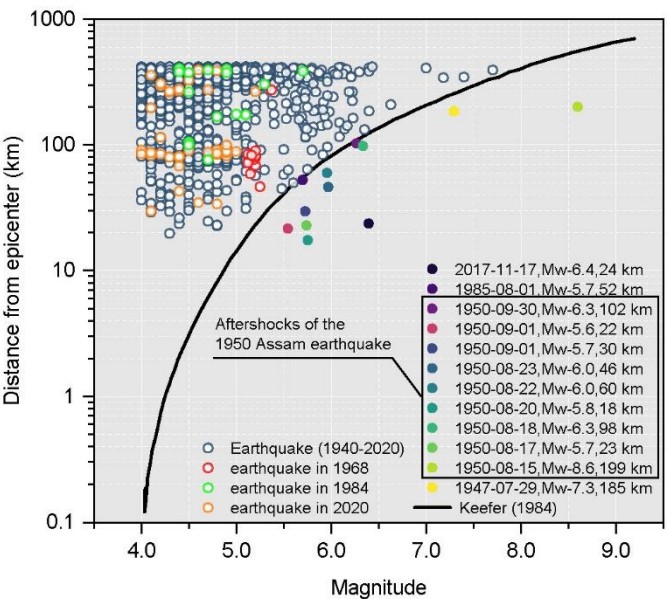

Figure 13: Distance from epicenters of the collected seismic events (1940-2020) to the Zelunglung vs. the seismic magnitude (the black solid curve refers to Keefer (1984)).

From 1940 to 2017, the annual mean and summer air temperatures at the ZLL kept relatively stable. However, in 2018, there was a sudden and significant increase in the annual mean and summer air temperatures, with an amplitude exceeding 2.5 °C. Since then, the temperatures have maintained at a high level **(Fig. 12c)**. There has been no significant change in annual and summer precipitation since 1940, but a slight decreasing trend has been observed since 2000 **(Fig. 12b)**. The rates of atmospheric warming in the Tibetan and Himalayan regions are far higher than the general global warming rate since 1960, which accelerates the rates of most glaciers shrinking and ice mass loss across the regions (Shugar et al., 2021; Zhang et al., 2020). Recent studies have shown that the on-going climate warming increases the frequency of such glacier-related slope failures. For instance, the number of rockfalls per decade showed a similar growing trend with mean annual air temperature in Chamonix, Mont Blanc massif, France since 1934 (Deline et al., 2015). The frequency of non-seismic rock avalanches in the glaciated Saint Elias Mountains of Alaska was associated with above-average temperatures and is expected to continue increasing with ongoing climate warming (Bessette-Kirton and Coe, 2020). Shugar et al. (2021) suggested that the 2021 Chamoli catastrophic ice-rock avalanche and subsequence mass flow resulted from a complex response of the geologic and topographic settings to regional climate change. Figure 12f highlights four distinct NVA peaks, which likely correspond to small mountain torrents or debris-flows, as suggested by Zhang and Shen (2011). These NVA peaks exhibit a lag of 2–4 years

relative to annual mean or summer air temperature peaks **(Figs. 12e and 12f)**. Similarly, the sharp increase in NVA caused by the 2020 debris flow event occurred two years after the 2018 warming anomaly **(Figs. 12b and 12d)**. This lag phenomenon has also been observed in other comparable regions (Deng et al., 2017; Stoffel et al., 2024). Even though there was no direct observation data of surface temperature in the ZLL highland, years of intense warming may change the thermal and hydrological conditions of the ZLL glaciers, such as the thermal regime at the rock-ice contact surface, melting rate of the surface ice and snow, englacial drainage system, fostering the instability of ice-rock blocks on the top. Previous intense seimic shaking could have widen rock fractures and reduce the ice-rock strength. It is no doubt that the 2020 ZLL event is the product of the interplay among geological movement, steep topography, and climate warming. However, based on the fact that the lag relationship between the fluctuation peaks of NVAs and temperature fluctuations from 1990 to 2020, it was likely that the 2020 event was triggered by the recent local warming.

It is evident that either earthquakes or climate change may increase the occurrence of periglacial debris flows and their sediment yield in southeastern Tibet (Du and Zhang, 1981; Deng et al., 2017; Wang et al., 2023). In the case of ZLL, the NVA closely related to the debris flows decreased until 1990 and then slightly fluctuated at a low level until 2020. That means the effects of the 1950 earthquake had decayed. The response of hillslopes or glaciers to earthquakes is immediate. Had the 2017 Milin earthquake strongly impacted the glaciers in the ZLL, ice-rock failures would have happened a few months later, like in the Sedongpu catchment (Zhang et al., 2022b). By contrast, the response of glaciers to warming will take longer. Meanwhile, approximately one month prior to the 2020 debris flow event, the maximum temperature recorded was 27°C, with a peak precipitation of only 17.5mm. Notably, on the day the 2020 debris flow occurred, the steel bridge deck remained dry, indicating that the precipitation was very light (Peng et al., 2022). On the other hand, the magnitude of the warming-driven debris flows is smaller than that of the earthquake-driven. We believe the abrupt 2.5 °C warming in 2018-2020 was dominant in initiating the 2020 ice-rock avalanche.

## 5.3 Recurrence and regime shift in sediment supply

In catchments where rainfall is the primary triggering process, such as the Multetta catchment in the Alps, debris-flow recurrence intervals have been shown to be insensitive to climate warming (Qie et al., 2024). This is because sediment supply in such catchments is controlled by weathering processes, whose rates are far lower than sediment supply from glaciers, permafrost, or other cryospheric processes. Zhang et al. (2022a) predicted that cryosphere degradation driving the increasing sediment yield in cold regions is likely to shift from a temperature-dependent regime toward a rainfall-dependent one in the next century. But in tectonically active high-altitude areas, the temperature-dependent and the earthquake-dependent regimes will alternate over the coming decades. The case of ZLL demonstrates that glacial detachment caused by the 1950 earthquake was not entirely flushed out of the channel but partly remained, providing a large amount of readily available sediment for debris-flow activity over the following forty years, thereby lowering the climate-triggering threshold for debris flows. Trends in NVA suggest that, as the legacy sediment had gradually been depleted, debris-flow magnitude and frequency had stabilized

(**Fig. 12d**), making current debris-flow activity more dependent on climate warming. A similar phenomenon occurred in the nearby periglacial Peilong catchment (Wang et al., 2021b). After decades of quiescence, sediment retained upstream following the 1981–1982 earthquakes created favorable conditions for three large debris flows between 1983 and 1985, after which activity declined. In recent decades, climate warming has promoted sediment accumulation through glacier movement and permafrost thaw, leading to highly active debris flows in 2015. Regardless of the regimes, sediment transport follows a mobilization–storage–remobilization pattern (Berger et al., 2010), and debris-flow recurrence intervals are controlled by the ease of these processes. Under an earthquake-dependent regime, the mobilization and storage phases are brief, with seismic events causing abrupt, large-scale sediment mobilization and substantial immediately available sediment storage; the remobilization of such sediment often requires only a minor triggering threshold. In contrast, under a temperature-dependent regime, sediment mobilization induced by climate-warming-driven glacier and permafrost degradation requires a relatively longer preconditioning period than in earthquake-dependent regime, and debris flows must exceed a higher triggering threshold to remobilize the sediment before it reaches a certain storage magnitude (Savi et al., 2021). Nonetheless, compared with non-glacierized catchments, ZLL still exhibits high sediment mobilization and storage capacity, and once sufficient sediment has accumulated, future hydrological changes induced by climate warming will further facilitate sediment transport (Hirschberg et al., 2020).

The period of the ZLL glacier surges has been shortening. Zhang (1985) supposed that the surging cycle of the ZLL glacier was about 20 years. According to the latest research by Guillet et al. (2022), the ZLL glacier showed signs of surge in 2004, 2005, and 2006. Moreover, there were more obvious signs of a surge in 2015 (**Fig. 14**). The interval between the last two surges was ten years, which indicates that the surging cycle of the ZLL glacier may be decreasing, and the next large-scale surge may occur in the next ten years. Furthermore, changes in the speed of glacier movement can strongly impact channel side moraines or terminal moraines and lead to slope failures (Richardson and Reynolds, 2000). The potential ice collapse area in the formation area of the ZLL catchment is 2.4 $km^2$, the rock collapse area reaches 0.96 $km^2$, and the loose moraine accumulation reaches 5.2 $km^2$ (Li et al., 2021; Liu et al., 2022). However, the 2020 debris flow was caused by a relatively small area of ice-rock collapses in the formation area, which is only the tip of the iceberg compared to the overall high-risk provenances in the formation area of the ZLL catchment. That means if intense earthquakes or extreme warming events happen not far away from the catchment, the risk of slope failures or glacier detachment on the steep slopes and ridges is high and huge amounts of sediment will be transported into the river by large-scale debris flows.

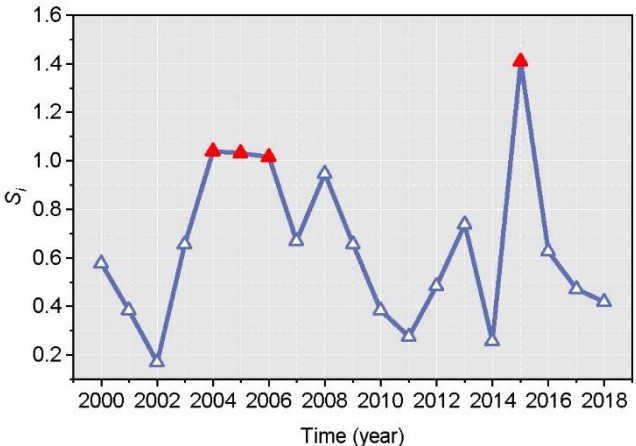

**Figure 14: Surge-index ($S_i$) of Zelunglung Glacier from 2000 to 2018. $S_i$ is a quantitative index of the surge magnitude, calculated by**

**the formula** $s_i = \dfrac{IPR_i}{k \cdot V_0}$ **, where $IPR_i$ is the inter-percentile range for year $i$, $k$ is a threshold for surge identification, and $V_0$ is the error-**

**weighted mean velocity for the study year. The years with $S_i>1$ are marked with red triangles. (Data source:**
**https://doi.org/10.5281/zenodo.5524861 (Guillet et al., 2022)).**

**5.4 Geomorphic and hydrological implications**

The moraine and old deposits on both sides of the ZLL channel provided numerous boulders for debris flows. The number of

coarse particles transported by the ZLL periglacial debris flows is very high, and there is no obvious particle sorting along the

flow path. Most of the boulders are gneiss with high hardness, and the wearing and disintegration effects are not significant

during the movement along the channel. Coarse particles are deposited on the platform at the bend and the top of the alluvial

fan, where the channel suddenly widens. Such phenomenon demonstrated that the movement, deposition, and particle size

distribution of the debris flow are not only related to the type of debris flow (Bardou et al., 2003) but also to topographic

conditions (Ghilardi et al., 2001; Zhou et al., 2019). In contrast, the deposition of the 2020 debris flow narrowed the Yarlung

Tsangpo River at the outlet of the ZLL, and the river bed was significantly elevated. The river flow hardly transports the

boulders on the alluvial fan. The peak discharge of the largest flood in the Yarlung Tsangpo recorded by the hydrologic station

at Nuxia, 40 km upstream of the ZLL, is 16800 $m^3$/s. The maximum size of the particles in such a flood is about 150 cm. The

floods capable of moving the coarsest boulders (> 600 cm) deposited on the ZLL fan should be on the order of $10^6$ $m^3$/s of

peak discharge (Lang et al., 2013). Such high-magnitude floods in the Yarlung Tsangpo were caused by catastrophic breaching

of landslide or glacial dams, e.g., several Quaternary megafloods in the middle and downstream of Yarlung Tsangpo (Hu et

al., 2018; Liu et al., 2015; Yang et al., 2022), rather than caused by monsoonal runoffs. Modern outburst floods higher than

$10^5$ $m^3$/s only happened on the Yigong River, a downstream tributary of the Yarlung Tsangpo Gorge (Hu et al., 2021).

Therefore, the time to evacuate the coarse sediments on the alluvial fan is two orders of magnitude of the recurrence period of

periglacial debris flows. The long-lived protruding fan forms a knickpoint at the confluence. The repeated glacial and landslide

dams in the margin of the Tibetan Plateau play significant roles in reducing the river incision into the plateau interior together with the moraine dams in the glaciation ages (Hu et al., 2021).

Comparable processes have been documented in other periglacial catchments. In the Sedongpu catchment, located downstream of ZLL, a sequence of ice-avalanche–debris-flow damming events between 2018 and 2024 repeatedly blocked the Yarlung Tsangpo, triggering frequent channel shifting, narrowing, and bed aggradation(Gao et al., 2023). In Peilong catchment, sustained supply of glacial debris flows aggraded the channel by ~53 m since 1983. Despite occasional incision by outburst floods, the transport capacity of the main river was insufficient to counteract the persistent input from debris flows, resulting in long-term channel aggradation(Wang et al., 2021b). In Switzerland, the 2025 Birch Glacier collapse mobilized $6 \times 10^6$ $m^3$ ice–rock mixture, entraining large quantities of debris and causing significant riverbed aggradation, instantaneously damming the Lonza River and forming a lake that posed considerable geomorphic and hydrological impacts on the downstream valley (Yin et al., 2025). In the Indian Himalaya, the Meru Bamak debris flow transported ~7.9 × $10^6$ m³ of sediment, with ~6.5 × $10^6$ m³ deposited at the glacier front. The resulting fan forced the Bhagirathi River to shift ~150 m laterally, fundamentally altering local fluvial morphology(Kumar et al., 2019). These events illustrate that debris flows in periglacial catchments, due to their massive sediment supply and extremely high energy, exert geomorphic impacts far exceeding those of rainfall-triggered events. These processes not only reshape alluvial fans and trunk channels at a local scale, but also profoundly influence river systems through damming, outburst flooding, and channel avulsion. Their impacts are characterized by both sudden catastrophic disturbances and long-term cumulative effects, underscoring the decisive role of periglacial debris flows in shaping river morphology and regulating hydrological processes in high mountain periglacial environments.

**5.5 Uncertainties**

The uncertainty of the DoD arises from multiple sources. On the one hand, the positional accuracy of the onboard GPS used to produce the UAV-derived DSMs is limited (Niu et al., 2024). On the other hand, surface characteristics such as vegetation and canopy cover, as well as flight parameters and environmental factors (e.g., illumination, shadow, and surface moisture), can also affect the quality of DSM reconstruction(Anders et al., 2020; Chaudhry et al., 2021; Kucharczyk et al., 2018). To minimize these uncertainties, several geomorphically stable ground control points (GCPs) were manually selected and used for co-registration between the two DSMs. Nevertheless, potential errors associated with GCP positioning, number, and spatial distribution cannot be entirely excluded (Han et al., 2019). Despite these limitations, our two-sigma DoD uncertainty (±0.493 m) falls within the reasonable range reported in previous studies (Müller et al., 2014; Prokešová et al., 2010). It should also be noted that, although the most recent pre- and post-event DSMs were used to reconstruct the debris-flow evacuation volume, our results may underestimate the actual evacuated material, as the UAV data were acquired during a period of high discharge in the main river, when some of the mobilized sediments might have been transported away or submerged.

Previous studies have shown that debris-flow volume is often empirically related to the extent of the inundation area (e.g., Iverson et al., 1998), which supports the use of NVA as a proxy for sediment volume. The absence of systematic depositional

thickness measurements prevents direct conversion of NVAs into absolute debris-flow volumes, so NVA only reflects relative fluctuations rather than precise values. Moreover, the empirical relationship between inundation area and debris-flow volume may vary with local geomorphic and hydrological conditions, such as fan slope and gully confinement, further complicating volume inference and limiting the applicability of uniform statistical error models. In practice, the NVA includes a fixed portion of the area inundated by the main river and is therefore slightly larger than the actual depositional area caused by debris flows. Technically, the contribution of the main river to NVA cannot be entirely excluded. Nevertheless, the riverbank line remained stable from the 1980s to the 2010s, during which no large periglacial debris flows occurred (**Figs. 11b and c**; Zhang and Shen, 2011), and the remote-sensing images we used were taken in similar seasons. It is therefore reasonable to assume that variations in river water level have little influence on changes in NVA. The one-grid cell uncertainty in the interpretation of the NVA ultimately translates into uncertainty in its representation of sediment volume (**Fig. 11e**). The magnitude of this uncertainty depends on factors such as the co-registration accuracy between the secondary and primary images, the time interval between image acquisitions, surface conditions (e.g., shadows), and the spatial resolution of the imagery (Paul et al., 2022). Although these uncertainties cannot be completely eliminated, their influence on the overall trend of NVA variation is minor and thus does not alter the main analytical results and conclusions.

While accurate estimation of sediment volume from NVA is beyond the scope of this study, we acknowledge the associated uncertainties, which warrant more rigorous treatment in future study. Future studies that integrate high-resolution LiDAR, UAV photogrammetry, or borehole surveys with field-based volume measurements could provide more robust statistical assessments of the NVA–volume relationship. The application of dense stereo-pair techniques for DSM extraction from historical and modern satellite archives also has considerable potential to provide three-dimensional constraints on sediment thickness and deposition, thereby improving the translation from inundation area to sediment volume.

**6 Conclusions**

High-magnitude sediment evacuation by periglacial debris flows is a crucial surface process that links sediment yield from high-altitude slopes to river sediment transportation. The ongoing glacier degradation in the Himalayan mountains in response to recent earthquakes and climate change increases the frequency of the debris flows and their sediment volume. The ZLL catchment in the tectonically active eastern Himalayan syntaxis with a high uplift rate has recorded five periglacial debris flow events since 1950. These events delivered huge volumes of sediment into the Yarlung Tsangpo River. We examine the history of the five events and their sediment characteristics, especially the ice-rock-avalanche-triggered event in 2020, through field investigations and remote sensing interpretations. Some findings are concluded as follows:

a) The periglacial debris flows have great capacities to erode channels, transport sediment, and impact obstacles. The maximum values of the erosion depth, the erosion width, and the impact force near the ZLL outlet are about 20 m, 14 m, and $3.64 \times 10^6$ kN, respectively, in the 2020 event. The debris flows transported a high concentration of coarse grains with the size > 50 cm. The 100-300 cm grains account for 77.4% of the coarse grains.

b) Most of the angular rocks moved by the 2020 avalanche were not delivered downward further. The boulders transported by subsequent debris flows probably originated from the middle of the downstream reaches. The grain size segregation was not observed between the middle reach and the alluvial channel.

c) The NVA of the ZLL fan reduced from 0.78 km$^2$ in 1950 to 0.067 km$^2$ in 1990, and kept at a stable low value until 2020, indicating the influence of the 1950 earthquake on the debris-flow sediment transportation could last 40 years. Compared with the 1999 Chi-chi earthquake and the 2008 Wenchuan earthquake in non-glaciated areas, the influence period of the 1950 earthquake is much longer.

d) The seismic and local meteorological data show that the recent warming events drove the 2020 debris-flow event during 2018-2020. The surging cycle of ZLL glaciers is getting short due to climate change. The correspondence between the recent increases in the local air temperature and the NVA implies that the debris flow occurrences in ZLL transfer from the tectonic-driven to the climatic-driven, with debris flows exhibiting a lagged response of 2-4 years to rising temperatures.

*Acknowledgments.* This research was funded by the Second Tibetan Plateau Scientific Expedition and Research Program (2019QZKK0902) and the National Natural Science Foundation of China (91747207, 41790434). MRG acknowledges the 'ANSO Scholarship for Young Talents' for his postgraduate study.

*Data availability.* All raw data can be provided by the corresponding authors upon request.

*Author contributions.* KHH conceptualized the study, interpreted the images, wrote and edited the manuscript. HL analyzed the data and wrote the manuscript draft. KHH, HL, SL, LW, XPZ, and BZ performed the field surveys. HL and MRG collected satellite and background data. LMZ provided constructive suggestions. All authors contributed to the preparation and editing of the paper.

*Competing interests.* The authors declare that they have no conflict of interests.

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
