# Peer review of "Variation of sediment supply by periglacial debris flows at Zelunglung"

_EGUsphere, 2024_

## Author Response (AR1)

**RC1:**

You describe an interesting 'longer term' link between earthquake occurrence and massive periglacial surface processes. I would mainly recommend a general (and professional) English review, and adding some lines about the physical relationship between the seismic and (peri) glacial processes. It is also important to highlight the relatively large distance (if I see well) between the 1950 Assam earthquake epicenter (or distance to activated fault) and the glacier valley.

- **Author's response:**

Thanks for your advice! The full text has been reviewed by native English speakers to enhance the manuscript's readability and professionalism.

As for the physical relationship between the seismic and (peri) glacial processes, we have already had a supplementary discussion in Section 6.1:

Strong ground vibrations caused by earthquakes can intensify cracking within the ice/rock mass, ultimately leading to the formation of substantial failure surfaces (Kilburn and Voight, 1998). Additional loading by earthquakes and coseismic-ice/rock avalanches could destruct the englacial conduit and subglacial drainage system. These changes can cause dynamic alterations to the glacier's thermal sensitivity, exacerbating its instability (Zhang et al., 2022). As critical solid material sources, these highly active ice/rock masses caused by seismic disturbance are prone to avalanches, calving, detachment and remobilization to form glacial debris flows (Deng et al., 2017; Zhang et al., 2022).

The epicenter of the 1950 Assam earthquake is approximately 195 kilometers away from the ZLL Valley. Notably, this seismic event not only triggered the glacier surge-debris flow chain but also incubated the 1953 debris flow in Guxiang Valley, approximately 50 kilometers northeast of the ZLL Valley. Although the Guxiang debris flow did not occur on the same day as the earthquake, the 1950 earthquake induced co-seismic avalanches, ice falls, and rockfalls of an unprecedented scale upstream of Guxiang Valley. These events contributed a substantial volume of loose material, which subsequently led to the 1953 debris flow.

- **Author's changes in manuscript:**

The manuscript has been reviewed by native English speakers to enhance the manuscript's readability and professionalism.

we have already had a supplementary discussion about the physical relationship between the seismic and (peri) glacial processes in Section 6.1.

we have emphasized the large-distance characteristics of the 1950 earthquake and the ZLL Valley in the first paragraph of section 6.1, and marked the location of the earthquake in Figure 1b.

**RC2:**

Hu and co-authors present an analysis of periglacial debris flows in a small catchment draining the Namche Barwa massif in the Himalaya using historical and modern remote sensing applications. I have some suggestions for improvements. In particular, I am not convinced by the discussion about the causes for the debris flows.

1. Motivation of the paper: First I suggest to sharpen the motivation of the paper. According to the introduction (L61 ff), "little is known about the roles the extreme hazards play in incrementing sediment erosion, transportation, and the control of the hazards between tectonic and climatic factors." I actually do not quite understand what is meant here.

● **Author's response:**

Recent observations indicate that episodic debris flows are predominately responsible for sediment transport from steeplands to rivers and channel erosion, being a major agent in landscape evolution in high mountain areas (Anderson et al., 2015; Kober et al., 2012; Lancaster and Casebeer, 2007; McCoy et al., 2013). Lin et al. (2006) presented that it could last for more than 10 years after the Chi-Chi earthquake. Cui et al. (2011) predicted that the effect of Wenchuan earthquake on post-quake debris flows would last for 10-20 years, while Huang (2011) predicted that it would be from 20 to 25 years. Recent publications demonstrate that only ~ 30% portion of coseismic sediment has been evacuated by debris flows and fluvial transport 12 years after the

Wenchuan earthquake (Dai et al., 2021), and the fine-grained landslide sediment mobilized by the earthquake may stay in the affected river catchments as long as a century (Wang et al., 2015). Until now, most of previous studies have focused on the residence time and transport of earthquake-triggered landslide sediment at an orogenic scale in no-glacierized environments (Dadson et al., 2004; Dai et al., 2021; Parker et al., 2011; Wang et al., 2015). Little attentions are paid on the sediment evacuation progress by post-seismic debris flows at a catchment in glacierized environments owing to relatively low likelihood of debris flows and absence of long-term site-specific data. Consequently, this paper is driven by the motivations:

- To describe how strong earthquakes and warming events escalate debris flows and associated sediment transport within small alpine watersheds.

- To clarify how long the effects of an earthquake on periglacial debris flows at a glaciated catchment.

- To examine the roles that climate and tectonics play in the development of extreme hazards in the tectonically active and climate-sensitive region of southeastern Tibet.

The reviewer's meticulous examination has highlighted an issue with our initial phrasing, which led to ambiguity. To clarify, we deleted the sentence and changed it with: " Little attentions are paid on the sediment evacuation progress by post-seismic debris flows at a catchment in glacierized environments owing to relatively low likelihood of debris flows and absence of long-term site-specific

data."

● **Author's changes in manuscript:**

We have revised this sentence.

2. **Second, it is not made clear in the following text how that knowledge gap is addressed by the study. Questions that would be good to answer in the introduction are: Why was that particular study area chosen? How do the approaches advance the gap that you are proposing? In the discussion or conclusion, you can also come back to that point.**

● **Author's response:**

As mentioned above, we want to study the sediment evacuation process dominated by debris flows in glacierized environments. In order to investigate the long-term effects of earthquakes on sediment evacuation in a glaciated catchment, we select the Zelunglung catchment, a tributary of the Yarlung Tsangpo river in southeastern Tibet that has large areas of temperate glaciers and disturbed intensely by the Ms 8.5 earthquake in 1950. Moreover, the catchment has long-term remote sensing imagery for interpretating glacier changes and associated debris flows and relatively well-documented records of at least four historical periglacial debris flows in 1950, 1968, 1972, and 1984 since the 1950 Assam earthquake.

● **Author's changes in manuscript:**

We added the explanation in the last paragraph of the Introduction.

3. **Methodology: I do not quite understand how you can take the non-vegetated area as a proxy for debris flow volume (e.g. L135). To get a volume from an area, I believe you need some estimate of thickness. How do you get that?**

- **Author's response:**

We appreciate the reviewer's valuable comment. As you pointed out, accurate estimation of debris flow volume is hindered by the lack of sediment thickness data. The sediment volume of debris flow may be influenced by various factors, such as fan area, average fan slope and average channel (e.g. Stoffel (2010)). Nevertheless, conducting a long-term time series analysis since 1950 presents challenges in acquiring adequate data for estimating sediment volume over the past century. Borehole method can give the accurate estimation of thickness, but is too expensive for our study. Consequently, this study employs an approximate alternative approach, wherein the fluctuation in debris flow volume is inferred from changes in the area of the accumulation fan. Actually, many previous studies consider debris flow volume is empirically a function of the accumulation area (e.g. Iverson et al. (1998)). It is crucial to underscore that our emphasis lies not on the absolute volume of debris flows but on their relative trends.

- **Author's changes in manuscript:**

We made no changes to this part of the manuscript.

4. **Causes of debris flows: First the discussion of causes for the debris flows is a bit confusing to me. In the abstract you write that the four events 1950 – 1984 were legacies of the 1950 Assam earthquake. The suggested link between the 1950 Earthquake and the 1950 debris flow seems solid (as per Fig. 15). The suggestion that the other events were also preconditioned by that earthquake seems less obvious to me. Is the only evidence that none of the other earthquakes could have caused the debris flows (L381ff)? What about other possible triggers, such as an extreme precipitation events? I am by no means an expert, but the argument for the influence of the 1950 earthquake on later debris flows doesn't seem very convincing to me. By the way, in the discussion (L381ff), it did not become clear that you actually suggest that the 1950 – 1984 events were all triggered by the earthquake. I only got that from the abstract.**

- **Author's response:**

The influence of a large earthquake on debris flows is a long lasting process, with variable durations of impact in different studies (Dai et al., 2021). For instance, following the Wenchuan earthquake, Tang et al. (2009) suggest a debris flow activity span of 5 to 10 years, whereas Cui et al. (2008) propose 20 to 30 years. Xie et al. (2009) indicate the possibility of strong debris flow activity persisting for 10 to 30 years, or more. Essentially, the above statement pertains

to the active duration of earthquake impacts on the loose source of debris flow channels. But, when the earthquake's effects is negligible, debris flow frequency-magnitude will resume to pre-quake level. Pre-1990 debris flow events as indicated by Figure 12d (in the Figures part of this document), represent a legacy of the 1950 earthquake. Of course, the post-seismic debris flows maybe directly triggered by extreme precipitation or temperature fluctuation. But the instability of the glacier/materials caused by the great earthquake is not negligible. Prior to 1990, NVA exhibited a consistent decline, whereas afterward, it displayed significant fluctuation and even a slight upward trend. This implies that the aftermath of the 1950 earthquake persisted until 1990, after which debris flow resumed activity on a relatively minor scale, influenced by climatic factors. Despite the occurrence of several small earthquakes preceding the events of 1968 and 1984 as noted by L381ff, these earthquakes were not captured by the Keefer curve considering magnitude and distance (figure 13 in manuscript), suggesting their minimal influence on the debris flow events.

- **Author's changes in manuscript:**

We supplement the above in Section 6.1.

5. Also, in the abstract you say that "recent warming events drove the 2020 event". And you start your discussion stating that either earthquakes or climate change increase the occurrence of debris flows. I do not think you have evidence to say that unless you have a much

longer time series. Lets assume the null-hypothesis that debris flow events occur randomly within some average recurrence interval. With the few events you study here, I would challenge you to show that the distribution of earthquakes is statistically distinguishable from a random occurrence. Similarly, I would challenge that a single debris flow after a single warming episode is evidence enough to conclude this flow is caused by the warming. I suggest to formulate these links much more carefully and "suggest" a link.

- **Author's response:**

We appreciate the valuable suggestion provided by the reviewer. The assumption of temporal random distribution of debris flows is probably right at an orogen scale. For a specific catchment, the debris flows are not random temporally, but controlled by earthquakes or climate events. That is confirmed by the 1999 Chi-chi earthquake and the 2008 Wenchuan earthquake. Whether earthquakes and climate change will definitely increase the occurrence of glacial debris flow is indeed to be further studied, but for the southeast Tibet region where ZLL is located, this seems to be based on evidence. For example, the Assam earthquake in 1950 caused frequent debris flow activities in Guxiang Valley for decades(Du and Zhang, 1981), and the increase of glacier ablation under the influence of climate change in the past 20 years has promoted the debris flow in Tianmo Valley (Deng et al., 2017) and Sanggu Valley (Wang et al., 2023). Therefore, in order to ensure objectivity, we change the relevant

statement to: "It is evident that either earthquakes or climate change may increase the occurrence of periglacial debris flows and their sediment yield in southeastern Tibet."

Figure 12 illustrates a notable decrease in rainfall since 2000, alongside an increase in temperature, with a particularly sharp rise in 2018. Remarkably, this pattern closely resembles the fluctuation of NVA. Although extreme rainfall may also induce debris flows, from the observation data of the Linzhi meteorological station (29.568° N, 94.467° E), the average maximum and minimum temperature from 15 August to 14 September in 2020 were 27 °C and 13 °C and the daily rainfall in this period ranged from 0 to 17.5 mm/d. According to an eyewitness video at 18:55 on 10 September 2020, the steel bridge deck was dry, which indicated that the precipitation was light on the event day (Peng et al., 2022). Therefore, combining the long-term climate trend and the daily value data before the event, we believe that the debris flow is caused by warming.

- **Author's changes in manuscript:**

we have added relevant references to the sentence:" It is evident that either earthquakes or climate change may increase the occurrence of periglacial debris flows and their sediment yield in southeastern Tibet".

We have added some sentences describing the weather in the month of the debris flow in the last paragraph of section 6.1.

**6. Line comments:**

**L105: Just visually. Fig. 2 doesn't particularly look like precipitation is increasing at all. How did you calculate that increase?**

- **Author's response:**

As depicted in Figure 2b, the precipitation at Linzhi Station exhibited significant fluctuations from 2000 to 2021. The linear regression analysis revealed a minimal growth rate of rainfall, specifically 0.065 mm/year, which was not readily discernible from the raw data. To enhance clarity, we have included fitted trend lines for both temperature and precipitation in figures 2a and b.

- **Author's changes in manuscript:**

We replaced the figure with a new one.

**L132: I would start the methods section with a summary sentence of the measurements that you are trying to make.**

- **Author's response:**

Thanks for your suggestion. We changed the structure of this section. We set the field surveys as a part of the methodology, and divided the original methodology section into two parts, they are NVA interpretation and Drone image interpretation. And we add a summary paragraph at the beginning of the sub methods section, as described below:

The study utilizes a combination of field surveys, aerial drone photography, and satellite imagery analysis to investigate debris flow events in the Zelunglung region. High-resolution orthoimages and digital surface models are generated

to assess terrain changes, while non-vegetated area (NVA) serves as a proxy for sediment volume for time series analysis. The integration of these methods offers a detailed insight into the debris flow history and its influencing factors.

● **Author's changes in manuscript:**

We changed the structure of this section. We set the field surveys as a part of the methodology, and divided the original methodology section into two parts, they are NVA interpretation and Drone image interpretation. And we add a summary paragraph at the beginning of the methodology.

**L159ff: I suggest to move the entire section on glacier changes before the methods. This is not your work as far as I understand, so it's a bit odd to have that as part of the other results. You could move it together with the study area section**

● **Author's response:**

Thanks for your suggestion, we have integrated this section into the study area section.

● **Author's changes in manuscript:**

We have integrated this section into the study area section, and meanwhile, we merged the figure 3 into figure 1c.

**L336: Can you explain where the interpretation of "some kind of dilute or hyper-concentrated flow" comes from?**

● Author's response:

Thank you for your review. We apologize for the oversight; this sentence was part of our original draft and was mistakenly left in, leading to a misunderstanding. We have now deleted it.

● **Author's changes in manuscript:**

We deleted the sentence.

**L351ff: A lot of that section reads like a discussion. I suggest to move it there.**

● **Author's response:**

Thanks for your suggestion, we moved the last 2 paragraph this section to the discussion section (new section 6.1), and named it with "The dominant factor for debris flows and sediment yield". As a consequence, the section 6.2 was renamed to "The future risk".

● **Author's changes in manuscript:**

We moved the last 2 paragraph this section to the discussion section (new section 6.1), and named it with "The dominant factor for debris flows and sediment yield". As a consequence, the section 6.2 was renamed to "The future risk".

**Another note on this paragraph: You write of three surges (L168) but then only explicitly note two of them (L169ff and L172ff). I guess the third surge**

is the one you describe in L179? Can you make that explicit?

- **Author's response:**

Thanks for your suggestion. Yes, the third surge occurred in 1984, and we wrote this date in the article.

- **Author's changes in manuscript:**

We have identified third surge in the article.

For all of the figures with geographic reference, it has to be clear where they are from with respect to the region. Figure 1 is missing a regional overview map. It was unclear to me where Figures 4 – 6 are taken within the study area or what extent Figs. 8-11 have. You can either mark their positions in Figure 1, or have little insets with every figure that show where in the study area that figure/picture is located. Also, not all figures have information on orientation and, if relevant, scale (e.g. Fig. 4-6. 7c&d, Photo of Fig. 9 missing north arrow etc.)

- **Author's response:**

Thanks for your suggestion. We added a regional overview map, and designated as Figure 1a. The interrelations among the sub-figures of Figure 1 are depicted using differently colored rectangular boxes. The camera's view direction of Figure 1d is marked with the orange arrow Figure 1c.

For extents of other independent images or view angle direction of photos in

the paper, we also marked them with rectangular boxes or arrows of different colors in Figure 1c. The details are as follows:

1. The scope of Figure 3a is indicated by a blue box in Figure 1c, whereas the extents of Figures 3b to 3f are delineated within Figure 3a itself.

2. The viewing angle in Figure 4 corresponds to the direction of the rose-red arrow in Figure 1c.

3. The viewing angle of the 3D satellite image in Figure 5a is aligned with the direction of the green arrow in Figure 1c.

4. The viewing angle directions for the photographs in Figures 6c and 6d are marked by black and red arrows, respectively, in Figure 1c.

5. We have standardized the display ranges of Figures 7, 8-a1, and 10 for uniformity, as indicated by a rose-red box in Figure 1c.

6. Given the constrained space in Figure 1c, the scope of Figure 9 is delineated using black boxes in Figure 7b.

7. The scope of Figure 11 is demarcated by a green box in Figure 1c

We have added north arrow in figures 1d, 3, 4, 5, 6c, 6d and 11. We have add scalebar in figures 3, 6a-b and 11. Add we also marked the view angle direction of photos 4, 5, 6c and 6d in figure 1c by rose-red, green, black and red coloured arrows, respectively.

- **Author's changes in manuscript:**

We changed the figures according to the comments, and made corresponding

modifications in the captions.

**Figure 1: The yellow text in panels b&c and the white text in panel a was hard to read on my printout.**

- **Author's response:**

Thanks for your suggestion. We have re-evaluated the text within the Figure 1 and implemented changes to the text color scheme. Additionally, we introduced a mask behind the text to enhance its contrast against the background, thereby improving legibility.

- **Author's changes in manuscript:**

We replaced the figure with a new one.

**Figure 4: The year numbers are a bit hard to see against the grey background**

- **Author's response:**

Thanks for your suggestion. We changed all the texts in to yellow to make it easier to recognize.

- **Author's changes in manuscript:**

We replaced the figure with a new one.

**Figure 6: There is no explanation in the figure or caption what T1-T2 are. Even if it is quite obvious, it would be good to define explicitly. The North**

**Arrow and scale-bar is really hard to see.**

- **Author's response:**

Thanks for your suggestion. Consequently, we have included explanations for T1 and T2 within the figure's caption. The North Arrow and scale bar in Figure a have been redesigned to enhance their visibility. Additionally, we have indicated the viewing angle direction of Figure b using a green arrow in Figure a.

- **Author's changes in manuscript:**

We replaced the figure with a new one, and included explanations for T1 and T2 within the figure's caption.

**Fig 8c: The person as scalebar is really hard to see**

- **Author's response:**

We have changed the photo in Figure c, which the person as scalebar in it is clearer. Red arrows have been employed to indicate the viewing angle direction of the photograph in Figure c, and the heights of the person have been annotated.

- **Author's changes in manuscript:**

We replaced the figure with a new one.

**Figure 12: Can you give some more information in the figure caption about where the 'on-site' sample is and when the sample was taken?**

- **Author's response:**

The 'on-site' sample was taken at the alluvial fan on September 11, 2020 (the day after the 2020 event). We have included this figure into figure 8, and numbered it as figure 8d. The location of the on-site sample was marked in figure 8a-1.

- **Author's changes in manuscript:**

We added the information in section 3.2.1 "Field surveys".

**RC3:**

1. The manuscript "Variation of sediment supply by periglacial debris flows at Zelunglung in the easter syntaxis of Himalayas since 1950 Assam Earthquake" by Hu et al., describes the occurrence of five debris flows that impacted the Zelunglung alluvial fan using a combination of field surveys, historical aerial imagery, and UAV flights. The manuscript contributes with observations on long-term changes in vegetation at the alluvial fan interpreted as a proxy for debris flow activity at the catchment. The manuscript aims to estimate relative qualitative debris flow activity compared to the 1950 event. Also, a detailed description of the 2020 debris flow event is presented using UAV photogrammetry. The authors discuss the results considering seismic activity, the Zelunglung glacier surge dynamics, and precipitation and air temperature. The manuscript exemplifies the complex interactions between tectonic and climate in debris flows triggering factors in glaciated areas. Despite the presented remote sensing interpretation, field observation, and literature review, further considerations are needed to strengthen the conclusions, particularly the conclusion referring to the influence of the 1950 earthquake in further debris flows until 1990.

- **Author's response:**

Thank you for your appreciation! As you mentioned, the debris flow activity in

glaciated areas is more complex than in non-glaciated areas. Our case study attempt to shed light on how tectonic and climatic factors influence sediment evacuation processes via debris flows. Lin et al. (2006) presented that it could last for more than 10 years after the Chi-Chi earthquake. Cui et al. (2011) predicted that the effect of Wenchuan earthquake on post-quake debris flows would last for 10-20 years, while Huang (2011) predicted that it would be from 20 to 25 years. Recent publications demonstrate that only ~ 30% portion of coseismic sediment has been evacuated by debris flows and fluvial transport 12 years after the Wenchuan earthquake (Dai et al., 2021), and the fine-grained landslide sediment mobilized by the earthquake may stay in the affected river catchments as long as a century (Wang et al., 2015). Compared with the 1999 Chi-chi earthquake and the 2008 Wenchuan earthquake, the influence period of the 1950 earthquake is longer. We think the earthquake effects last longer in glaciated areas.

- **Author's changes in manuscript:**

We added a sentence in the third point of the conclusion "".

2. **The presented historical aerial imagery is a valuable source of information that could be explored to strengthen the manuscript. For example, if available, the source and affected area of the historical periglacial debris flows (1950,1968,1972,1984) could be identified.**

- **Author's response:**

We show our gratitude to the reviewers for their valuable feedback. Regrettably, we lack the necessary resources to support this research. Obtaining high-resolution images of both the source and affected areas, especially from immediately before and after the events, presents significant challenges due to the lack of advanced satellite remote sensing technology during that time period. Apart from the 1972 event, the occurrence of the other three events is well-documented in previous literature (as we quoted in my manuscript).

- **Author's changes in manuscript:**

We made no changes to this part of the manuscript.

3. **I missed in the manuscript a clear explanation about why you consider that the 1950 earthquake has a stronger influence than the glacier surges in the triggering of the debris flows after 1950 and, therefore, impacting the vegetation changes between 1950 and 1990 (line 484). In lines 187-188, it is stated that the debris flows were triggered by the glacier instability. Also, the 1950 debris flow coincides with a glacier surge.**

- **Author's response:**

As Zhang and Shen (2011) have stated: "The Chayu violent earthquake (Ms 8.5, on Aug 15, 1950) evoked Zelongnong glacier surge (Zhang 1985). After the rapid motion the ice cube carrying a mass of solid matter moved to the lower reaches. Ice block thawed and evolved into debris-flows during this process, a

great deal of ice block and grit deposited at the convergence mouth." Therefore, the events triggered by the 1950 earthquake constituted a chain reaction of glacier surge and debris flow, wherein the glacial process was integral to the debris flow mechanism. The impact of such a seismic event does not dissipate suddenly. Pre-1990 debris flow events, in our view, represent a legacy of the 1950 earthquake. While their occurrence may have been directly triggered by glacial advancement or extreme precipitation, the root cause is the continued understability of the glacier/materials caused by the great earthquake. Prior to 1990, NVA exhibited a consistent decline, whereas afterward, it displayed significant fluctuation and even a slight upward trend. This implies that the aftermath of the 1950 earthquake persisted until 1990, after which debris flow resumed activity on a relatively minor scale, influenced by climatic factors.

- **Author's changes in manuscript:**

We explained more details in section 6.1.

We have modified the expression of lines 187-188: Four large-magnitude debris flows accompanied by glacier instability occurred in 1950, 1968, 1973, and 1984.

4. **The photo interpretation from Planet Lab images shows an ice-rock residual under the detachment area of the 2020 debris flow. The authors conclude that the entrained volume is at least 16 times the initiated volumes (line 227), consistent with the 1.14 Mm$^3$ previously presented by Peng et al., (2022). This is highly relevant for the**

**calibration of Debris Flow models.**

- **Author's response:**

Thank you for your suggestion! Yes, this data can be used to calibrate ice-rock avalanche and debris flow models.

- **Author's changes in manuscript:**

No Change.

5. **The proposed methodology involved a considerable amount of remote sensing data manual interpretation and general information on the uncertainty of the manual mapping of non-vegetation areas was given. Regardless, I missed a short sentence on the reconstruction of the DSM from the UAV surveys (e.g., software use, ground control points, alignment), and the uncertainty on the elevation change that propagates to the presented volumes (line 346).**

- **Author's response:**

The reconstruction and differencing of DSMs are carried out in Pix4DMapper and Arcmap10.8. Since we did not deploy ground control points during drone photography, we generated DSM and DOM of September 9 in Pix4DMapper, and then selected 20 relatively stable points that were not affected by debris flow events as control points in Arcmap with DOM of September 9 as reference. These control points were then used in Pix4DMapper to generate the September 11 DSM and DOM.

To determine the uncertainty for our UAV DSMs of difference (DoD) differencing result we follow methods outlined in Shugar et al. (2021). We identified a series of fifteen stable areas on old debris flow terraces adjacent to the valley floor (Mainly roads and unseeded farmlands) and retrieved the standard deviation of DoD values within these areas and used these to estimate a two-sigma DoD uncertainty. The uncertainty was ±0.493 m.

- **Author's changes in manuscript:**

We have included a description of this in section 3.2.3 "Drone image interpretation".

The manuscript is well structured and the figures are illustrative.

**6. Line comments:**

**Line 130-131: Did you present results on the surveys in 2021 and 2022 in this manuscript?**

- **Author's response:**

We conducted three field surveys, as described in lines 68-69 of the manuscript. The initial two were scheduled one day before and one day after the 2020 event, on September 9 and 11, 2020, respectively. Predominantly, these surveys involved drone photography before the disaster and included drone photography, measurements, and sampling in the post-disaster assessment. The third survey took place on December 21, 2022, and was specifically designed to capture aerial photography of the ZLL basin's upper reaches, an area

inaccessible in the prior surveys due to adverse weather conditions. The outcomes of the 2022 survey are detailed in Figures 1d, 4, and 6c, providing insights into the upstream channel and the debris flow initiation zone. We did not execute a field survey in 2021. We offer our apologies for any confusion that our initial errors may have caused.

- **Author's changes in manuscript:**

We retraced field survey timeline in section 3.2.1.

**Figure 4: Can you add a north arrow and a scale? Maybe you can try to use a different font color. The years of the images are hard to see.**

- **Author's response:**

Thanks for the reviewer's suggestion. We have added north arrows and scalebars in all sub-figures, and changed all the texts in to yellow to make it easier to recognize.

- **Author's changes in manuscript:**

We replaced the figure with a new one.

**Figure 9: c) Which distance is presented on the x-axis? distance from the outlet?**

- **Author's response:**

The x-axis in figure c represents the distance from P1 to P2 (i.e., the farthest end of the alluvial fan) along the main channel in Figure a-1. NOTE: To distinguish

them from T1 and T2 in Figure 5, we have modified the original labels in Figure a-1 to P1 and P2, respectively.

- **Author's changes in manuscript:**

We changed the "T1" and "T2" into "P1" and "P2" in both figure and figure caption.

**Figure 11: Could you please extend the cross-section before the bridge to include the deposition areas? You could also include some information on the deposited particle sizes to exemplify.**

- **Author's response:**

Following your feedback, we have the cross-section before the bridge to include the deposition areas. As shown in Figure 10c, the left bank edge of the channel has significant deposition, while the right bank platform actually has some fine particles or slurry deposited, but the graphic result is weak erosion, which may be due to the bias between the two phases of DSM. Due to the limitation of DOM resolution and the fact that debris flow slurry and particles with small size are mainly deposited in this area, particles with particles smaller than 50cm are difficult to be separated from DOM, and only a few coarse particles with size > 50cm are detected on the left bank edge, which are not enough to support our further analysis.

- **Author's changes in manuscript:**

We replaced the figure with a new one.

**Line 341. The maximum erosive depth of 20.47m is in the main channel or correspond to lateral erosion? Please include the mean erosive depth at the channel.**

- **Author's response:**

The maximum erosive depth of 20.47m is in the main channel. The mean erosive depth at the channel is 4.17m (The calculation area is the upstream area of the Zhibai bridge as shown in Figure 10).

- **Author's changes in manuscript:**

We add the description of mean erosive depth in manuscript.

**Line 346. Could you please discuss how much underestimated is the volume you are presenting compared to Peng et al., (2022) and what is the expected error from the photogrammetric workflow?**

- **Author's response:**

Our estimated final deposit volume is $12.8 \times 10^4$ m$^3$, with an uncertainty of $\pm 0.493$ m according to DoD, and the uncertainty of deposit volume is $\pm 1.85 \times 10^4$ m$^3$. Peng et al., (2022) estimated the final deposit volume is $37.5 \times 104$ m$^3$, and our result is 65.8% smaller than Peng et al., (2022). It is noteworthy that our DSMs were derived from data collected in September, during a period of high water levels in the Yarlung Tsangpo River. This situation could result in the neglect of certain sediment, potentially submerged by the water. In contrast, Peng et al.,

(2022) utilized data from December, a time characterized by low water levels in the Yarlung Tsangpo River. During this period, previously submerged sediments became exposed, and the area of the deposit, as delineated by Chen et al., was nearly twice the size of ours. This discrepancy accounts for the primary reason behind the underestimation of volume in our study. Furthermore, Peng et al. utilized data from the Ziyuan-3 satellite, which has a coarse resolution (2.5m). This lower resolution may also impact volume estimations.

● **Author's changes in manuscript:**

We have included the above discussions in the end the 2nd paragraph of section 4.2.3.

**Figure 12: The figure is not referenced in the text. Maybe you can merge Figure 12 and Figure 9 and include the location of the on-site sample.**

● **Author's response:**

Thanks for the reviewer's suggestion. We have included the graph of Cumulative grain size distribution into figure 8, and numbered it as figure d. The location of the on-site sample was marked in figure 8a-1. We referenced the figure 8d in the 2nd paragraph of section 4.2.3.

● **Author's changes in manuscript:**

We deleted figure 12, and replaced the figure8 with a new one.

**Figure 13: Can you add a north arrow?**

- **Author's response:**

Thanks for the reviewer's suggestion. We added the scale-bars and the north arrows in the Figure.

- **Author's changes in manuscript:**

We replaced the figure with a new one.

**Figure 14: Could you also include in Figure d) the other 4 debris flows for comparison?**

- **Author's response:**

Thanks for the reviewer's suggestion. We have added the other 4 debris flows in Figure d for comparison.

- **Author's changes in manuscript:**

We replaced the figure with a new one.

**Figures:**

We have updated the figures in the manuscript, and the revised figures are as follow:

[revised manuscript text omitted]

---

## Referee Report (RR1)

Review of *Variation of sediment supply by periglacial debris flows at Zelunglung in the eastern syntaxis of Himalayas since the 1950 Assam Earthquake* submitted to Earth Surface Dynamics

This is my second review of this paper. I visited the changes to the manuscript found that the authors have addressed most of the comments posed by the reviewers well. However, I am still skeptical of how their data can be interpreted in terms of the effects of earthquakes and climate. The reason is not that I don't believe that climate and earthquakes affect debris flows. Rather, there are three main points, I am not sure I cannot quite get on board with:

(i) The authors suggest (I think) that the 1950 Earthquake led to elevated sediment transport for around 40 years based, I think, on the rate of decline of NVA values between 1950 and 1990. I would need a bit more detail on how the NVA forms to follow the point. Debris flows will directly increase NVA. There are three debris flows in the 40 years after the 1950 event, but their effects on the NVA seem small (or at least they do not show up clearly on Fig 12d). Rather, in the 40 years post 1950, the NVA slowly declines. Isn't the 40 year of declining NVA just representative of the timescale that it takes to re-vegetate the surface? In other words, couldn't it be that the decline of NVA has to do with the timing of vegetation growth rather than the impact of the 1950 earthquake on increased sediment transport? What is happening to the fan surface in-between the major debris flows that you report?

(ii) I am still not sure about the proposed link between the 2020 event and the role of climate. You write that "The correspondence between the recent increases in the local air temperature and the NVA implies that the debris flow occurrences transfer from the tectonic-driven to the climatic-driven", and "based on the fact that the trend of the 1990-2020 NVAs shows a good agreement with that of the air temperature in the same period, it is likely that the 2020 event was driven by the recent local warming rather than by geological events". I have some questions/challenges with that rational

- First, I do not see the correspondence between NVA and temperature. Sure, there is an increase in the unvegetated area after the 2020 debris flow – which is just a product of a debris flow happening. The NVA does not seem to increase just with warming (which starts in 2018).

[Figure]

- Second, given that NVA is driven by debris flows, isn't the argument (a link between climate and NVA changes are indicative of the role of climate on debris flows) a bit circular or redundant? It feels the same as saying: The fact that the 2020 debris flow occurred within a warming trend means that it was triggered by warming. In particular, the occurrence of one debris flow event within a warming period does not seem to me enough evidence to say that warming triggered that debris flow.
- Third, you seem to discount Earthquakes as a trigger for the 2020 event even though you have the highest earthquake frequency on your record right around the 2020 event. As far as I understand, the reason you discount the earthquake trigger is (also?) because the earthquakes do not fall below the Keefer curve (similar to the 1968 and the 1984 events). How robust is this characteristic? What triggered the 1968 and 1984 events then if it wasn't earthquakes or warming? Isn't it possible that the 2020 event is just similar to these two events and was not directly triggered by the warming?

(iii) There are also suggestions in the manuscript in places that the climate change increases the frequency of debris flow events. (e.g. L439: "Undoubtedly, the on-going warming increases the frequency of such glacier-related slope failures".). I am not convinced the data in this work speak to the presence or absence of such a link, and the link between climate change and glacier-related natural hazards can be complex. For example, the frequency of GLOFs does not necessarily just simply increase with climate warming (Veh et al., 2019; Veh et al., 2023).

**Line comments**

Several of the new sections would benefit from review of the language (e.g.: "Little attentions are paid on" or "imagery for interpretating glacier changes")

L47: Maybe "triggered by" rather than "driven by"?

L71: "It is believed that historical earthquakes and ongoing climate warming drove these events" needs a citation or justification. Who believes that?

L129: As I said before, I do not see an increase in precipitation rates here. The increase is so small it must surely be within the uncertainty of the scatter of the data. If you plotted the confidence bands of the regression or added the standard error of the slope of the line, I would bet it is way within uncertainty of 0 or a decreasing trend. Also, in the discussion, you say there is no significant trend in precipitation.

L414: What do you mean by "the Keefer curve did not detect any of these seismic events"? A curve cannot detect anything.

L440: "show a similar growing trend". Similar to what?

L448: "the trend of the 1990-2020 NVAs shows a good agreement with that of the air temperature in the same period".

L459: Unclear which of the debris flows "the debris flow" corresponds to (presumably the 2020 one, but it's unclear from this paragraph).

I hope these comments are useful and remain with best wishes

Aaron Bufe

Veh, G., Korup, O., von Specht, S., Roessner, S., and Walz, A., 2019, Unchanged frequency of moraine-dammed glacial lake outburst floods in the Himalaya: Nature Climate Change, v. 9, no. 5, p. 379-383.

Veh, G., Lützow, N., Tamm, J., Luna, L. V., Hugonnet, R., Vogel, K., Geertsema, M., Clague, J. J., and Korup, O., 2023, Less extreme and earlier outbursts of ice-dammed lakes since 1900: Nature.

---

## Author Response (AR2)

**Author's response to reviewers**

Thank you for the reviewer's various lines of constructive criticism which were very helpful in improving analysis and the manuscript. Our revisions will be made on the revised manuscript submitted last time, and point-by-point reply for each reviewer can be found in the following:

**Reviewer #1**

This is my second review of this paper. I visited the changes to the manuscript found that the authors have addressed most of the comments posed by the reviewers well. However, I am still skeptical of how their data can be interpreted in terms of the effects of earthquakes and climate. The reason is not that I don't believe that climate and earthquakes affect debris flows. Rather, there are three main points, I am not sure I cannot quite get on board with:

Thank you for your appreciation! And we are very grateful for various lines of constructive criticism which were very helpful in improving analysis and the manuscript. Point-by-point replies can be found in the following:

(i) The authors suggest (I think) that the 1950 Earthquake led to elevated sediment transport for around 40 years based, I think, on the rate of decline of NVA values between 1950 and 1990. I would need a bit more detail on how the NVA forms to follow the point. Debris flows will directly increase NVA. There are three debris flows in the 40 years after the 1950 event, but their effects on the NVA seem small (or at least they do not show up clearly on Fig 12d). Rather, in the 40 years post 1950, the NVA slowly declines. Isn't the 40 year of declining NVA just representative of the timescale that it takes to re-vegetate the surface? In other words, couldn't it be that the decline of NVA has to do with the timing of vegetation growth rather than the impact of the 1950 earthquake on increased sediment transport? What is happening to the fan surface in-between the major debris flows that you report?

● Author's response:

We appreciate the reviewer's insightful comments and questions regarding the interpretation of NVA and its relationship with sediment transport following the 1950 earthquake. Here, we

provide a detailed response to address the concerns raised:

We conducted a new orthophoto acquisition of the ZLL watershed on November 17, 2024, using the DJI Mavic 3E aerial drone. Based on the control points selected from the drone orthophotos taken on September 10, 2020, we generated an unbiased orthophoto for 2024 in Pix4D. We selected two bare land areas at the watershed outlets caused by the 2020 event, both located at higher terraces and unaffected by subsequent debris flows, with a total area of 2375.708 m². Since the orthophotos were taken in mid-November, some vegetation stems and leaves had withered, but thanks to the high resolution of the images (~0.04m*0.04m), we visually interpreted the newly grown vegetation area (VA) within these two regions in 2024 on the Qgis platform, totaling 506.112 m² (Fig S1 a) (the visual interpretation error is described in section 3.2.2 of the manuscript). During the four growing periods from September 2020 to November 2021, the proportion of VA per unit area in these two regions increased by 0.053 annually. However, with the NVA (Non-Vegetated Area) of 780,000 m² in 1950 as a reference, the proportion of NVA per unit area decreased at a rate of 0.023 per year from 1950 to 1990 (Fig S1 b). This indicates that the 40-year decline in NVA does not represent the timescale required for surface vegetation to re-grow, as the rate of vegetation re-growth is much higher than the rate of NVA decline caused by debris flows.

[Figure]

**Figure S1: (a) Interpretation of newly grown vegetation on two bare land areas caused by the 2020 event; (b) Changes in the proportion of NVA per unit area and changes in the proportion of newly grown vegetation area (VA) per unit area.**

The three debris flow events following the 1950 earthquake are well-documented in the literature (Zhang and Shen, 2011; Zhang, 1985, 1992). Figure S2-a2 shows significant deposition in the middle of the fan in 1969, characterized by a rough fan surface and indistinct channels, indicating a large-scale event prior to December 1969 (likely in 1968). By February 1972, these deposits were noticeably flattened (Fig. S2-b2). Images from March 1973 show a marked collapse of the terminal glacier compared to February 1972 (Fig. S2-b3 and 2-c3), suggesting that the event referred to as the 1973 event by Peng et al (2022) likely occurred in the summer of 1972. The depositional fan in December 1975 exhibits significant brightness variations (Fig. S2-d2), with pronounced channelization above the glacier (Fig S2-d1), indicating possible debris flow activity

prior to this time. Compared with 1975, the depositional fan in 1979 displays a flatter terrain and more distinct channelization (Fig. S2-e2), indicating the modification of the rough fan surface by debris flow activity. This also implies that, due to limited information at the time, additional events during this period may have gone unrecorded. By 1982, noticeable vegetation had recovered in the middle part of the depositional fan. Additionally, glacier ablation intensified, exposing lateral moraines (Fig. S2-e3 and S2-f3), and the lowest section of the glacier developed numerous crevasses (Fig. S2-f4 and S2-f5). These fractured ice bodies and moraine materials, under the impact of the 1984 ice avalanche at 3700 m described by Zhang (1992), contributed to the formation of the large-scale debris flow.

However, from 1950s to 1980s, available remote sensing data are scarce, especially in remote areas such as the ZLL watershed. High-resolution remote sensing images immediately before or after the major events were not accessible. As a result, the NVA caused by these three events cannot be specifically delineated. Nevertheless, as previously mentioned, due to highly active glaciers and moraine materials, more than three debris flows occurred between 1950 and 1990, beyond those documented in the literature, as evidenced by the NVA interpreted from high-quality images of other years (Fig. 12d in manuscript). Due to the depletion of available material sources and the gradual decline in glacial activity over time caused by the earthquake, the scale of these debris flows progressively decreased, and the NVA of the 1968, 1972, and 1984 events lies within the overall declining trend of NVA (Fig. 12d in manuscript).

We hope this clarification addresses the reviewer's concerns and enhances the understanding of our findings regarding the long-term impact of the 1950 earthquake on sediment transport processes.

[Figure]

**Figure S2: Variations of the Zelunglung alluvial fan and channel during 1969 – 1982. The images were taken from Keyhole reconnaissance satellites (https://earthexplorer.usgs.gov/).**

- Author's changes in manuscript:

We have replaced Figure 3 with Figure S2 and added more descriptions regarding the changes in the fan surface and glacier in Section 4.1 as follows:

The fan in December 1975 exhibits significant brightness variations (Fig. 3-d2), with pronounced channelization above the glacier (Fig. 3-d1), suggesting possible debris flow activity prior to this time. Compared with 1975, the fan in 1979 displays a flatter terrain and more distinct channelization (Fig. 3-e2), indicating the modification of the rough fan surface by debris flow activity. This also implies that, due to limited information at the time, additional events during this period may have gone unrecorded. By 1982, noticeable vegetation had recovered in the middle part of the fan (Fig. 3-f2). Concurrently, accelerated glacier ablation exposed lateral moraines (Fig. 3-e3 and 3-f3), while the glacier terminus developed an extensive crevasse network (Fig. 3-f4 and 3-f5). These fractured ice bodies and moraine materials, under the impact of ice avalanche at 3700 m described by Zhang (1992), contributed to the formation of the 1984 large-scale debris flow.

(ii) I am still not sure about the proposed link between the 2020 event and the role of climate. You write that "The correspondence between the recent increases in the local air temperature and the NVA implies that the debris flow occurrences transfer from the tectonic-driven to the climatic-driven", and "based on the fact that the trend of the 1990-2020 NVAs shows a good agreement with that of the air temperature in the same period, it is likely that the 2020 event was driven by the recent local warming rather than by geological events". I have some questions/challenges with that rational

- First, I do not see the correspondence between NVA and temperature. Sure, there is an increase in the unvegetated area after the 2020 debris flow – which is just a product of a debris flow happening. The NVA does not seem to increase just with warming (which starts in 2018).

- Author's response:

Thank you very much! Yes, we can not conclude that there is correspondence between NVA and temperature. An increase in the unvegetated area after the 2020 debris flow is a direct result of a debris flow happening. From Fig.S3, it is observed that the local air temperature and the NVA exhibit similar trend between 1990 and 2020 despite the sudden increase in NVA is 2-4 years later

than that in the local temperature.

Studies have shown a correlation between climate warming (particularly summer warming) and slope failures. For instance, in the Saint Elias Mountains of Alaska, non-seismic rock avalanches occurring between 1964 and 2019 were associated with above-average temperatures. The frequency of such events is expected to continue increasing with ongoing climate warming (Bessette-Kirton and Coe, 2020). In the Mont Blanc region of the Swiss Alps, the warmest periods since 1860 strongly correlate with 58 recorded rockfall events (Deline et al., 2015). However, the correlation is not a strict one-to-one relationship (i.e., a higher air temperature in a specific year does not necessarily correspond to a higher rate of slope instability in the same year). Some findings explicitly indicating a delay or lag in the response, rather than being an immediate reaction. Long-term rockfall records from Täschgufer in the Swiss Alps (1920–2020) demonstrate that sustained warming facilitates rockfall release, with activity changes significantly correlated with temperature variations on interannual and decadal timescales (Stoffel et al., 2024). Notably, rockfall activity during 1949–1953 increased following the sustained warm summers of 1943–1947, and the 1994–1995 peak in activity similarly followed the warm summers of 1990–1993 (Stoffel et al., 2024). Comparable correlations between increased rockfall and warming have been documented in other locations within the region (Stoffel et al., 2024). Evidence from a case study in Tianmo Valley, approximately 60 km from the ZLL catchment, also indicates that prolonged and sustained warming has a significant impact on the triggering of peri-glacial debris flows (Deng et al., 2017). Furthermore, glacial lake outburst flood events in the Himalayan region exhibit delayed responses to warming (Zou et al., 2024), indirectly reflecting the lagged responses of ice/rock avalanches (as triggers for such events, including debris flows) to climatic changes. It is also worth mentioning that prolonged increases in air temperature drive changes in moraines, supplying abundant active debris that is highly susceptible to triggering by high-altitude rockfalls, ice-rock avalanches, and other processes—mechanisms critical to large-scale debris flow formation (Deng et al., 2017).

[Figure]

**Figure S3 (a) Seismic events within a 200 km distance to the Zelunglung from 1940 to the present. (b) Changes in the annual and summer precipitation in the Zelunglung from 1940 to the present. (c) Changes in the annual mean and summer air temperatures in the Zelunglung from 1940 to the present. (d) Changes in the non-vegetated area of the Zelunglung alluvial fan from 1969 to the present (although the deposition of the 1950 event did not happen at the Zelunglung's outlet like the later events, we plot the NVA of the 1950 event as the starting point). (e) Changes in the annual and summer precipitation in the Zelunglung from 1990 to 2017. (f) Changes in the non-vegetated area of the Zelunglung alluvial fan from  1990 to 2017.**

- Author's changes in manuscript:

We have replaced Figure 12 with Figure S3, and added explanations in L455 and L464:

**L455:** Figure 12f highlights four distinct NVA peaks, which likely correspond to small mountain torrents or debris-flows, as suggested by Zhang and Shen (2011). These NVA peaks exhibit a lag of 2–4 years relative to annual mean or summer air temperature peaks (Figure 12e and 12f). Similarly, the sharp increase in NVA caused by the 2020 debris flow event occurred two years after the 2018 warming anomaly (Fig. 12b and 12d). This lag phenomenon has also been observed in other comparable regions (Stoffel et al, 2024)

**L464:** However, based on the fact that the lag relationship between the fluctuation peaks of NVAs and temperature fluctuations from 1990 to 2020, it is likely that the 2020 event was triggered by the recent local warming.

- Second, given that NVA is driven by debris flows, isn't the argument (a link between climate and NVA changes are indicative of the role of climate on debris flows) a bit circular or redundant? It feels the same as saying: The fact that the 2020 debris flow occurred within a warming trend means that it was triggered by warming. In particular, the occurrence of one debris flow event within a warming period does not seem to me enough evidence to say that warming triggered that debris flow.

- Author's response:

Thank you very much! You pointed out there is a bit circular in our expression. Our expression is not exact, which leads to misunderstanding. As explained above, it is not a corresponding relationship but ta similar trend between the local air temperature and the NVA between 1990 and 2020. Historical records and high-resolution remote sensing imagery only capture five documented debris flow events in this catchment. Therefore, it is necessary to identify an alternative indicator, such as NVA, to infer debris flow occurrences and their magnitude. For this reason, we argue that the relationship between climate and NVA changes serves as a proxy for understanding the relationship between climate and debris flows, representing a unidirectional rather than cyclical process, i.e., climate → debris flows → NVA changes.

As shown in Figure S3 (e) and (f), both temperature and NVA exhibited a similar trend from 1990 until the warming anomaly in 2018. Notably, Figure S3(f) highlights four distinct NVA peaks,

which likely represent small-scale flash flood or debris flow events (Zhang and Shen, 2011). These NVA peaks show a lag of 2–4 years relative to annual mean or summer air temperature peaks. Similarly, the sharp increase in NVA caused by the 2020 debris flow event occurred two years after the warming anomaly in 2018 (Fig. S3-e and d). This lag phenomenon has also been observed in other comparable regions (Stoffel et al., 2024). Therefore, we conclude that the 2020 event, along with the small mountain torrents or debris-flows between 1990 and 2017, was driven by climate warming.

Regarding why earthquake triggers are excluded as a factor, this will be addressed in detail in our response to the third comment.

- Author's changes in manuscript:

Consistent with the previous response, we have replaced Figure 12 with Figure S3, and added explanations in L455 and L464:

L455: Figure 12f highlights four distinct NVA peaks, which likely represent small mountain torrents or debris-flows, as Zhang and Shen (2011) mentioned. These NVA peaks show a lag of 3– 5 years relative to annual mean or summer air temperatures (Figure 12e and 12f). Similarly, the sharp increase in NVA caused by the 2020 debris flow event occurred two years after the warming anomaly in 2018 (Figure 12b and 12d). This lag phenomenon has also been observed in other comparable regions (Stoffel et al, 2024)

L464: However, based on the fact that the lag relationship between the fluctuation peaks of NVAs and temperature fluctuations from 1990 to 2020, it is likely that the 2020 event was triggered by the recent local warming.

- Third, you seem to discount Earthquakes as a trigger for the 2020 event even though you have the highest earthquake frequency on your record right around the 2020 event. As far as I understand, the reason you discount the earthquake trigger is (also?) because the earthquakes do not fall below the Keefer curve (similar to the 1968 and the 1984 events). How robust is this characteristic? What triggered the 1968 and 1984 events then if it wasn't earthquakes or warming? Isn't it possible that the 2020 event is just similar to these two events and was not directly triggered by the warming?

● Author's response:

Large earthquakes (≥ Mw 6.0) can significantly influence mass-wasting processes (Jones et al., 2021). Our previous studies on the 2017 Milin earthquake (Mw 6.4) revealed that the maximum intensity (VIII degree) region covered approximately 310 km², with an average impact radius of about 10 km. Co-seismic landslides were almost entirely confined to this region, and no slope failures were observed in the ZLL catchment (Hu et al., 2019). As highlighted in the manuscript (L470), if the 2017 Milin earthquake had strongly impacted the glaciers in the ZLL, ice-rock failures would likely have occurred within a few months, similar to what was observed in the Sedongpu catchment. However, no such events were documented in the ZLL. While the highest frequency of earthquakes occurred near the time of the 2020 event, the maximum magnitude during this period was only Mw 5.2, with an average magnitude of Mw 4.6 (Fig. S3 a and Fig. S4). Furthermore, the epicenters were all located more than 30 km from the ZLL catchment (Fig. S4), well beyond the impact radius of the 2017 Mw 6.4 Milin earthquake. We applied the criteria established by Keefer (1984) to identify earthquakes within the magnitude and distance thresholds, a methodology also employed in studies of rock avalanches in Alaska (Coe et al., 2018). None of these seismic events during this period fell within the range of influence as defined by the Keefer curve (Fig. S4). These analyses collectively lead us to conclude that earthquakes were not the trigger for the 2020 event.

[Figure]

**Figure S4: Distance from epicenters of the collected seismic events to the Zelunglung vs. the seismic magnitude (the black solid curve refers to Keefer (1984)).**

As stated in the previous answer, the events of 1950, 1968, and 1984 are well-documented in the literature (Zhang and Shen, 2011; Zhang, 1985, 1992). The 1950 event coincided with the Assam earthquake and is considered directly related to this seismic event. This disaster occurred immediately after the earthquake, destabilizing the terminus of the ZLL Glacier at an elevation of 3,650 m and forming a glacier dam in the Yarlung Tsangpo River. During the 1968 event, the ZLL Glacier again formed a dam over 50 m high in the Yarlung Tsangpo River. By 1982, the fractured glacier had been divided into six segments due to differential melting. The First Qinghai-Tibet Scientific Expedition observed in 1984 that a localized rupture occurred at 3,700 m on the ZLL Glacier. The broken ice, carrying loose moraine debris, traveled horizontally up to 150 m. By 1989, three of the glacier segments had melted, detached from the main glacier, and were buried under thick moraine deposits. In fact, the 1972 event was also caused by the collapse of the terminal glacier (Fig. S2-b3 and 2-c3). These events clearly indicate that the frequent ruptures, collapses, and differential melting of the ZLL Glacier were associated with disturbances caused by the 1950 earthquake. While rising temperatures likely contributed to the melting and instability of the glacier, temperature alone could not have caused the direct collapse of the main trunk glacier and the subsequent debris flows. The root cause was the structural damage and exposure of the glacier to lower-altitude warming following the earthquake. Since 1990, the only large-scale event in this region has been the 2020 event, which was triggered by an ice-rock avalanche on the southern ridge of the catchment, unrelated to the trunk glacier. The frequency of large-scale disaster events in the 40 years after the 1950 earthquake was significantly higher than that after 1990, suggesting that glacier instability during this period can be regarded as a preconditioning effect of the earthquake. Therefore, we consider the 1968 and 1984 events to be related to the 1950 earthquake.

●   Author's changes in manuscript:

We have replaced Figure 13 with Figure S4. The narrative logic of the first paragraph in Section 6.1 has been reorganized, and additional explanations have been incorporated as follows:

L423: While the highest frequency of earthquakes occurred near the time of the 2020 event, they could be ignored due to their small magnitude (Mw≤5.2) and long distance (>30km) (Fig 13). This is because even the 2017 Mw 6.4 Milin earthquake, with an epicenter 24 km from the

ZLL, had a very limited impact area (310 km², ~10 km impact radius) (Hu et al., 2019), and there were no report or sign of such glacier-related hazards in the ZLL.

L418: The 1950 debris flow event was directly triggered by the 1950 Assam earthquake (Zhang, 1992), and the root causes of the 1968, 1972 and 1984 events were the structural damage to the glacier and its exposure to lower altitudes with higher temperatures, both resulting from the 1950 earthquake.

(iii) There are also suggestions in the manuscript in places that the climate change increases the frequency of debris flow events. (e.g. L439: "Undoubtedly, the on-going warming increases the frequency of such glacier-related slope failures".). I am not convinced the data in this work speak to the presence or absence of such a link, and the link between climate change and glacier-related natural hazards can be complex. For example, the frequency of GLOFs does not necessarily just simply increase with climate warming (Veh et al., 2019; Veh et al., 2023)

- Author's response:

Thank you very much for your insightful comments. It is important to clarify that the statement in L439 is not derived directly from the data in this study. We agree that the wording in its current form is overly absolute. A more appropriate revision would be: "Recent studies have shown that the on-going climate warming increases the frequency of such glacier-related slope failures" Following this statement, we have cited case studies from similar regions globally to support the argument that the 2020 event in the ZLL catchment was very likely driven by warming.

- Author's changes in manuscript:

To clarify, we revised the sentence as follows:

Recent studies have shown that the on-going climate warming increases the frequency of such glacier-related slope failures. For instance, the number of rockfalls per decade shows a similar growing trend with mean annual air temperature in Chamonix, Mont Blanc massif, France since 1934 (Deline et al., 2015). The frequency of non-seismic rock avalanches in the glaciated Saint Elias Mountains of Alaska was associated with above-average temperatures and is expected to continue increasing with ongoing climate warming (Bessette-Kirton and Coe, 2020). Shugar et al (2021) suggested that the 2021 Chamoli catastrophic ice-rock avalanche and subsequence mass

flow resulted from a complex response of the geologic and topographic settings to regional climate change.

**Line comments**

Several of the new sections would benefit from review of the language (e.g.: "Little attentions are paid on" or "imagery for interpretating glacier changes")

●     Author's response:

Thank you very much for your valuable comments and suggestions. We have carefully reviewed the language in the new sections and made the necessary corrections.

●     Author's changes in manuscript:

We have revised "Little attentions are paid on" to "Little attention has been given to…", and "imagery for interpretating glacier changes" to "imagery for interpreting glacier changes". We also have made the necessary corrections in other sections.

L47: Maybe "triggered by" rather than "driven by"?

●     Author's response:

Thank you for your suggestion. We agree that "triggered by" is more appropriate in this context. We have revised "driven by" to "triggered by" accordingly.

●     Author's changes in manuscript:

We revised "driven by" to "triggered by".

L71: "It is believed that historical earthquakes and ongoing climate warming drove these events" needs a citation or justification. Who believes that?

●     Author's response:

Thank you for your suggestion. The driving factors behind the historical debris flow events in the ZLL catchment require further discussion in this manuscript. Therefore, we have revised "these events" to "such events" for greater precision. Additionally, we have included references to studies of similar hazards in other regions worldwide at the end of this sentence to provide proper

justification.

- Author's changes in manuscript:

  we revised the sentence as follows:

  It is believed that historical earthquakes and ongoing climate warming drove such events (Bessette-Kirton and Coe, 2020; Deline et al., 2015; Stoffel et al., 2024; Zhang et al., 2022).

L129: As I said before, I do not see an increase in precipitation rates here. The increase is so small it must surely be within the uncertainty of the scatter of the data. If you plotted the confidence bands of the regression or added the standard error of the slope of the line, I would bet it is way within uncertainty of 0 or a decreasing trend. Also, in the discussion, you say there is no significant trend in precipitation.

- Author's response:

  We sincerely thank the Reviewer for their careful review. We agree with the Reviewer's point that the observed increase in precipitation rates may fall within the uncertainty range of the data. As suggested, we have re-analyzed the data by calculating the standard error of the slope (Fig. S5). The results show that the slope value is 0.065, the standard error is 4.215. The $t$-statistic value is approximately 0.0153, which is much smaller than the critical t-value of 2.086 at the significance level of $\alpha=0.05$. This indicates that the trend is not statistically significant. In the revised manuscript, we have clarified that there is no significant trend in precipitation over the study period, aligning the discussion with the updated statistical analysis.

- Author's changes in manuscript:

  We have replaced Figure 2 with Figure S5, and changed the L109 as: The annual precipitation ranges from 514 mm to 972 mm, exhibiting notable inter-annual variation, with no distinct trend over the past 20 years.

[Figure]

**Figure S5: Annual temperature and precipitation data from 2000 to 2021 at Linzhi Meteorological Station (Data source: https://www.ncei.noaa.gov/maps/annual/).**

L414: What do you mean by "the Keefer curve did not detect any of these seismic events"? A curve cannot detect anything.

- Author's response:

Thank you for pointing out this unclear statement. The phrase "the Keefer curve did not detect any of these seismic events" is indeed poorly worded and could be misleading. The statement "the Keefer curve did not detect any of these seismic events" is a metaphorical way of saying that the seismic events did not meet the criteria defined by the Keefer curve for having a significant impact on landslide or debris flow activity. The Keefer curve relates the magnitude of an earthquake to the maximum distance from the epicenter where landslides or other seismic-induced geological phenomena are likely to occur.

In the context of the research, the authors are suggesting that while there were seismic events recorded, none of them were of a magnitude and distance combination that would be expected to trigger significant landslides or debris flows according to the Keefer curve. Therefore, the curve serves as a threshold or benchmark to assess whether the seismic activity is likely to have caused geological disturbances such as debris flows. The phrase "did not detect" is used to imply that the seismic events fell below the threshold of impact as defined by the curve.

- Author's changes in manuscript:

To clarify, we revised the sentence as follows:

Although 13 earthquakes of Mw > 5.1 occurred in 1968 and 6 earthquakes of Mw ≥ 4.5 occurred in 1984, none of these seismic events fell within the range of influence as defined by the Keefer curve (Fig. 13). This suggests that these earthquakes did not have a significant influence on the debris flow events of 1968 and 1984.

L440: "show a similar growing trend". Similar to what?

- Author's response:

There is an important increase in the frequency of rock falls and a strong correlation between the warmest periods and the occurrence of 58 rock falls in the Mont Blanc massif. As shown in Figure S6, the number of rockfalls per decade show a similar growing trend with mean annual air temperature in Chamonix, Mont Blanc massif, France since 1934 (Deline et al., 2015).

[Figure]

**Figure S6: Meanannualair temperature in Chamonix (1,040 masl) since 1934 and number of rock falls per** decade **in the West face of the Drus and on the North side of the Aiguilles de Chamonix, Mont Blanc massif, France. Dashed lines: linear regressions.**

- Author's changes in manuscript:

We made no changes to this part of the manuscript.

L448: "the trend of the 1990-2020 NVAs shows a good agreement with that of the air temperature in the same period".

●    Author's response:

As mentioned in our previous response, the correlation between slope failures and air temperature is not a strict one-to-one relationship (i.e., a higher air temperature in a specific year does not necessarily correspond to a higher rate of slope instability in the same year). This relationship is more likely to manifest on interannual and decadal time scales. Over the longer time scale of 1990–2020, there is a positive correlation between the debris-flow induced NVA and air temperature (Fig. S1). Of course, to avoid ambiguity, we changed this sentence to: However, based on the fact that the lag relationship between the fluctuation peaks of NVAs and temperature fluctuations from 1990 to 2020, it is likely that the 2020 event was triggered by the recent local warming.

●    Author's changes in manuscript:

To clarify, we revised the sentence as follows:

However, based on the fact that the lag relationship between the fluctuation peaks of NVAs and temperature fluctuations from 1990 to 2020, it is likely that the 2020 event was triggered by the recent local warming.

L459: Unclear which of the debris flows "the debris flow" corresponds to (presumably the 2020 one, but it's unclear from this paragraph).

●    Author's response:

Thank you for your suggestion. Yes, the term "the debris flow" in this paragraph refers to the debris flow that occurred in 2020. We have revised the text to replace "the debris flow" with "the 2020 debris flow" for clarity.

●    Author's changes in manuscript:

We have revised the text to replace "the debris flow" with "the 2020 debris flow".

One of the problems is that the authors do not present any higher resolution imagery between the 1950 event and 1969 (first image). The conclusions about the influence of the 1950 Assam earthquake on debris flow activity from the glacial area over 40 years are based on weak data (first manuscript version). The increasing influence with time of climatic triggers is not enough taken into consideration. Also the stats-graphs are not clear about that influence.

As such, defenders of the pure climatic influence on high-mountain hazards (see some recent papers about this analysis in the Andes) will continue to believe that those types of hazards are influenced by earthquakes only immediately after the event.

Additionally, I think that without modelling this long-term effect of earthquakes on natural hazards (and see .. site located more than 100! km away from the Assam epicentral region!), especially on such high-mountain hazards, will not be proved especially if no high-quality image material is available for the time before and after the earthquake (as in this case, with the main seismic event in 1950 !). Even for the Wenchuan earthquake, the longterm influence on increased geohazard activity is considered to have finished after 10years (as indicated by the authors as well), even for sites very close to the activated fault.

Concluding: as the authors present a very detailed study, I would still recommend a major revision (and not rejection) - but the discussion should more highlight the extreme uncertainty affecting this 'exteme long-term' effect of a major earthquake in high-mountain areas (which are obviously the most strongly affected by climatic variations), especially as the zone is located so far from the epicentral area, and also of the fault (if I see well).

- Author's response:

  Thank you very much for your insightful comments. Unfortunately, we were unable to obtain high-resolution remote sensing imagery for the period between the 1950 event and 1969, which indeed would provide the most direct evidence. However, the 1950, 1968, and 1984 events are well-documented in the literature (Zhang and Shen, 2011; Zhang, 1985, 1992). The 1950 event occurred immediately following the Assam earthquake and is considered to be directly associated with the seismic activity. Notably, the earthquake triggered simultaneous debris flows in as many as 13 gullies in the Yarlung Tsangpo Grand Canyon area (Liu, 1984). During the 1950 event, the

terminus of the ZLL trunk glacier advanced from 3650 m a.s.l to the Yarlung Tsangpo at 2810 m a.s.l, forming a glacial dam in the Yarlung Tsangpo River.

During the 1968 event, the ZLL Glacier again formed a dam over 50 m high in the Yarlung Tsangpo River, while the 1972 event was similarly caused by the local collapse of the glacier terminus (Fig. S7-b3 and c3). The images from 1979 also show significant glacier fragmentation and differential ablation (Fig. S7-e3). Compared to 1979, glacier ablation intensified in 1982, exposing lateral moraines (Fig. S7-e2 and S7-f3), and the lowest section of the glacier developed numerous crevasses (Fig. S7-f4 and S7-f5). These fractured ice bodies and moraine materials, under the impact of the 1984 ice avalanche at 3700 m observed by the First Qinghai-Tibet Scientific Expedition (Zhang, 1992), contributed to the formation of the 1984 large-scale debris flow. By 1989, three of the glacier segments had melted, detached from the main glacier, and were buried under thick moraine deposits. These events clearly indicate that the frequent ruptures, collapses, and differential melting of the ZLL Glacier were associated with disturbances caused by the 1950 earthquake. While rising temperatures likely contributed to the melting and instability of the glacier, temperature alone could not have caused the direct collapse of the stable trunk glacier and the subsequent debris flows. The root causes were the structural damage to the glacier and its exposure to lower altitudes with higher temperatures, both of which were consequences of the 1950 earthquake. Since 1990, the only large-scale event in this region has been the 2020 event, which was triggered by an ice-rock avalanche on the southern ridge of the catchment, unrelated to the trunk glacier. The frequency of large-scale disaster events in the 40 years after the 1950 earthquake was significantly higher than that after 1990, suggesting that glacier instability during this period can be regarded as a preconditioning effect of the earthquake.

[Figure]

**Figure S7: Variations of the Zelunglung alluvial fan and channel during 1969 – 1982. The images are taken from Keyhole reconnaissance satellites (https://earthexplorer.usgs.gov/).**

The duration of earthquake preconditioning effects shows considerable regional heterogeneity. Even for the same earthquake, estimates of the preconditioning duration can vary across studies. For example, the ChiChi earthquake preconditioning lasted for 2-5 years (Marc et al., 2015). Cui et al (2011) predicted that the Wenchuan earthquake's influence on post-seismic debris flows

would persist for 10–20 years, while Huang (2011) estimated 20–25 years. Thus, the impact of large earthquakes on glacial or peri-glacial regions, especially in small glaciated catchments, may differ significantly from those in non-glacial regions.

Furthermore, the 2015 Gorkha Mw 7.8 earthquake in Nepal triggered co-seismic landslides extending up to 130 km from the epicenter. Beyond this range, a slight increase in landslide frequency was attributed to the contribution of the Mw 7.3 Dolakha aftershock (~140 km from the mainshock) (Martha et al., 2017). The 1950 Assam earthquake, with its epicenter approximately 199 km from the ZLL catchment, had a very high magnitude (Mw 8.6) and occurred in the tectonically active eastern Himalayan syntaxis. Coudurier-Curveur et al. (2020) recalibrated the aftershock distribution within the first four months post-earthquake, revealing a proximal aftershock up to Mw 5.5 adjacent (~1km) to the ZLL catchment (Fig. S8). Therefore, coupled with subsequent high-magnitude aftershocks near the ZLL catchment (Fig S9), the seismic impact on the ZLL catchment was significantly amplified despite the distance. This seismic event also triggered a prolonged period of debris flow activity, persisting for decades, in Guxianggou, approximately 50 kilometers northeast of the ZLL Valley (Du and Zhang, 1981).

[Figure]

**Figure S8: Relocated aftershocks of the 1950 Assam earthquake (using fixed depth relocation) (Coudurier-Curveur et al., 2020).**

[Figure]

**Figure S9: Distance from epicenters of the collected seismic events to the Zelunglung vs. the seismic magnitude (the black solid curve refers to Keefer (1984)).**

● Author's changes in manuscript:

We have replaced Figure 3 with Figure S7 and Figure 13 with Figure S8. Additionally, the following explanation has been added to the manuscript:

L221: The fan in December 1975 exhibits significant brightness variations (Fig. 3-d2), with pronounced channelization above the glacier (Fig. 3-d1), indicatingpossible debris flow activity prior to this time. Compared with 1975, the fan in 1979 displays a flatter terrain and more distinct channelization (Fig. 3-e2), indicating the modification of the rough fan surface by debris flow activity. This also implies that, due to limited information at the time, additional events during this period may have gone unrecorded. By 1982, noticeable vegetation had recovered in the middle part of the fan (Fig. 3-f2). Concurrently, accelerated glacier ablation exposed lateral moraines (Fig. 3-e2 and 3-f3), while the glacier terminus developed an extensive crevasse network (Fig. 3-f4 and 3-f5). These fractured ice bodies and moraine materials, under the impact of ice avalanche at 3700 m described by Zhang (1992), contributed to the formation of the 1984 large-scale debris flow.

L409: Notably, the impact distance of a large earthquake can reach hundreds of kilometers. For example, the co-seismic landslides triggered by the 2015 Gorkha Mw 7.8 earthquake extended

to a distance of over 130 km from the epicenter (Martha et al., 2017). The 1950 Assam earthquake, with its epicenter approximately 199 km from the ZLL, had a very high magnitude (Mw 8.6) and occurred in the tectonically active eastern Himalayan syntaxis. Coupled with subsequent high-magnitude aftershocks near the ZLL (Fig. 13), the seismic impact on the ZLL was significantly amplified despite the distance.

L418: The 1950 debris flow event was directly triggered by the 1950 Assam earthquake (Zhang, 1992), and the root causes of the 1968, 1972 and 1984 events were the structural damage to the glacier and its exposure to lower altitudes with higher temperatures, both resulting from the 1950 earthquake.

**Other changes by the author**

1. We have corrected and replaced Figures 8 and 14 because of errors in the image sequence number and font.

2. In addition to the language expression problems pointed out by the reviewers, we further checked and proofread the possible language errors.

3. We have modified all the reference formats according to the requirements of Copernicus Publications.

**Reference:**

Bessette-Kirton, E. K. and Coe, J. A.: A 36-Year Record of Rock Avalanches in the Saint Elias Mountains of Alaska, With Implications for Future Hazards, FRONT EARTH SC-SWITZ, 8, https://doi.org/10.3389/feart.2020.00293, 2020.

Coe, J. A., Bessette-Kirton, E. K., and Geertsema, M.: Increasing rock-avalanche size and mobility in Glacier Bay National Park and Preserve, Alaska detected from 1984 to 2016 Landsat imagery, LANDSLIDES, 15, 393–407, https://doi.org/10.1007/s10346-017-0879-7, 2018.

Coudurier-Curveur, A., Tapponnier, P., Okal, E., Van der Woerd, J., Kali, E., Choudhury, S., Baruah, S., Etchebes, M., and Karakaş, Ç.: A composite rupture model for the great 1950 Assam earthquake across the cusp of the East Himalayan Syntaxis, Earth and Planetary Science Letters, 531, 115928, https://doi.org/10.1016/j.epsl.2019.115928, 2020.

Cui, P., Chen, X.-Q., Zhu, Y.-Y., Su, F.-H., Wei, F.-Q., Han, Y.-S., Liu, H.-J., and Zhuang, J.-Q.: The Wenchuan Earthquake (May 12, 2008), Sichuan Province, China, and resulting geohazards, NAT HAZARDS, 56, 19–36, https://doi.org/10.1007/s11069-009-9392-1, 2011.

Deline, P., Gruber, S., Delaloye, R., Fischer, L., Geertsema, M., Giardino, M., Hasler, A., Kirkbride, M., Krautblatter, M., Magnin, F., McColl, S., Ravanel, L., and Schoeneich, P.: Ice Loss and Slope Stability in High-Mountain Regions, in: Snow and Ice-Related Hazards, Risks, and Disasters, edited by: Shroder, J. F., Haeberli, W., and Whiteman, C., Academic Press, Boston, USA, 521–561, https://doi.org/10.1016/B978-0-12-394849-6.01001–5, 2015.

Deng, M., Chen, N., and Liu, M.: Meteorological factors driving glacial till variation and the associated periglacial debris flows in Tianmo Valley, south-eastern Tibetan Plateau, Nat. Hazards Earth Syst. Sci., 17, 345–356, https://doi.org/10.5194/nhess-17-345-2017, 2017.

Du R. and Zhang S.: CHARACTERISTICS OF GLACIAL MUD-FLOWS IN SOUTH-EASTERN QINGHAI-XIZANG PLATEAU, Journal of Glaciology and Geocryology, 10–16, 81–82, 1981.

Hu, K., Zhang, X., You, Y., Hu, X., Liu, W., and Li, Y.: Landslides and dammed lakes triggered by the 2017 Ms6.9 Milin earthquake in the Tsangpo gorge, Landslides, 16, 993–1001, https://doi.org/10.1007/s10346-019-01168-w, 2019.

Huang, R.: AFTER EFFECT OF GEOHAZARDS INDUCED BY THE WENCHUAN EARTHQUAKE, gcdzxb, 19, 145–151, 2011.

Jones, J. N., Boulton, S. J., Stokes, M., Bennett, G. L., and Whitworth, M. R. Z.: 30-year record of Himalaya mass-wasting reveals landscape perturbations by extreme events, NAT COMMUN, 12, 6701, https://doi.org/10.1038/s41467-021-26964-8, 2021.

Keefer, D. K.: Landslides caused by earthquakes, Geol Soc America Bull, 95, 406, https://doi.org/10.1130/0016-7606(1984)95<406:LCBE>2.0.CO;2, 1984.

Liu S.: DEBRIS FLOWS IN THE MT.NAMJAGBARWA REGION, Mountain Research, 212–215, 1984.

Marc, O., Hovius, N., Meunier, P., Uchida, T., and Hayashi, S.: Transient changes of landslide rates after earthquakes, GEOLOGY, 43, 883–886, https://doi.org/10.1130/G36961.1, 2015.

Martha, T. R., Roy, P., Mazumdar, R., Govindharaj, K. B., and Kumar, K. V.: Spatial characteristics of landslides triggered by the 2015 Mw 7.8 (Gorkha) and Mw 7.3 (Dolakha) earthquakes in Nepal, LANDSLIDES, 14, 697–704, https://doi.org/10.1007/s10346-016-0763-x, 2017.

Peng, D., Zhang, L., Jiang, R., Zhang, S., Shen, P., Lu, W., and He, X.: Initiation mechanisms and dynamics of a debris flow originated from debris-ice mixture slope failure in southeast Tibet, China, Engineering Geology, 307, 106783, https://doi.org/10.1016/j.enggeo.2022.106783, 2022.

Shugar, D. H., Jacquemart, M., Shean, D., Bhushan, S., Upadhyay, K., Sattar, A., Schwanghart, W., McBride, S., de Vries, M. V. W., Mergili, M., Emmer, A., Deschamps-Berger, C., McDonnell, M., Bhambri, R., Allen, S., Berthier, E., Carrivick, J. L., Clague, J. J., Dokukin, M., Dunning, S. A., Frey, H., Gascoin, S., Haritashya, U. K., Huggel, C., Kääb, A., Kargel, J. S., Kavanaugh, J. L., Lacroix, P., Petley, D., Rupper, S., Azam, M. F., Cook, S. J., Dimri, A. P., Eriksson, M., Farinotti, D., Fiddes, J., Gnyawali, K. R., Harrison, S., Jha, M., Koppes, M., Kumar, A., Leinss, S., Majeed, U., Mal, S., Muhuri, A., Noetzli, J., Paul, F., Rashid, I., Sain, K., Steiner, J., Ugalde, F., Watson, C. S., and Westoby, M. J.: A massive rock and ice avalanche caused the 2021 disaster at Chamoli, Indian Himalaya, Science, 373, 300–306, https://doi.org/10.1126/science.abh4455, 2021.

Stoffel, M., Trappmann, D. G., Coullie, M. I., Ballesteros Cánovas, J. A., and Corona, C.: Rockfall from an increasingly unstable mountain slope driven by climate warming, NAT GEOSCI, 17, 249–254, https://doi.org/10.1038/s41561-024-01390-9, 2024.

Zhang, J. and Shen, X.: Debris-flow of Zelongnong Ravine in Tibet, J. Mt. Sci., 8, 535–543, https://doi.org/10.1007/s11629-011-2137-0, 2011.

Zhang W.: Some features of the surge glagier in the MT. Namjagbarwa, Mountain Research, 234–238, 1985.

Zhang, W.: Identification of glaciers with surge characteristics on the Tibetan Plateau, Ann. Glaciol., 16, 168–172, https://doi.org/10.3189/1992AoG16-1-168-172, 1992.

Zhang, X., Hu, K., Liu, S., Nie, Y., and Han, Y.: Comprehensive interpretation of the Sedongpu glacier-related mass flows in the eastern Himalayan syntaxis, J. Mt. Sci., 19, 2469–2486, https://doi.org/10.1007/s11629-022-7376-8, 2022.

Zou Q., Zhou B., Yang T., Chen S., Yao H., Jiang H., and Zhou W.: Spatio-Temporal Differentiation Characteristics of  Glacial Lake Outburst in the Himalayas, Earth Science, 49, 4047–4062, https://doi.org/10.3799/dqkx.2024.083, 2024.

---

## Author Response (AR3)

**Reviewer 1**

**The manuscript provides a comprehensive and data-rich investigation of periglacial mass movements in the Zelunglung catchment, Eastern Himalayas, and presents the first long-term analysis of sediment dynamics in this seismically active region since the 1950 Assam earthquake. The authors combine satellite data, field surveys, UAV imagery, and long-term meteorological and seismological data. The study offers a valuable case study on the interaction of tectonics, climate, and geomorphological processes in high mountain regions. The manuscript features a high diversity of data. The use of 30 satellite image datasets (1969–2022), UAV surveys, and field data is of great value. The methodology for identifying non-vegetated areas (NVA) is transparent and mostly comprehensible. The contextualization of the five documented events (1950, 1968, 1972, 1984, 2020) in relation to seismic and climatic changes allows for well-founded conclusions on their temporal evolution. The integration of satellite data, UAV data, visual interpretation, field measurements, and literature sources leads to robust results.**

We are grateful for the reviewer's positive recognition of our manuscript. And we are very grateful for various lines of constructive criticism which were very helpful in improving analysis and the manuscript. Point-by-point replies can be found in the following:

*(Please note that all references in this response, including line numbers, section titles, and figure/table numbers, are based on the Manuscript_trackchanges.docx file, so that the reviewer can clearly locate the revisions and compare them with the original content.)*

1. **Quantification of Uncertainties:**
   a) **The uncertainties in deriving the NVA as a proxy for sediment volume could be treated more rigorously using statistical methods. A quantitative error assessment is missing.**

- **Author's response:**

We greatly appreciate this valuable comment. We fully agree that a rigorous statistical treatment would strengthen the reliability of the NVA as a proxy for sediment volume. The sediment volume of debris flow may be influenced by many factors, such as fan area, fan slope and sediment thickness. However, the scarcity of long-term debris-flow volume measurements in our study region limits the feasibility of developing a robust statistical model. Borehole method can give the accurate estimation of thickness, but is too expensive for our study. It should be pointed out that accurate estimation of sediment volume from the NVA is not the purpose of this study. Our emphasis lies not

on the absolute volume of debris flows but on their temporal trends. Although uncertainties remain, the overall temporal trends of the two quantities are similar. Consequently, this study employs an approximate alternative approach, wherein the fluctuation in debris flow volume is inferred from changes in the area of the accumulation fan. Actually, many previous studies consider debris flow volume is empirically a function of the accumulation area (e.g. Iverson et al. (1998)). We acknowledge that the uncertainties in deriving the NVA as a proxy for sediment volume and could be treated more rigorously in future study, and we have highlighted this issue in the revised manuscript.

● **Author's changes in manuscript:**
We have added a new discussion section to emphasize both the validity and the limitations of using NVA as a proxy:

5.4 Limitation
Previous studies have shown that debris-flow volume is often empirically related to the extent of the inundation area (e.g., Iverson et al., 1998), which supports the use of NVA as a proxy for sediment volume. Multi-periodic periglacial debris flows are strongly associated with variations in the NVA of the alluvial fan, suggesting that NVA can capture long-term trends in debris-flow activity. In practice, however, the NVA includes a fixed portion of the area inundated by the main river and is therefore slightly larger than the actual depositional area caused by debris flows. Technically, the contribution of the main river to NVA cannot be entirely excluded. Nevertheless, the riverbank line remained stable from the 1980s to the 2010s, during which no large periglacial debris flows occurred (Figs. 11b and c; Zhang and Shen, 2011). It is therefore reasonable to assume that variations in river water level have little influence on changes in NVA, and that NVA primarily reflects the relative volume trends of sediment transported by debris flows.
Despite these considerations, several uncertainties remain in using NVA as a volume proxy. The absence of systematic depositional thickness measurements prevents direct conversion of NVAs into absolute debris-flow volumes, so NVA only reflects relative fluctuations rather than precise values. Delineation errors in historical imagery, particularly in earlier black-and-white aerial photographs with limited tonal contrast, may affect accuracy. The spatial resolution of imagery also varies markedly across decades, which inevitably affects the precision of NVA delineation and may lead to scale-dependent biases when comparing different periods. Error ranges for NVA were plotted to illustrate these uncertainties (Fig. 12d), and although they cannot be entirely eliminated, they do not alter the main analytical results. Moreover, the empirical relationship between inundation area and debris-flow volume may vary with local geomorphic and hydrological conditions, such as fan slope and gully confinement, further complicating volume inference and limiting the applicability of uniform statistical error models.
While accurate estimation of sediment volume from NVA is beyond the scope

of this study, we acknowledge the associated uncertainties, which warrant more rigorous treatment in future study. Future studies that integrate high-resolution LiDAR, UAV photogrammetry, or borehole surveys with field-based volume measurements could provide more robust statistical assessments of

the NVA–volume relationship. The application of dense stereo-pair techniques

for DSM extraction from historical and modern satellite archives also has considerable potential to provide three-dimensional constraints on sediment thickness and deposition, thereby improving the translation from inundation area to sediment volume.

**b) The translation of NVA into actual sediment volumes should be discussed more critically.**

● **Author's response:**

We agree with the reviewer that NVA cannot be directly translated into absolute debris-flow volumes due to the absence of depositional thickness data. In this study, we use NVA only as a proxy to reflect relative changes in debris-flow activity through time, rather than to estimate exact sediment volumes. We have added a Limitations section to clarify this distinction more explicitly and emphasize the assumptions and uncertainties underlying this approach.

● **Author's changes in manuscript:**

We have added a new discussion section (5.4 Limitation) to emphasize both the validity and the limitations of using NVA as a proxy:

**c) For the differential elevation models shown, stable areas should also be presented, and the DoD values in those areas should be critically discussed. At minimum, the arithmetic mean, RMSE, maximum value, minimum value, and standard deviation should be reported.**

● **Author's response:**

Thank you for your suggestion. We have improved this part in the first revision. The reconstruction and differencing of DSMs are carried out in Pix4DMapper and Arcmap10.8. Since we did not deploy ground control points during drone photography, we generated DSM and DOM of September 9 in Pix4DMapper, and then selected 20 relatively stable points that were not affected by debris flow events as control points in Arcmap with DOM of September 9 as reference. These control points were then used in Pix4DMapper to generate the September 11 DSM and DOM. To determine the uncertainty for our UAV DSMs of difference (DoD) differencing result we follow methods outlined in Shugar et al. (2021). We identified a series of fifteen stable areas on old debris flow terraces adjacent to the valley floor (Mainly roads and unseeded

farmlands) and retrieved the standard deviation of DoD values within these areas and used these to estimate a two-sigma DoD uncertainty. The resulting elevation uncertainty was ±0.493 m, corresponding to a DoD volumetric uncertainty of ±1.85 × 10⁴ m³ (Line 422). Owing to the large extent of the study area, and to avoid redundancy while presenting geomorphic changes in the debris-flow impact zone more clearly, we did not include the results from the stable areas in the manuscript figure, but we provide them here in this response (Fig. S1).

[Figure]

Figure S1. DoD results for the debris-flow inundation zone and geomorphically stable areas.

● **Author's changes in manuscript:**
We have already included the relevant additions in the first revision; please refer to Section 3.2.3 for details.

2. **Assessment of Climatic Control:**
   **The attribution of the 2020 event solely to temperature rise is largely understandable but should be formulated more cautiously. The link between the sudden temperature increase (2.5 °C in 2018) and**

**immediate ice/rock instability appears more speculative than causal.**

● **Author's response:**

Thank you for your suggestion. Yes, we agree with your perspective. In this revision, we have adopted a more conservative phrasing. In addition, we conducted a more in-depth analysis and found that the NVA has exhibited four peaks since 2000. These peaks likely correspond to the small flash-flood debris flow events described by Zhang et al. Interestingly, these peaks lag behind the summer temperature maxima by 2–4 years, which is consistent with the 2020 event occurring two years after the abrupt warming in 2018. Notably, similar lag phenomena have also been observed in other comparable regions(Deng et al., 2017; Stoffel et al., 2024). We consider this regular pattern as further evidence supporting the contribution of temperature.

[Figure]

Figure 12: (a) Seismic events within a 200 km distance to the Zelunglung from 1940 to the present. (b) Changes in the annual mean and summer air temperatures in the Zelunglung from 1940 to the present. (c) Changes in the annual and summer precipitation in the Zelunglung from 1940 to the present. (d) Changes in the non-vegetated area of the Zelunglung alluvial fan from 1969 to the present (although the deposition of the 1950 event did not happen at the Zelunglung's outlet like the later events, we plot the NVA of the 1950 event as the starting point). (e) Changes in the annual and summer precipitation in the Zelunglung from 1990 to 2017. (f) Changes in the non-vegetated area of the Zelunglung alluvial fan from 1990 to 2017.

- **Author's changes in manuscript:**

We have updated the figures and conducted a more in-depth analysis and discussion. Please refer to the revised manuscript for details.

3. **Structure and Readability:**
   **The text is very data-heavy, but at times difficult to follow. A clearer structuring of the results (e.g., subchapters for each individual event) would be helpful.**

- **Author's response:**

Thank you for your suggestion. We have revised the structure of the Conclusions section.

- **Author's changes in manuscript:**

The "4.1 Rapid glacier changes" section has been moved to the study area description, as it is based on literature review. The "5 Multi-periodic sedimentation in the confluence" section has been incorporated into "4 Results." The final "4 Results" section is divided into three subsections: "4.1 Multi-periodic glacial debris flows," "4.2 Sediment characteristics of the 2020 event," and "4.3 Multi-periodic sedimentation in the confluence." Section 4.1 mainly focuses on analyses of historical single debris-flow events, including a brief introduction to the 2020 event, while Section 4.2 presents a detailed sedimentological analysis of the 2020 typical event.

4. **Comparison with Other Regions / Studies:**
   **A broader contextualization of the Zelunglung cases in comparable scenarios would be beneficial, particularly with regard to recurrence intervals, mobilized volumes, and downstream effects on river systems. This should be incorporated into the discussion.**

- **Author's response:**

Thank you for your suggestion; we have added the relevant discussion.

- **Author's changes in manuscript:**

We have added the relevant discussion in the following sections:

[revised manuscript text omitted]

5. **Further Suggestions with Line References:**
   a) **Line 11: The phrasing that the earthquake "triggered debris flows" should be improved. Earthquakes do not directly trigger debris flows but may prepare material for them. Ultimately, a significant amount of water is still needed for debris flows to occur.**

● **Author's response:**

Thank you for your suggestion. We have revised the phrasing accordingly and have also reviewed the rest of the manuscript to avoid similar expressions elsewhere.

● **Author's changes in manuscript:**
We have revised Line 11 as follows:
Periglacial debris flows in alpine mountains are influenced by strong earthquakes or climatic warming and play a crucial role in delivering sediment from hillslopes and downslope channels into rivers.

   b) **Line 33: The statement "rising temperatures and increased extreme precipitation events" requires a citation.**

● **Author's response:**

Thank you for your suggestion. We cited the studies by Giorgi et al. (2016), Luan and Zhai (2023), Castino et al. (2016), and Frich et al. (2007) and Myhre et al. (2019) to demonstrate that in high mountain regions such as the Alps, the High Mountain Asia, and the Andes, and even globally, there is a regional trend of increasing extreme rainfall events under the context of climate warming. Correspondingly, we referenced several regional and global statistical studies (Wang et al., 2024; Zhang et al., 2023) to support the evidence that, under this context, the frequency and magnitude of glacier-related hazards have increased.

*Refenence:*

Castino, F., Bookhagen, B., and Strecker, M. R.: Rainfall variability and trends of the past six decades (1950–2014) in the subtropical NW Argentine Andes, CLIM DYNAM, 48, 1049–1067, https://doi.org/10.1007/s00382-016-3127-2, 2016.

Frich, P., Alexander, L., Della-Marta, P., Gleason, B., Haylock, M., Klein Tank, A., and Peterson, T.: Observed coherent changes in climatic extremes during the second half of the twentieth century, CLIM RES, 19, 193–212, https://doi.org/10.3354/cr019193, 2007.

Giorgi, F., Torma, C., Coppola, E., Ban, N., Schär, C., and Somot, S.: Enhanced summer convective rainfall at Alpine high elevations in response to climate warming, NAT GEOSCI, 9, 584–589, https://doi.org/10.1038/ngeo2761, 2016.

Luan L. and Zhai P.: hanges in rainy season precipitation properties over the Qinghai-Tibet Plateau based on multi-source datasets, Climate Change Research, 173–190, 2023.

Myhre, G., Alterskjær, K., Stjern, C. W., Hodnebrog, Ø., Marelle, L., Samset, B. H., Sillmann, J., Schaller, N., Fischer, E., Schulz, M., and Stohl, A.: Frequency of extreme precipitation increases extensively with event rareness under global warming, SCI REP-UK, 9, 16063, https://doi.org/10.1038/s41598-019-52277-4, 2019.

Wang, H., Wang, B.-B., Cui, P., Ma, Y.-M., Wang, Y., Hao, J.-S., Wang, Y., Li, Y.-M., Sun, L.-J., Wang, J., Zhang, G.-T., Li, W.-M., Lei, Y., Zhao, W.-Q., Tang, J.-B., and Li, C.-Y.: Disaster effects of climate change in High Mountain Asia: State of art and scientific challenges, ADV CLIM CHANG RES, 15, 367–389, https://doi.org/10.1016/j.accre.2024.06.003, 2024.

Zhang, T., Wang, W., Shen, Z., and An, B.: Increasing frequency and destructiveness of glacier-related slope failures under global warming, SCI BULL, 69, 30–33, https://doi.org/10.1016/j.scib.2023.09.042, 2023.

- **Author's changes in manuscript:**
  We have revised the statement on Line 30 and updated the corresponding citation in the References section:

  Under climate change, characterized by rising temperatures and more frequent extreme precipitation events(Castino et al., 2016; Frich et al., 2007; Giorgi et al., 2016; Luan and Zhai, 2023; Myhre et al., 2019), high-altitude regions are increasingly affected by more destructive and frequent ice/rock avalanches, as well as low-angle glacier detachments (Wang et al., 2024; Zhang et al., 2023).

**c) Line 81: The percentage of glaciation within the catchment should be calculated and provided.**

● **Author's response:**

Thank you for your suggestion. We have added information on the proportion of glacier area.

● **Author's changes in manuscript:**
 We revised Line 87-88 to:
The catchment extends over 41.21 km², with 17.06 km² (41.4%) covered by glaciers (RGI 7.0).

**d) Line 88: What time period do the referenced earthquakes cover? Although historical earthquakes are mentioned, the exact time frame would be interesting to know.**

● **Author's response:**

Thank you for your suggestion. The time span of the earthquakes shown in the figure 1b covers 1940–2020, which is consistent with the historical earthquake data used for statistical analysis later in the text. To avoid any potential confusion for readers, we have added this information to the figure caption.

● **Author's changes in manuscript:**
We have revised the figure title to:
Historical earthquakes from 1940 to 2020 were sourced from the United States Geological Survey (USGS) National Earthquake Information Center (NEIC) (https://earthquake.usgs.gov/earthquakes/search/).

**e) Line 107: While there is a clear increase in temperature, the precipitation data, in my opinion, shows no discernible trend. The cited precipitation trend of 0.65 mm/decade may be due to measurement uncertainty. Thus, its climatic significance is questionable. There are strong fluctuations and differences over time, which should be discussed more thoroughly. A general statement about increasing precipitation should be avoided.**

● **Author's response:**

We sincerely thank the Reviewer for their careful review. We agree with the Reviewer's point that the observed increase in precipitation rates may fall within the uncertainty range of the data. As suggested, we have re-analyzed the data by calculating the standard error of the slope (Fig. 2). The results show that the slope value is 0.065, the standard error is 4.215. The t-statistic value is approximately 0.0153, which is much smaller than the critical t-value of 2.086 at the significance level of $\alpha=0.05$. This indicates that the trend is not statistically significant. In the revised manuscript, we have clarified that there

is no significant trend in precipitation over the study period, aligning the discussion with the updated statistical analysis.

**Author's changes in manuscript:**

We have updated Figure 2, and changed the L120 as: The annual precipitation ranges from 514 mm to 972 mm, exhibiting notable inter-annual variation, with no distinct trend over the past 20 years.

[Figure]

Figure 2: Annual temperature and precipitation data from 2000 to 2021 at Linzhi Meteorological Station. (Data source: https://www.ncei.noaa.gov/maps/annual/ ).

**f) Line 112: A subheading "satellite images" should be inserted here.**

● **Author's response:**

Thank you for your suggestion. We have inserted the subheading "3.1.1 Satellite images" and also listed the earthquake and meteorological data used under "3.1.2 Earthquake and climate."

● **Author's changes in manuscript:**

We have inserted the subheading "3.1.1 Satellite Images" in L151 and additionally added the following subsections:

*3.1.2 Earthquake and climate*

Earthquake and climate datasets were used to investigate the potential linkages between these factors and debris-flow occurrence. Earthquake records within approximately 400 km of the ZLL catchment during 1940–2020 were obtained from the United States Geological Survey (USGS) National Earthquake Information Center (NEIC) (https://earthquake.usgs.gov/earthquakes/search/). In addition, gridded mean values of annual mean air temperature, summer air temperature, annual

precipitation, and summer precipitation for the ZLL catchment during 1940–2021 were derived from the 1-km monthly precipitation and mean temperature dataset for China (1901–2021) (Peng, 2019, 2020). The reliability of these datasets has been verified against 496 independent meteorological observation stations across China (Peng et al., 2019) .

**g) Lines 128–130: The procedure (cross-sectional morphology, debris flow particle characteristics, and the extent of damage to the Zhibai Bridge) should be described in more detail.**

● **Author's response:**

Thank you for your suggestion. We have provided a more detailed description in this section, including the specific locations of the survey, the sampling sites, and the instruments and methods used for the measurements.

● **Author's changes in manuscript:**

We revised Lines 180–192 as follows:

We conducted three field surveys at ZLL between 2020 and 2022. The first survey (September 9, 2020) employed a DJI MAVIC-2 UAV to perform geomorphological photogrammetry of the downstream channel and alluvial fan. During the second survey (September 11, 2020), we combined low-altitude UAV photogrammetry with measurements from an IMETER LF1500A laser rangefinder to characterize the downstream channel morphology, particularly near Zhibai Bridge, to analyze debris-flow erosion and deposition patterns. UAV photographs also provided close-up views of inaccessible upstream sections. Tape measurements were used to determine bridge displacement and boulder sizes on the fan, while low-altitude UAV orthophotos supported post-event interpretation of boulder distribution. Fine-grained deposits (< 100 mm) were sampled from the fan apex for laboratory analyses. The third survey (December 21, 2022) used UAV imaging to generate a complete 3D view of ZLL (Fig. 1d).

**h) Lines 138–139: Potential sources of error and uncertainties should be discussed.**

● **Author's response:**

Thank you for your suggestion; we have added a discussion on errors and uncertainties.

● **Author's changes in manuscript:**

We have added a new section to discuss the potential sources of error and uncertainties:

5.4 Limitation

Previous studies have shown that debris-flow volume is often empirically related to the extent of the inundation area (e.g., Iverson et al., 1998), which

supports the use of NVA as a proxy for relative debris-flow magnitude. Multi-period periglacial debris flows are strongly associated with variations in the NVA of the alluvial fan, suggesting that NVA can capture long-term trends in debris-flow activity. In practice, however, the NVA includes a fixed portion of the area inundated by the main river and is therefore slightly larger than the actual inundation area caused by debris flows. Technically, the contribution of the main river to NVA cannot be entirely excluded. Nevertheless, the riverbank line remained stable from the 1980s to the 2010s, during which no large periglacial debris flows occurred (Figs. 11b and c; Zhang and Shen, 2011). It is therefore reasonable to assume that variations in river water level have little influence on changes in NVA, and that NVA primarily reflects the relative volume trends of sediment transported by debris flows.

Despite these considerations, several uncertainties remain in using NVA as a volume proxy. The absence of systematic sediment thickness measurements prevents direct conversion of NVAs into absolute debris-flow volumes, so NVA only reflects relative fluctuations rather than precise values. Delineation errors in historical imagery, particularly in earlier black-and-white aerial photographs with limited tonal contrast, may affect accuracy. The spatial resolution of imagery also varies markedly across decades, which inevitably affects the precision of NVA delineation and may lead to scale-dependent biases when comparing different periods. Error ranges for NVA were plotted to illustrate these uncertainties (Fig. 12d), and although they cannot be entirely eliminated, they do not alter the main analytical results. Moreover, the empirical relationship between inundation area and debris-flow volume may vary with local geomorphic and hydrological conditions, such as fan slope and gully confinement, further complicating volume inference and limiting the applicability of uniform statistical error models.

Future research integrating high-resolution LiDAR, UAV photogrammetry, or borehole surveys with field-based volume measurements would enable more rigorous statistical assessments of the NVA–volume relationship. The application of dense stereo-pair techniques for DSM extraction from historical and modern satellite archives also has considerable potential to provide three-dimensional constraints on sediment thickness and deposition, thereby improving the translation from inundation area to sediment volume.

**i) Line 140: This should be supported by a graphic. The methodology should be explained better using visual examples.**

● **Author's response:**

Thank you for your suggestion. We have prepared a conceptual diagram illustrating the identification of NVA to help readers better understand. The inundation of debris flow on the alluvial fan often destroys vegetation cover and causes the affected area desertification. Based on differences in color,

tone, texture, and shading between vegetated and non-vegetated areas in satellite imagery for a given year, we delineated the debris flow inundation zone (i.e., NVA) for that year. When subsequent debris flows occurred and extended beyond the gully outlet, three scenarios were observed: (1) the subsequent debris flow was of a larger magnitude, exceeding the previous inundation zone; (2) the subsequent debris flow was smaller in magnitude but caused damage to newly established vegetation; and (3) the subsequent debris flow was very small in scale, confined to the channel or a very limited area along its banks, and did not affect vegetation. The third scenario is more appropriately classified as a minor seasonal flood with negligible sediment transport compared with debris flows. Therefore, in our interpretation and statistical analyses, the NVA was restricted to the first two scenarios.

[Figure]

Figure 3: Conceptual illustration of non-vegetated area (NVA) interpretation

● **Author's changes in manuscript:**

We have added a conceptual diagram illustrating the identification of NVAs in the manuscript (L214–L215), and provided a more detailed explanation in L202–L210:

Based on differences in color, tone, texture, and shading between vegetated and non-vegetated areas in satellite imagery for a given year, we delineated the debris flow inundation zone (i.e., NVA) for that year. When subsequent debris flows occurred and extended beyond the gully outlet, three scenarios were observed: (1) the subsequent debris flow was of a larger magnitude, exceeding the previous inundation zone; (2) the subsequent debris flow was smaller in magnitude but caused damage to newly established vegetation; and (3) the subsequent debris flow was very small in scale, confined to the channel or a very limited area along its banks, and did not affect vegetation. The third scenario is more appropriately classified as a minor seasonal flood with negligible sediment transport compared with debris flows. Therefore, in our interpretation and statistical analyses, the NVA was restricted to the first

two scenarios.

**j)  Line 158: The heading "Results" should be inserted here.**

● **Author's response:**

Thank you for your suggestion. We have inserted the "4 Results" section heading here.

● **Author's changes in manuscript:**

We have inserted the "4 Results" section heading here.

**k)  Line 331: The value range of the DoD figure should be symmetrically centered around zero (e.g., –21 m to +21 m) to visually balance erosion and deposition and facilitate interpretation. Stable areas should also be shown, and a DoD error assessment should be carried out.**

● **Author's response:**

Thank you for your suggestion. We have updated the DoD range to –21 m to +21 m (Fig. 10). In our first revision, we quantified the DoD volumetric uncertainty as ±1.85 × 10⁴ m³ (Line 422). Owing to the large extent of the study area, and to avoid redundancy while presenting geomorphic changes in the debris-flow impact zone more clearly, we did not include the results from the stable areas in the manuscript figure, but we provide them here in this response (Fig. S1).

[Figure]

Figure 10: Geomorphic changes of the channel and alluvial fan after the debris flows of 2020. (a) Erosion and deposit depth caused by the debris flows. The base map is taken by UAV on 10 Sep 2020. (b) Photo of the channel after the debris flows. The red line represents the cross-section next to the Zhibai Bridge (photo taken on 11 Sep 2020). (c) Cross-sections before (black) and after (red) the debris flows.

- **Author's changes in manuscript:**
 We have updated Fig. 10.

6. **Entire Results Chapter:**
   **There is already extensive discussion included in the results chapter. A better separation between the results and the actual discussion would be desirable. This would give the paper a clearer structure and organization. Alternatively, the results and discussion could be merged into a single chapter. However, the current chapter structure suggests a clear separation between results and discussion, whereas in the text, these two sections blur together, and the actual discussion is ultimately rather brief.**

- **Author's response:**

Thank you for your suggestion. We have reorganized the structure of the manuscript.

- **Author's changes in manuscript:**
We moved the "4.1 Rapid glacier changes" section to the study area description, as it is based on literature review. The "5 Multi-periodic Sedimentation in the Confluence" section was incorporated into "4 Results." The final "4 Results" section is divided into three subsections: "4.1 Multi-periodic glacial debris flows," "4.2 Sediment characteristics of the 2020 event," and "4.3 Multi-periodic sedimentation in the confluence." We consider these three subsections to present direct conclusions derived from our methods, including the basic characteristics of historical debris flows, detailed sedimentary features of the 2020 event, and the interpreted distribution of non-vegetated areas (NVA).
In the Discussion, the original "6.1 The dominant factor and future risk" was split into "5.1 The dominant factor for debris flows and sediment yield" and "5.2 Debris flow recurrence intervals and future risk." We provide a more in-depth discussion of earthquake and climate influences, linking different events, NVA changes, and the effects of climate and seismic activity. In "5.2 Debris flow recurrence intervals and future risk" and "5.3 Effects on river geomorphology," we performed comparative analyses addressing reviewer comments on recurrence intervals, sediment mobilization, and downstream impacts on the river system. A new subsection, "5.4 Limitations," was added to discuss methodological uncertainties and directions for future research.

7. **Conclusion:**

**This manuscript represents a significant contribution to the study of periglacial mass movements in high mountain regions. It is methodologically sound, thoroughly documented, and addresses a relevant topic in the context of climate change and geomorphological dynamics. The study is of great interest to the mountain and environmental geoscience community. After minor revisions regarding uncertainty analysis, structure, and clearer framing of climatic controls, the publication is fully recommended.**

- **Author's response:**

We are grateful for the reviewer's positive recognition of our manuscript. We greatly appreciate the constructive comments and have revised the manuscript accordingly.

**Reviewer 2**

Thank you for your appreciation! And we are very grateful for various lines of constructive criticism which were very helpful in improving analysis and the manuscript. Point-by-point replies can be found in the following:

*(Please note that all references in this response, including line numbers, section titles, and figure/table numbers, are based on the Manuscript_trackchanges.docx file, so that the reviewer can clearly locate the revisions and compare them with the original content.)*

**1. Page 3, Line 87. average gradient of 27.5% (not 275%)**

● **Author's response:**

Thank you for your careful review. Due to our oversight, we mistakenly wrote ‰ instead of %, and we have now corrected it.
● **Author's changes in manuscript:**
We have corrected "275%" to "275‰".

**2. Page 15, line 96. better: stopped quickly.**

● **Author's response:**

Thank you for your suggestion. We agree that "stopped quickly" is better. We have revised "quickly stopped" to "stopped quickly" accordingly.
● **Author's changes in manuscript:**
We revised "quickly stopped" to "stopped quickly".

**3. Page 20, line 70. Lateral erosion took place ...**

● **Author's response:**

Thank you for your suggestion. We agree that "took place" is better. We have revised "happened" to "took place" accordingly.
● **Author's changes in manuscript:**
We revised "happened" to "took place".

**4. Page 20, line 74. Could part of the material be submerged? more likely it was eroded and washed away by powerful Yarlung Tsangpo stream.**

● **Author's response:**

Thank you for your suggestion. Yes, our consideration here was indeed not comprehensive. In fact, the ZLL debris-flow deposits contain a high proportion of boulders and coarse particles (as noted in the section 4.2.2), so some of the boulders may have been temporarily submerged by the river, but they are unlikely to be transported away by the strong flow of the Yarlung Tsangpo

River. Meanwhile, the fine-grained sediments within the debris flow could indeed be eroded and carried away by the main river, resulting in partial sediment loss. We have accordingly revised the relevant statements in the main text.

● **Author's changes in manuscript:**

We revised Lines 423–425 as follows:

the true volume may be seriously underestimated because part of the sediment may be submerged or washed away by the Yarlung Tsangpo River, which is a bias caused by the difference in data acquisition time and the errors associated with DoD processing.

**5. Page 21, line 76. Multi-periodic sedimentation ...**

● **Author's response:**

Thank you for your careful review. We have changed the initial letters of non-sentence-initial words to lowercase.

● **Author's changes in manuscript:**

We revised "Sedimentation" to "sedimentation".

**6. Page 22, line 11. In such case more important is not the distance from the epicenter, but distance from the earthquake source (causative fault). Energy is released not from the point (hypocenter), but from the entire source.**

● **Author's response:**

We agree with your comment. The maximum distance from the fault rupture zone to landslides indeed provides a more accurate upper bound, since seismic energy is released along the entire fault. We cited the long-distance effects of the 2015 Gorkha earthquake to illustrate the possible influence of the 1950 Assam earthquake on ZLL; however, we acknowledge that this comparison may not be appropriate, given the potential differences in causative fault length between earthquakes.

Nevertheless, previous studies have confirmed the strong impact of the 1950 Assam earthquake on the ZLL region. The 1950 Assam earthquake occurred at the Assam sub-Himalayan syntaxis and was triggered by rupture of the Mishmi Thrust (MT) and Main Himalayan Frontal Thrust (MFT). Coudurier-Curveur et al. (2020) recalibrated the aftershock distribution within the first four months following the mainshock and revealed proximal aftershocks near the ZLL basin (about 1 km away) with magnitudes up to Mw 5.5 (Fig. 1b). The 1950 earthquake directly caused severe damage to houses and temples in this region. The northernmost aftershocks further suggest that the influence of the MT fault extended beyond its distance to ZLL (Fig. 1b).

In the absence of detailed fault information, the epicentral distance curve can still serve as a quick and approximate estimation tool. For this reason, we

have retained Figure 13 in the revised manuscript.

[Figure]

Figure 1: (a) Regional overview map of southeastern Tibet. (b) Regional settings of the eastern syntaxis of Himalayas. Fault data were obtained from Wu et al. (2024). Historical earthquakes from 1940 to 2020 were sourced from the United States Geological Survey (USGS) National Earthquake Information Center (NEIC) (https://earthquake.usgs.gov/earthquakes/search/). Relocations of aftershocks within the first four months following the 1950 Assam mainshock were taken from Coudurier-Curveur et al. (2020).(MFT: Main Himalayan Frontal Thrust, MT: Mishmi Thrust, DBF: Damu-Bianba Fault, DDKF: Daduka Foult) (c) Topographic, geological and glacier terminus change maps of the Zelunglung catchment. The lithological information is based on Zhang and Shen (2011), and the glacier map is derived from the RGI 7.0 dataset (RGI, 2023). The orange, rose-red, green, black and red coloured arrows represent the view angle direction of figures 1d, 5a, 5b, 6c and 6d. (d) Aerial photo of the Zelunglung glacier and channels on December 21, 2022.

● **Author's changes in manuscript:**

We deleted "Notably, the impact distance of a large earthquake can reach hundreds of kilometers. For example, the co-seismic landslides triggered by the 2015 Gorkha Mw 7.8 earthquake extended to a distance of over 130 km from the epicenter (Martha et al., 2017). The 1950 Assam earthquake, with its epicenter approximately 199 km from the ZLL, had a very high magnitude (Mw 8.6) and occurred in the tectonically active eastern Himalayan syntaxis. Coupled with subsequent high-magnitude aftershocks near the ZLL (Fig. 13), the seismic impact on the ZLL was significantly amplified despite the distance". (L461-466).

We added a sentence in L458-459 as follows: In the absence of detailed fault information, we conducted a rapid and preliminary assessment of the impacts of historical earthquakes using this curve.

We added a sentence in L468-470 as follows: Relocated aftershocks of the 1950 earthquake (Coudurier-Curveur et al., 2020) indicate that the seismogenic faults—the MFT and MT—extend their influence well beyond the ZLL (Fig. 1b).

We have added the relocated aftershock distribution of the 1950 Assam earthquake from Coudurier-Curveur et al. (2020) in Fig. 1b.

**Other changes by the author**

1. Owing to the reorganization of the institute, the original first affiliation "Key Laboratory of Mountain Hazards and Earth Surface Processes, Chinese Academy of Sciences, Chengdu, 610041, China" has been removed. In addition, as the institute has relocated, the postal code has been updated to "610213."

2. Figures 4 and 5 have been merged into a single figure.
3. In addition to the language expression problems pointed out by the reviewers, we further checked and proofread the possible language errors.

**Reference**

[revised manuscript text omitted]

---

## Author Response (AR4)

**Reviewer**

**I have now received to reviews for your manuscript entitled "Variation of sediment supply by periglacial debris flows at Zelunglung in the eastern syntaxis of Himalayas since the 1950 Assam Earthquake". Both reviewers highlight the high quality of your work, in particular noting the integration of diverse data. While the second referee requests only a few techical corrections, the first reviewer has a few suggestions for further improving your work in particular concerning the quantification of uncertainties. Also, the reviewer notes that there should be a better separation of the results and discussion chapter.**

**To go forward with your manuscript, please address the points raised by the reviewers. I am looking forward to your revisions in due time.**

Thank you for your constructive suggestions. In response to your comments regarding the quantification of uncertainties and the structure of the manuscript (results and discussion chapter), we have made targeted revisions, which are detailed in the following point-by-point responses:

**1. Quantification of Uncertainties**

● **Author's response:**

Thank you for your constructive suggestions. We have made the following targeted revisions:

1) In Section 3.2.2 "Drone image interpretation," we clarified the quantification methods for DoD uncertainty and volume estimation uncertainty.
2) In Section 4.2 "Sediment characteristics of the 2020 event," we provided the arithmetic mean, RMSE, maximum, minimum, and standard deviation of the DoD values in the stable areas, and quantified the uncertainty of the volume estimation using the two-sigma DoD uncertainty. In addition, we updated Figure 9 (formerly Figure 10) to display the DoD values of the stable areas.
3) In Section 5.5 "Uncertainties," we discussed the sources of errors in DoD and volume estimation, as well as the uncertainties associated with NVA interpretation.

● **Author's changes in manuscript:**
We have revised or added descriptions in the following sections:

[revised manuscript text omitted]

**2. The reviewer notes that there should be a better separation of the results and discussion chapter.**

- **Author's response:**

Thank you for your constructive comments. We have moved the extended discussion that was previously included in the Results chapter to a newly added Section 5.1, "Erosion and sedimentation of periglacial debris flows," in the Discussion chapter. The revised Results chapter now objectively presents the characteristics of historical debris-flow events (based on literature review), the features of the 2020 event, and the NVA changes on the depositional fan, focusing on the "data, imagery, and trends" directly derived from the applied methodologies. The newly added Section 5.1 mainly discusses the depositional characteristics and mechanisms of glacial debris flows based on the sedimentary phenomena observed during the 2020 event.

Due to these structural changes, some figures have been updated, and their order within the manuscript has also been adjusted.

- **Author's changes in manuscript:**

Due to the structural adjustments, presenting all the changes here would be redundant. Since the Results chapter was only streamlined and its logical order reorganized, we focus here on presenting the newly added discussion content here. For detailed revisions to the Results chapter, please refer to the manuscript.

[revised manuscript text omitted]